# From Ticks to Flows: Dynamics of Neural Reinforcement Learning in Continuous Environments

**Saket Tiwari**[*]
Department of Computer Science
Brown University

**Tejas Kotwal**
Department of Applied Mathematics
Brown University

**George Konidaris**
Department of Computer Science
Brown University

## Abstract

We present a novel theoretical framework for deep reinforcement learning (RL) in continuous environments by modeling the problem as a continuous-time stochastic process, drawing on insights from stochastic control. Building on previous work, we introduce a viable model of actor–critic algorithm that incorporates both exploration and stochastic transitions. For single-hidden-layer neural networks, we show that the state of the environment can be formulated as a two time scale process: the environment time and the gradient time. Within this formulation, we characterize how the time-dependent random variables that represent the environment's state and estimate of the cumulative discounted return evolve over gradient steps in the infinite width limit of two-layer networks. Using the theory of stochastic differential equations, we derive, for the first time in continuous RL, an equation describing the infinitesimal change in the state distribution at each gradient step, under a vanishingly small learning rate. Overall, our work provides a novel nonparametric formulation for studying overparametrized neural actor-critic algorithms. We empirically corroborate our theoretical result using a toy continuous control task.

## 1 Introduction

A reinforcement learning agent (RL) (Sutton & Barto, 1998) learns to behave intelligently by interacting with an environment to maximize the rewards it receives. Neural networks have contributed significantly to the improvement and advance of RL in the recent past. An agent equipped with neural networks and trained using RL can effectively learn intelligent behavior by optimizing the rewards it receives over time. Neural networks, in conjunction with RL, have been effective not only for simulated arcade games (Mnih et al., 2013; 2015) and robotic control (Lillicrap et al., 2016) but have also been employed for real-world robotic control problems (Levine et al., 2016; Zhu et al., 2020; Song et al., 2023; Kaufmann et al., 2023). One of the most popular subclasses of deep RL algorithms is the actor critic framework with neural network function approximations (Sutton et al., 1999; Silver et al., 2014; Lillicrap et al., 2015; Haarnoja et al., 2018). The incorporation of neural networks into reinforcement learning harnesses their universal function approximation capabilities, allowing agents to model and navigate a wide range of environments. Despite these advances and the development of numerous algorithms, a gap remains in our theoretical understanding of deep RL.

A related field of study is deep supervised learning which also employs neural networks to solve stationary problems. We have seen several new theories explaining their efficacy in a supervised learning setting (Jacot et al., 2018; Allen-Zhu et al., 2019a; Roberts et al., 2021; Couillet & Liao, 2022) as to why neural networks that are highly overparameterized or "wide" and "deep" are successful in approximating functions (Krizhevsky et al., 2012; He et al., 2016; Szegedy et al., 2017) learned using gradient updates. One common approach to deep learning theory is to study them in the limit of the

---

[*]Corresponding author: sakett786@gmail.com

width tending to infinity (Cybenko, 1989; Lee et al., 2017; Mei et al., 2018; Neyshabur et al., 2018; Jacot et al., 2018). One of the most critical and effective features of these models of neural networks is to model the inputs, parameters, and outputs as a probability distribution and individual activation being a sample drawn from this distribution. Theories exploring the evolution of these distributions through training phases, particularly as they undergo gradient updates, provide insightful perspectives for neural networks (Yang & Hu, 2020; Berthier et al., 2024; Ben Arous et al., 2022).

Despite recent work explaining the efficacy of deep RL (Cai et al., 2019a; Wang et al., 2019; Agarwal et al., 2021; Lyle et al., 2022b), its success in control tasks remains largely unexplained from a theoretical perspective. Although there have been some theoretical analyses in RL with continuous states and actions (Fazel et al., 2018; Lutter et al., 2021; Huang et al., 2024), progress in developing a theory of continuous control with neural network function approximation has been limited. The primary challenge that accounts for gaps in deep RL theory, despite having numerous theoretical results in deep supervised learning, is that the data distribution also changes with gradient steps in RL. To better understand the learning process of an RL agent, we would like to ideally characterize how the distribution changes as the agent learns with gradient-based updates.

It is easier to describe how the state distribution changes from moment to moment than to describe the full evolution over *environment time*. This is the philosophy behind the study of stochastic differential equations (SDEs) in the continuous setting. These ideas have been successfully applied to optimal control of continuous systems (Kushner & Dupuis, 2001; Oksendal, 2013). We adopt this idea to theoretically derive equations for how the state distribution changes locally at small gradient steps. We combine methods from deep learning theory that study change over gradient steps for NNs and methods from control theory that study the change in environment time to provide equations that encapsulate changes across both time scales: environment and gradient.

We formulate agent's learning in continuous state and action environments using the continuous-time actor-critic framework provided by Jia & Zhou (2022), with fixes to exploratory dynamics. We show that our exploratory dynamics can be simulated in discrete time while remaining faithful to the underlying continuous-time process, with a single source of noise that is equivalent to a system with both environment and exploration noise. We use a linearized single hidden layer NN, for both actor and critic, as a theoretical model to study over-parameterized NNs (Lee et al., 2019; Cai et al., 2019b). One of our key insights is to use the Itô -Taylor expansion (Kloeden & Platen, 1992) to present the time-dependent state variable as a polynomial in the parameters of the linearized NN and thereby tracking the changes in this polynomial expression. Combined with the Gaussian nature of the neural network outputs (Lee et al., 2017), we are able to present a nonparametric equation that captures the changes in the state of the environment over both the environment time and the gradient steps, up to an error term. Our main result shows that, strikingly, this closed system has only five time-dependent variables which describe one step gradient change. We empirically corroborate the exploratory nature of our simulation and also demonstrate that the RL agent is able to learn a near optimal policy using the episodic actor-critic which we analyze, in the linear quadratic regulator (LQR) environment.

## 2  STOCHASTIC PROCESSES

To formalize the idea behind stochasticity in continuous environments, we introduce continuous time stochastic processes. A one-dimensional Wiener process, taking values in the Euclidean space $\mathbb{R}$, is one of the central building blocks of the theory of stochastic processes (Karatzas & Shreve, 2014; Oksendal, 2013).

**Definition 2.1.** *A stochastic process $w_t$ is called a Wiener process if the sample paths of $w_t$ are almost surely continuous square-integrable Martingale with $w_0 = 0$ and $\mathbb{E}[(w_t - w_s)^2] = t - s$.*

The multidimensional Wiener process is a concatenation of such single-dimensional processes. Such a process has stationary independent increments, and that makes it ideal to model noise in the environment. The general form of a time invariant stochastic differential equation (SDE) in $\mathbb{R}^k$ is

$$dX_t = b(X_t)dt + \sigma(X_t)dw_t, \tag{1}$$

where $w_t$ is an $m-$dimensional Wiener process, $X_t$ is the random variable corresponding to the random variable $X$ at time $t$, $b$, which is the *drift* component of the equation, is a function such that $b : \mathbb{R}^k \to \mathbb{R}^k$ and $\sigma$ is another function such that $\sigma : \mathbb{R}^k \to \mathbb{R}^m$. $b$ determines the direction of the deterministic part of the transition dynamics and $\sigma$ that of the stochastic part. The solution to this

SDE, under certain conditions over $b$ and $\sigma$, is obtained using the Itô integral. The natural filtration generated by $X = \{X_t, t \geq 0\}$ is denoted by $\{\mathcal{F}_t\}_{t \geq 0}$ (see Section A for definitions).

For an equally spaced partition of the time intervals $0 = t_0 < t_1 < t_2 \ldots < t_n < \ldots < T$, consider the discrete summation.

$$X_{t_n}^{\Delta t} = X_0 + \sum_{j=0}^{n-1} b(X_{t_j}) \Delta t + \sum_{j=0}^{n-1} \sigma(X_{t_j}^{\Delta t}) \Delta w_j, \tag{2}$$

where $t_{j+1} - t_j = \Delta t > 0$ and $\Delta w_j = w_{t_{j+1}} - w_{t_j} \sim \mathcal{N}(0, \Delta t)$. The limit as $\Delta t \to 0$, of the right-hand side of the equation, when it exists, defines the Itô integral in probability:

$$X_t = X_0 + \int_0^t b(X_l)dl + \int_0^t \sigma(X_l)dw_l. \tag{3}$$

See Section A for details on the conditions under which the solution to equation 1 exists. In RL, such a process can be used to define the moment-to-moment changes in the environment's state such that the transitions have independent noise added to them and solution to the equation represents the time-dependent random variables.

## 3 CONTINUOUS-TIME REINFORCEMENT LEARNING

Commonly RL in discrete time models environment data that correspond to ticks: the agent observes the state of the environment, takes an action that changes the state of the environment, and receives a reward. Instead, we consider a continuous-time model of RL (Baird, 1994; Doya, 2000; Wang et al., 2020; Jia & Zhou, 2021). Several continuous-time formulations already exist, our approach is distinct in being explicitly exploratory: both the policy and the environment contribute to the transition noise. It is also structured in a way that makes analysis with neural networks more tractable. We define continuous-time reinforcement learning in a control affine *Markov decision process* (MDP) which is defined by the tuple $\mathcal{M} = (\mathcal{S}, \mathcal{A}, \langle g, h, \sigma \rangle, r, s_0, \beta)$ over time $t \in [0, T)$. Here, $\mathcal{S} \subseteq \mathbb{R}^{d_s}$ is the set of all possible states of the environment and $s_0 \in \mathcal{S}$ is the state of the environment at the start time. $\mathcal{A} \subseteq \mathbb{R}^{d_a}$ is the set of actions available to the agent. $r : \mathbb{R}^{d_s} \to \mathbb{R}$ is the reward function that determines the reward the agent receives in a given environment state. $\beta \in (0, 1)$ is the discounting factor that ensures future rewards are less valuable than current rewards. $g : \mathbb{R}^{d_s} \to \mathbb{R}^{d_s}$ and $h : \mathbb{R}^{d_s} \to \mathbb{R}^{d_s \times d_a}$ represent the deterministic component of the transition dynamics. $\sigma : \mathbb{R}^{d_s} \to \mathbb{R}^{d_s \times d_s}$ accounts for the stochasticity of the transition in any given state, which is assumed to be independent of the action. At time $t$ and infinitesimally small time discretizations, the agent's state, $s$ changes according to the following SDE:

$$ds_t = (g(s_t) + h(s_t)a_t) \, dt + \sigma(s_t)dw_t,$$

where $w_t$ is a $d_s$-dimensional Wiener process and action $a_t$. We further assume that $g, h, \sigma, r$ are all smooth, meaning infinitely differentiable. Although this may seem restrictive, the study of smooth functions, which have an analytical form, has been used in theoretical settings to better understand both control systems (Jurdjevic, 1997) and neural networks (Montanari & Subag, 2024).

The agent is equipped with a smooth policy $\pi : \mathcal{S} \to \mathcal{A}$ that determines its decision making process. A policy determines the action the agent takes in a state. This is similar to feedback control in control theory. The SDE corresponding to a policy $\pi$ is therefore obtained by replacing $a_t = \pi(s_t)$. which has a unique solution in probability under Lipschitz continuity of the dynamics. The time-dependent state random variable is defined on a filtered probability space $(\Omega, \mathcal{F}, \mathbb{P}^W; \{\mathcal{F}_t^W\}_{t \geq 0})$. Therefore, the Itô integral solution, as in equation 3, of the above SDE is denoted by $s_t^\pi$.

The state value function, in context of RL, is defined as

$$v^\pi(s, t) = \mathbb{E} \left[ \int_t^T e^{-\beta(l-t)} r(s_l^\pi)dl \, \Big| \, s_t = s \right],$$

which is the expected cumulative discounted rewards given that the agent starts in state $s$ (or $s_l = s$) and follows policy $\pi$ from time $t$ until it terminates at time $T$. The expectation is on the stochasticity of the environment dynamics. The agent's goal is to maximize the objective $J(\pi) = v^\pi(s_0, 0)$ by

learning the optimal policy $\pi^* \in \Pi$, where $\Pi$ is the family of policies available to the agent, e.g. the set of neural networks with one hidden layer. Unlike in control theory, the agent does not have access to the dynamics of the system: $g, h, \sigma$ and optimizes its policy by collecting data points. These data points are in the form of indexed state, action, and reward tuples by time. To collect these data, the agent *explores* different parts by taking random actions. Therefore, we also define the following SDE:

$$d\hat{s}_t^\pi = (g(\hat{s}_t^\pi) + h(\hat{s}_t)\pi(\hat{s}_t^\pi)) \, dt + h(\hat{s}_t)dw_t' + \sigma(\hat{s}_t^\pi)dw_t,$$

which has noise from policy $w_t'$ and from the environment $w_t$. For this exploratory SDE to be effective, we need to justify a numerical scheme where the exploratory noise is associated with the policy and can be simulated in a discrete time. Therefore, for fixed $\Delta t > 0$, and deterministic policy $\pi$ consider the following numerical scheme (similar to Equation 2):

$$s_{t_n}^{\Delta t, \pi} = s_0 + \sum_{j=1}^{n-1} \left( g(s_{t_j}^{\Delta t, \pi}) + h(s_{t_j})(\pi(s_{t_j}^{\Delta t, \pi}) + \Delta b_j) \right) \Delta t + \sigma(s_{t_j}^{\Delta t, \pi})\Delta w_j,$$

$$\text{where } \Delta w_j \sim \mathcal{N}(0, \Delta t), \Delta b_j \sim \mathcal{N}(0, 1/\Delta t), \Delta t = t_j - t_{j-1} \forall j \in \mathbb{N}.$$

For $d_s = d_a = 1$ we prove the equivalence of the exploratory dynamics and above the numerical scheme. Proving results in 1D is a common approach in numerical methods for control theory (Kushner & Dupuis, 2001) for simplicity and tractability. We anticipate that higher dimensional results follow similarly.

**Lemma 3.1.** *Suppose that $g, h, \sigma$ and $\pi$ are Lipschitz continuous and satisfy linear growth condition:*

$$||g(x)|| \le K_g(1 + |x|), ||h(x)|| \le K_h(1 + |x|), ||\pi(x)|| \le K_\pi(1 + |x|),$$

*then $s_t^{\Delta t, \pi} \to s_t$ weakly where $s_t^\pi$ is solution to the SDE:*

$$ds_t^\pi = (g(s_t^\pi) + h(s_t^\pi)\pi(s_t^\pi))dt + h(s_t^\pi)dw_t' + \sigma(s_t^\pi)dw_t.$$

*Moreover, the solution to this SDE has the same pathwise distribution as the following SDE:*

$$d\tilde{s}_t^\pi = (g(\tilde{s}_t^\pi) + h(\tilde{s}_t^\pi)\pi(\tilde{s}_t^\pi))dt + \sqrt{h(\tilde{s}_t^\pi)^2 + \sigma(\tilde{s}_t^\pi)^2}dw_t. \tag{4}$$

The proof is in Section B. We refer to equation 4 as exploratory dynamics and use it in our analysis as a proxy for the state random variable under exploration. We depart from the *relaxed-control* formulation of exploratory dynamics introduced by Wang et al. (2020) because, in that model, the policy's stochasticity vanishes whenever the environment is deterministic (i.e. $\sigma(\cdot) = 0$ almost everywhere). Given these dynamics and the objective, the agent's goal is to learn an optimal policy from a set of admissible policies.

## 4 CONTINUOUS-TIME ACTOR CRITIC

Actor-Critic algorithms (Sutton & Barto, 1998) train an agent with an *actor* that is the decision making entity and a *critic* that is an estimate of the value function that guides the improvement of the policy. Algorithms from this family learn the critic and the actor alternately using samples from the rollouts of the policy. In deep RL both are parameterized by a neural network. As demonstrated in a series of papers on continuous-time RL (Wang et al., 2020; Jia & Zhou, 2021; 2022; 2023) the gradient updates for learning the actor and policy are different compared to discrete-time algorithms. We adapt the results presented by Jia & Zhou (2021; 2022) to develop an algorithm for actor-critic learning in a continuous-time setting under our exploratory dynamics. We first define an admissible policy as follows:

**Definition 4.1.** *A policy $\pi : \mathbb{R}^{d_s} \to \mathbb{R}^{d_a}$, which is a function that maps from $\mathbb{R}^{d_s}$ to $\mathbb{R}^{d_a}$ is called admissible if:*

1. *The function $\pi$ is smooth everywhere.*

2. *The SDE (equation 4) admits a weak solution in the sense of probability (see Section A.1).*

3. *$\pi(x)$ is uniformly Lipschitz continuous in $x$, which means that there exists a constant $C > 0$ such that:*
$$||\pi(x) - \pi(x')|| \le C||x - x'||.$$

Furthermore, let $\mathcal{L}^\pi$ be the infinitesimal generator associated with the process in Equation 4:

$$\mathcal{L}^\pi f(x,t) = \frac{\partial f(x,t)}{\partial t} + (g(x) + h(x)\pi(x)) \circ \frac{\partial f(x,t)}{\partial x} + \frac{1}{2}\tilde{\sigma}(x)^2 \circ \frac{\partial^2 f(x,t)}{\partial x^2},$$

which captures the local change over time in a function $f$. We also make the assumption that $\tilde{\sigma}(\cdot) = \sqrt{h(\cdot)^2 + \sigma(\cdot)^2} \neq 0$ almost everywhere in $\mathbb{R}^{d_s}$. We state a result by Jia & Zhou (2022) in our deterministic policy setting under exploratory dynamics (Equation 4) which provides a relationship between the solution to an equation with the infinitesimal generator and the value function corresponding to an admissible policy.

**Lemma 4.2.** *Assume that there is a unique viscosity solution $v \in C(\mathbb{R}^{d_s} \times [0,T])$ to the following partial differential equation (PDE):*

$$\mathcal{L}^\pi v(x,t) + r(x, \pi(x)) - \beta v(x,t) = 0,$$

*with terminal condition $v(x,T) = 0$, $x \in \mathbb{R}^{d_s}$, which satisfies $|v(x,t)| \leq C(1 + ||x||)^\mu$ for some constants $\mu, C$. Then $v$ is the value function corresponding to admissible policy $\pi$, that is, $v(x,t) = v^\pi(x,t)$ for all $(x,t) \in \mathbb{R}^{d_s} \times [0,T]$.*

In policy gradient setup, the agent's policy is parameterized by parameters $\theta$ and the agent learns by taking gradient steps in the direction of steepest ascent of the objective: $J(\pi^\theta)$. We will denote this by $J(\theta)$. The deterministic policy analog of Theorem 2 by Jia & Zhou (2022), for $d_a = d_s = 1$, gives a policy update formulation similar to the discrete time policy gradient (Sutton et al., 1999; Silver et al., 2014) in a continuous time setting. Since estimating the expectation above requires multiple trajectories, in practice the agent learns in a stochastic manner by sampling a single trajectory and updating the parameters based on it. In addition, the value estimate is parameterized by parameters $\phi$. This is called episodic RL. Gradient-based updates for the estimate of value estimate parameters and policy, which bear similarities to coagent networks (Thomas, 2011; Kostas et al., 2020), are as follows:

$$\widehat{\mathbb{G}}(\phi, \theta) = \int_0^T e^{-\beta l} \frac{\partial v(\tilde{s}_l^{\pi^\theta}, l; \phi)}{\partial \phi} \left[ \partial_t v(\tilde{s}_l^{\pi^\theta}, l; \phi) + r(\tilde{s}_l^{\pi^\theta}) - \beta v(\tilde{s}_l^{\pi^\theta}, l; \phi) \right] dl,$$

$$\widehat{\mathcal{G}}(\phi, \theta) = \int_0^T e^{-\beta l} \frac{\partial \pi(\tilde{s}_l^{\pi^\theta}; \theta)}{\partial \theta} \left[ \partial_t v(\tilde{s}_l^{\pi^\theta}, l; \phi) + r(\tilde{s}_l^{\pi^\theta}) - \beta v(\tilde{s}_l^{\pi^\theta}, l; \phi) \right] dl. \tag{5}$$

---

**Algorithm 1** Episodic Actor-Critic (Continuous-Time Gradients)

---

**Inputs:** initial state $s_0$, horizon $T$, time step $\Delta t$, number of episodes $N$, number of mesh grids $K = \lfloor T/\Delta t \rfloor$, and a learning rate $\eta$, discount $\beta$, the value $v(s, t; \phi)$ and policy $\pi(s; \theta)$.
**Required:** an environment simulator $(s', r) = \text{ENVIRONMENT}(s, a, \Delta t)$ according to the dynamics 4 that maps $(s, a)$ to next state $s'$ and reward $r$ at time $t$.

1: Initialize $\theta, \phi$.
2: **for** episode $j = 1$ to $N$ **do**
3:     Initialize $k \leftarrow 0$. Observe $x_0$ and set $s_{t_k} \leftarrow x_0$.
4:     **while** $k < K$ **do**
5:         Generate action $a_{t_k} = \pi(x_{t_k}; \theta)$.
6:         Apply $a_{t_k}$ in environment: $(s_{t_{k+1}}, r_{t_k}) \leftarrow \text{ENVIRONMENT}_{\Delta t}(t_k, s_{t_k}, a_{t_k})$.
7:         $k \leftarrow k + 1$.
8:     **end while**

9:     **Compute continuous-time gradient estimates:**
10:     For $t_i = i\Delta t$, define

$$\delta_i = \partial_t v(s_{t_i}, t_i; \phi) + r_{t_i} - \beta v(s_{t_i}, t_i; \phi).$$

11:     Then set

$$\Delta\phi = \sum_{i=0}^{K-1} e^{-\beta t_i} \frac{\partial v(s_{t_i}, t_i; \phi)}{\partial \phi} \delta_i \Delta t, \qquad \Delta\theta = \sum_{i=0}^{K-1} e^{-\beta t_i} \frac{\partial \pi(s_{t_i}; \theta)}{\partial \theta} \delta_i \Delta t.$$

12:     **Update (value estimate and policy parameters):**

$$\phi \leftarrow \phi + \eta \, \alpha_\phi \, \Delta\phi, \qquad \theta \leftarrow \theta + \eta, \alpha_\theta \, \Delta\theta.$$

13: **end for**

---

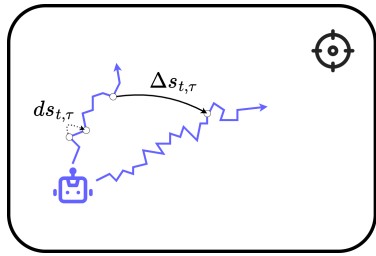

Figure 1: We illustrate $\Delta s_{t,\tau}$ using an agent (the robot in blue) whose goal is to reach the target in the top right corner, starting from the bottom left. The jagged blue trajectories correspond to its non-smooth stochastic paths in the environment. At gradient time $\tau$, the agent follows the trajectory on the left, and after one gradient step, it moves along the trajectory on the right, closer to the goal. While the moment-to-moment change in environment time is represented by Equation 4 (dotted curve), Theorem 6.1 provides an expression for the change over a gradient step, $\Delta s_{t,\tau}$ (solid curve).

## 5 LINEARIZED TWO-LAYER NEURAL NETWORKS

In deep reinforcement learning both actor and critic are parametrized by neural networks. Often times, to study a complex object such as a neural network, researchers utilize a simplified model (Lee et al., 2019; Cai et al., 2019b; Arora et al., 2019) that makes the problem tractable. We consider two-layer feedforward neural networks with $\tanh$ activations, $\varphi$, to ensure smoothness of dynamics. The network output is given by:

$$F(s; W, C) = \frac{1}{\sqrt{n}} \sum_{\kappa=1}^{n} C_\kappa \varphi(W_\kappa \cdot s), \tag{6}$$

where $W = [W_1, \ldots, W_n] \in \mathbb{R}^{nd_s}$, $C = [C_1, \ldots, C_n] \in \mathbb{R}^{d_a \times n}$. The parameters are initialized as $C_k \sim \mathrm{Unif}(-1, 1)$, $W_k \sim \mathcal{N}(0, I_{d_s}/d_s)$. During training, only $W$ is updated, while $C$ is fixed. Let $W^0$ be the initialization of the first layer. The linearized approximation of the policy for wide neural networks (Allen-Zhu et al., 2019b; Gao et al., 2019; Lee et al., 2019) around $W^0$ is:

$$F_\pi^{\mathrm{lin}}(s; W) = F_\pi(s; W^0) + \Phi(s; W^0)(W - W^0), \tag{7}$$

with $\Phi(s; W^0) = \frac{1}{\sqrt{n}} \left[ C_1^0 \varphi'(W_1^0 \cdot s)s^\top, \ldots, C_n^0 \varphi'(W_n^0 \cdot s)s^\top \right] \in \mathbb{R}^{d_a \times nd_s}$. This formulation is linear in $W$, and nonlinear in $s$ and $W^0$, in addition to being admissible. The linearized value estimate function: $F_v^{\mathrm{lin}}(s, t; U)$ is defined in a similar way (see Section E). Moreover, Tiwari et al. (2025) have shown empirically that the linearized two-layer NN performs similarly to the canonical NN in the complex and non-linear MuJoCo Cheetah environment at very large widths. We assume $\tanh$ activations because they are symmetric and smooth. Further, it is assumed that the learning rate $\eta$ scales as $O(1/\sqrt{n})$. Under this parameterization, the gradient-based updates (Equation 5) are denoted as $\widehat{\mathbb{G}}(U, W), \widehat{\mathcal{G}}(U, W)$. Equipped with this parameterization, we state our main result about the gradient time dynamics of state variable.

## 6 MAIN RESULT: CHANGE IN STATE WITH GRADIENT STEP

A natural question to ask is: *how does the time dependent state random variable change over learning steps in actor-critic setting?* Understanding this change moment to moment, meaning from one gradient update to another, would give us an idea of how the agent learns. To do so we describe the *gradient dynamics* as follows, at gradient time step $\tau$ the parameters of the policy ($W^\tau$) and the value estimate ($U^\tau$) are updated as:

$$W^{\tau+\eta} = W^\tau + \eta \widehat{\mathcal{G}}(U^\tau, W^\tau), \quad U^{\tau+\eta} = U^\tau + \eta \widehat{\mathbb{G}}(U^\tau, W^\tau).$$

Since the environment state is also a function of $W^\tau$, the state depends on both the environment time $t$ and the gradient time step $\tau$: $s_{t,\tau}$. Although its dynamics in environment time are given by Equation 4 its dynamics in gradient time are also governed by the actor-critic algorithm described in Section 4 (see Figure 1). The agent therefore has two "clocks": environment clock and gradient clock. In Algorithm 1, the faster environment clock goes from 0 to $T$ whereas the slower gradient

clock only moves by $\eta$, in parallel. A scalar random variable is Gaussian up to an error of $O(1/\sqrt{n})$ when its distribution is close to the Gaussian cumulative distribution function (CDF), $\nu$, and satisfies: $\sup_{x \in \mathbb{R}} |\Pr(X_n < x) - \nu(x)| = O(1/\sqrt{n})$. In the setting introduced in previous sections, we are able to derive a closed system, that is, the changes in these variables over gradient steps depend only on one another.

**Theorem 6.1.** *An agent equipped with linearized single hidden layer policy and value estimate function of width $n$, $\tanh$ activation, under exploratory environment dynamics (Equation 4), where $g, h, \sigma$ are all smooth, with Lipschitz continuous smooth reward, $r$, that learns using episodic actor-critic updates and learning rate $\eta = O(1/\sqrt{n})$ has the following variables at gradient time $\tau$ and environment time $t$: the state $\tilde{s}_{t,\tau}$, action $a_{t,\tau} = F_\pi^{lin}(\tilde{s}_{t,\tau}; W^\tau)$, derivative of the action $a'_{t,\tau} = \partial_s F_\pi^{lin}(\tilde{s}_{t,\tau}; W^\tau)$, value estimate $v_{t,\tau} = F_v^{lin}(\tilde{s}_{t,\tau}, t; U^\tau)$ and time derivative of the value estimate $v'_{t,\tau} = \partial_t F_v^{lin}(\tilde{s}_{t,\tau}, t; U^\tau)$. Furthermore, suppose that neural networks are initialized i.i.d:*

$$\text{Policy parameters: } W_\kappa^0 \sim \mathcal{N}(0,1),\ C_\kappa^0 \sim Unif(-1,1)$$

$$\text{Value estimate parameters: } U_{\kappa,2}^0, U_{\kappa,2}^0 \sim \mathcal{N}(0,1),\ B_\kappa^0 \sim Unif(-1,1).$$

*Thus the law of $\tilde{s}_{t,\tau}, a_{t,\tau}, , v_{t,\tau}$ is defined with respect to both trajectory randomness and initialization. Denote by $q_{l,\tau} = v'_{l,\tau} + r(\tilde{s}_{l,\tau}) - \beta v_{l,\tau}$. Conditioned on the values of $\tilde{s}_{t,\tau}, a_{t,\tau}, a'_{t,\tau}, v_{t,\tau}, v'_{t,\tau}$, for $t \in [0, T]$, the change in these variables over a single gradient step: $\Delta v_{t,\tau}, \Delta v'_{t,\tau}, \Delta a_{t,\tau}, \Delta a'_{t,\tau}$, up to an error of $O(1/\sqrt{n})$, are as follows:*

$$\Delta v_{t,\tau} \text{ is Gaussian with mean } \eta \int_0^T e^{-\beta l} \mathbb{E}\left[B^2 \varphi'(U \cdot [\tilde{s}_{t,\tau}, t]) \varphi'(U \cdot [\tilde{s}_{l,\tau}, l])(\tilde{s}_{l,\tau}\tilde{s}_{t,\tau} + lt)q_{l,\tau}\right] dl,$$

$$\text{and variance } (\Delta s_{t,\tau})^2 \mathbb{E}\left[B^2 U_1^2 \left(\varphi''(U \cdot [\tilde{s}_{t,\tau}, t]) - \varphi''(U.[\tilde{s}_{t,\tau}, t])\left(U_1 \tilde{s}_{t,\tau} + U_2 t\right)\right)^2\right],$$

$$\Delta v'_{l,\tau} \text{ is Gaussian with mean } \eta \int_0^T e^{-\beta l} \mathbb{E}\left[B^2 U_2 \varphi''(U \cdot [\tilde{s}_{t,\tau}, t]) \varphi''(U \cdot [\tilde{s}_{l,\tau}, l])(\tilde{s}_{l,\tau}\tilde{s}_{t,\tau} + lt)q_{l,\tau}\right] dl$$

$$\text{and variance } (\Delta s_{t,\tau})^2 \mathbb{E}\left[B^2 U_1^2 U_2^2 \left(\varphi'''(U \cdot [\tilde{s}_{t,\tau}, t]) - \varphi'''(U.[\tilde{s}_{t,\tau}, t])\left(U_1 \tilde{s}_{t,\tau} + U_2 t\right)\right)^2\right],$$

$$\Delta a_{t,\tau} \text{ is Gaussian with mean } \eta \int_0^T e^{-\beta l} \mathbb{E}\left[C^2 \varphi'(W \tilde{s}_{t,\tau}) \varphi'(W \tilde{s}_{l,\tau}) \tilde{s}_{l,\tau}\tilde{s}_{t,\tau} q_{l,\tau}\right] dl$$

$$\text{and variance } (\Delta s_{t,\tau})^2 \mathbb{E}\left[C^2 W^2 \left(\varphi''(W \tilde{s}_{t,\tau}) - \varphi''(W \tilde{s}_{t,\tau}) W \tilde{s}_{t,\tau}\right)^2\right]$$

$$\Delta a'_{l,\tau} \text{ is Gaussian with mean } \eta \int_0^T e^{-\beta l} \mathbb{E}\left[C^2 W \varphi''(W \tilde{s}_{t,\tau}) \varphi'(W \tilde{s}_{l,\tau}) \tilde{s}_{l,\tau}\tilde{s}_{t,\tau} q_{l,\tau}\right] dl,$$

$$\text{and variance } (\Delta s_{t,\tau})^2 \mathbb{E}\left[C^2 W^4 \left(\varphi'''(W \tilde{s}_{t,\tau}) - \varphi'''(W \tilde{s}_{t,\tau}) W \tilde{s}_{t,\tau}\right)^2\right],$$

$$\text{where expectation is over } W \sim \mathcal{N}(0,1), C, B \sim Unif(-1,1),\ \text{and } U = [U_1, U_2] \sim \mathcal{N}(0,1).$$

*Finally, the change in $s_{t,\tau}$, is defined using $Z_{t,l,\tau}$:*

$$Z_{t,l,\tau} = Y_{t,\tau} \int_0^t Y_{u,\tau}^{-1} h(\tilde{s}_{u,\tau}) \mathcal{C}_{u,l,\tau} du, \text{ where } Y_{t,\tau} \text{ is solution to}$$

$$dY_{t,\tau} = (a_{t,\tau} + a'_{t,\tau}) Y_{t,\tau} dt + \sigma'(\tilde{s}_{t,\tau}) Y_{t,\tau} dw_t$$

$$\mathcal{C}_{u,l,\tau} = \mathbb{E}\left[C^2 \varphi'(\tilde{s}_{l,\tau} W) \varphi'(\tilde{s}_{u,\tau} W)\right], \text{ where } C \sim Unif(-1,1), W \sim \mathcal{N}(0,1), \text{ same as above.}$$

*Additionally, define $\mathbb{Z}_{t,\tau} = \int_0^t Z_{t,l,\tau} q_{l,\tau} dl$. Therefore, the change in the state random variable $\tilde{s}_{t,\tau}$ is:*

$$\Delta \tilde{s}_{t,\tau} = \eta \mathbb{Z}_{t,\tau} - M_{t,\tau} + G_{t,\tau} + O(1/n), \text{ where}$$

$$M_{t,\tau} = \tilde{s}_{t,\tau} - s_0 - \int_0^t (g(\tilde{s}_{u,\tau}) + h(\tilde{s}_{u,\tau}) a_{u,\tau}) du,$$

*and $G_{t,\tau}$ is a random variable and the martingale component of $x_{t,\tau}$, which follows the dynamics:*

$$dx_{t,\tau} = (g(x_{t,\tau}) + h(x_{t,\tau}) a_{t,\tau}) dt + \tilde{\sigma}(x_{t,\tau}) dw'_t,$$

*where $w'_t$ is an independent Wiener process and therefore $\mathbb{Z}_{t,\tau} = x_{t,\tau} - \mathbb{E}[x_{t,\tau}]$, where the expectation is over the random dynamics.*

The key takeaway is that the gradient time dynamics of an actor-critic algorithm with policy and value estimate parameterized by single hidden layer neural networks can be expressed as a closed system of five variables. As stated earlier, we provide the above result for the setting $d_a = d_s = 1$ as is common in control theory. We believe that high-dimensional results would require additional effort because both the policy gradient and the change is state variable are complicated with $d_s \times d_s$ terms. Nevertheless, our result are on how an agent learns with non-linear function approximations in a non-linear environment and allows us to present the main result in a tractable manner.

## 6.1 PROOF SKETCH AND INTERPRETATION

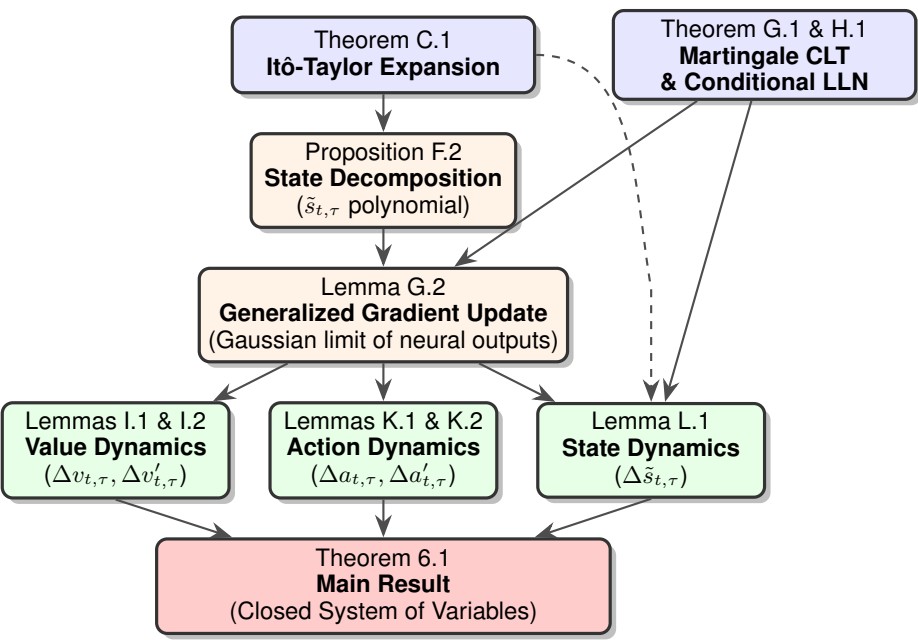

Figure 2: Flowchart illustrating the proof structure. The blue boxes denote the foundations of the proof and red is the main result, while rest are intermediate.

See Section M for the proof and Figure 2 for a flow chart. The proof begins by observing that the state random variable can be written as an infinite polynomial using the Itô–Taylor expansion (Kloeden & Platen, 1992), which is almost surely equivalent to the solution

$$d\tilde{s}_{t,\tau} = \big(g(\tilde{s}_{t,\tau}) + h(\tilde{s}_{t,\tau})F_\pi^{\mathrm{lin}}(s_{t,\tau}; W^\tau)\big)dt + \tilde{\sigma}(\tilde{s}_{t,\tau})dw_t.$$

This implies that the process can be reformulated as a polynomial in $W^\tau - W^0$. We provide detailed background on the Itô–Taylor series with examples in Section C. In Section D, we also present a simplified example of the dynamics of a linear parameterized SDE with a single parameter evolving on a different time scale, using the Itô-Taylor series. This example is intended to give a simplified sense of the broader proof. We then derive the change in the value estimate (Section F) over one and two gradient steps. For the value estimate, we apply the martingale central limit theorem (Haeusler, 1988) (Theorem G.1) together with its corresponding law of large numbers (Theorem H.1) to obtain a conditional central limit theorem (CLT) and the corresponding law of large numbers (LLN). This yields the Gaussian limits described earlier, with an additional error term of order $O(1/\sqrt{n})$. These are similar to the Berry–Esséen theorem (Theorem F.1), accouting for the $O(1/\sqrt{n})$ error. Applying a similar procedure to the action and its derivative results in the Gaussian formulations presented above. Finally, in Section L, we derive the change in the state variable $\Delta s_{t,\tau}$.

Notice that the changes in each of the auxiliary variables, $a_{t,\tau}, a'_{t,\tau}, v_{t,\tau}, v'_{t,\tau}$, depend on $\Delta s_{t,\tau}$ and the TD error like expression $q_{l,\tau}$. This is due to the fact that their distributions are a push-forward of the distribution of the state. In turn, the change in $s_{t,\tau}$ depends only on $q$ and other variables at time $t, \tau$. Intuitively, the variables $(Z_{t,l,\tau}, \mathcal{C}_{u,l,\tau}, Z_{t,\tau})$ capture the infinitesimal change in the environment state dynamics over a single gradient step, and we notice that the expression also contains $a_{t,\tau}, a'_{t,\tau}$ which include the changes the action over environment time. Further notice that

the change in environment's state $\Delta s_{t,\tau}$ is not of order $O(\eta)$, this is because of the divergence in the dynamics, which is a result of the stochasticity in the environment but this stochastic part is mean 0. More formally, the expression that is not $O(1/\sqrt{n})$ in $\Delta s_{t,\tau}$ is $G_{t,\tau} - M_{t,\tau}$ are influenced by the stochasticity in the environment and exploratory dynamics, which are both dependent on the underlying Wiener processes. We illustrate this in Figure 1.

# 7 EMPIRICAL VALIDATION

## 7.1 EXPLORATORY DYNAMICS: OURS VS CANONICAL EXPLORATION

We verify the exploratory nature of our proposed dynamics (Equation 4) and contrast it with dynamics with the additive Wiener process: $\pi(s_t) + w_t$, for a deterministic environment. The environment is defined by $g(s) = 0.1, h(s) = 0.5, \sigma(s) = 0.0, s_0 = 1.0$. We observe that an additive Wiener process does not explore state-action pairs effectively (Figure 3), which is evident by the smoothness of the trajectories whereas under exploratory dynamics (Figure 4) we see stochastic jumps which indicate better coverage of state action pairs.

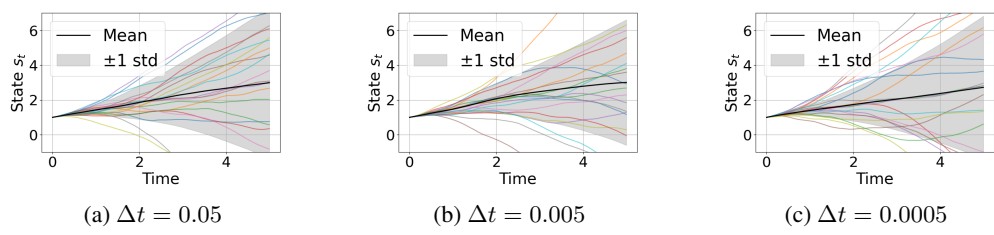

(a) $\Delta t = 0.05$      (b) $\Delta t = 0.005$      (c) $\Delta t = 0.0005$

Figure 3: Simulation results with additive Wiener noise, showing the state trajectory (y-axis) over time (x-axis) for increasingly fine discretizations.

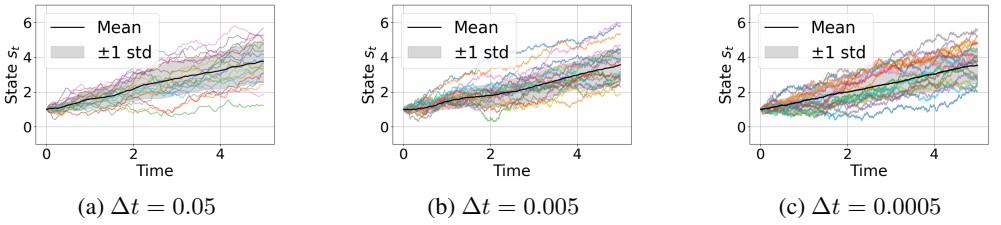

(a) $\Delta t = 0.05$      (b) $\Delta t = 0.005$      (c) $\Delta t = 0.0005$

Figure 4: Simulation results under the exploratory dynamics of Equation 4. As the discretization step decreases ($\Delta t \to 0$), the approximated mean trajectory converges smoothly.

## 7.2 EPISODIC CONTINUOUS TIME ACTOR-CRITIC

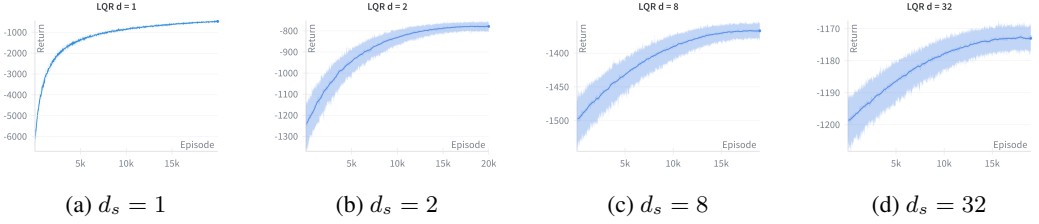

(a) $d_s = 1$      (b) $d_s = 2$      (c) $d_s = 8$      (d) $d_s = 32$

Figure 5: Episodic continuous-time actor–critic with linearized networks. For each dimension $d_s \in \{1, 2, 8, 32\}$ the agent learns a near-optimal policy. Results averaged over 20 seeds.

We analyze algorithm 1 through a theoretical model and also evaluate it empirically using a linearized single–hidden-layer actor and critic. This shows that the method behaves as expected in a toy setting. We test it in an LQR environment with $g(s) = s, h(s) = 1, \sigma(s) = 0.1$, exploratory noise scaled by 0.05, and discount factor $\beta = 0.1$. The initial state is $s_0 = 2.0$, the action range is $[-5, 5]$, and the

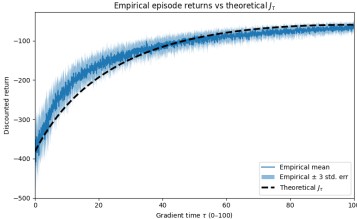

Figure 6: We show that the simulation using our theoretical model (black dotted curve) from theorem 6.1 closely matches the empirical result from an online episodic actor-critic algorithm (Algorithm 1).

reward $r(s) = -500s^2$ drives the agent toward the origin. We use a time step $\Delta t = 0.02$, horizon $T = 1$, and therefore 50 steps per episode. The environment dynamics are governed by Equation 4. We also present results for higher-dimensional environments $d_s = 2, 8, 32$. In these environments the start state is $\mathbb{1}_{d_s}$, the noise matrix is identity $I_{d_s}$, the reward is scaled to avoid overflow issues and actions are similarly truncated. The exploration noise and discounting are the same as in $d_s = 1$.

## 8 RELATED WORK

Our formulation is based on continuous-time RL (Doya, 1995; Jia & Zhou, 2022; 2023), and is complementary to recent advances such as Croissant et al. (2024). It is also inspired by the ODE view of RL learning dynamics in parameter space (Borkar & Meyn, 2000; Borkar et al., 2021), and connects to early analyses of continuous environments for value-based methods (Antos et al., 2007). Related strands in discrete settings have developed neural RL theory and careful empirical science (Gaur et al., 2023; Cai et al., 2019b; Wang et al., 2019; Lyle et al., 2022c;a; Yamamoto et al., 2024). Our contribution is novel in casting actor–critic learning with over-parameterized networks into a nonparametric framework over continuous state, action, and time, and in explicitly characterizing gradient-time dynamics of the state distribution. We have also presented how a continuous time model of RL can lead to theoretical analysis in continuous state and action settings. Our work builds on how neural networks are formulated in a theoretically tractable manner (Jacot et al., 2018; Lee et al., 2019; Roberts et al., 2021; Arora et al., 2019). Moreover, we remark that the limitations from formulating the problem in this manner restrict us to the "lazy regime" where the features do not change and the learning rate is too slow in comparison to more realistic settings (Yang & Hu, 2020; Ghorbani et al., 2019). We believe that our work can be extended to deeper networks in the linearized setting as done by Lee et al. (2019). Our results, with rigor, can be extended to the finite width scenario (Hanin & Nica, 2020) and feature learning (Nichani et al., 2023) in the future.

## 9 DISCUSSION

Our results recast deep RL in continuous control as a two-clock stochastic system: environment time and gradient time, enabling local infinitesimal characterizations of how state, action, and value estimates evolve under actor–critic learning. By combining Itô –Taylor expansions with infinite-width linearizations, we obtain a nonparametric description of policy and value estimate updates and derive, to our knowledge, the first equation for the gradient-time evolution of the state distribution under vanishing step size for neural networks. This provides a principled bridge between stochastic control and modern over-parameterized RL and opens room for simpler theoretical models that can explain the learning dynamics of actor-critic algorithms. Most importantly, we show that there is a simplification of over-parameterized neural networks that emerges from fundamental principles of probability theory and stochastic processes. The analysis currently relies on smooth dynamics, single-hidden-layer models, and asymptotic width; extending to finite-width networks , non-smooth activations, partial observability, and richer continuous control benchmarks is a natural next step. As noted, the results for higher-dimensions are also subject to future research since they would add additional complexity to the proof. Empirical results in a toy LQR environment corroborate both the exploratory dynamics and the validity of algorithm 1. We believe that this non-parametric formulation of neural network-based learning could lead to empirical advancements by extending the understanding of Deep RL in the research community. Further regret analysis and convergence analysis are natural next steps.

## 10 ACKNOWLEDGMENTS

We acknowledge funding from ONR REPRISM MURI N00014-24-1-2603, ONR grant 00014-22-1-2592 and partial funding was also provided by the Robotics and AI Institute. We thank the Brown IRL lab and Oscar CCV for support and compute resources. We thank Professor Paul Dupuis for fruitful conversation and guidance. Saket Tiwari is grateful to his friends and colleagues: Rafael Rodriguez-Sanchez, Sam Lobel, Alessio Mazzetto, Akhil Bagaria, Ji Won Chung, Ashkley Kwon, Kalaiyarasan Arumugam, and Ben Spiegel for their support and conversations.

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

# INDEX

## A  BACKGROUND ON PROBABILITY SPACES AND FILTRATION FOR CONTINUOUS-TIME RANDOM PROCESSES

To establish the background, for continuous-time RL, we introduce ideas from probability theory and stochastic processes, which are used to prove our main result of concentration of states around a low-dimensional manifold. These preliminaries can also be found in chapters 2 and 3 by Øksendal (1987) for the one-dimensional case or in chapter 6 by Gliklikh (2010) for the multidimensional case. A complete probability space is defined by the triple $(\Omega, \mathcal{F}, P)$ where,

1. $\Omega$ is the sample space, for example it could be the set of all values from a die roll with 6 faces $\{1, 2, 3, 4, 5, 6\}$,

2. $\mathcal{F}$ is a set of events, a single event could be a subset of policies from $\Omega$,

3. $P$ is the probability function that maps an event to the probability of it occurring, for example it could be the probability of observing these rolls.

Note that the set $\mathcal{F}$ is complete under union, intersection and complement in addition to containing the null set and is called the $\sigma$-algebra. We will consider random variables on a probability space $(\Omega, \mathcal{F}, P)$ and taking values in the finite-dimensional space $\mathbb{R}^d$. Let $\eta(t)$ be a stochastic process that takes values in $\mathbb{R}^d$ on a probability space $(\Omega, \mathcal{F}, P)$. We will generally describe with stochastic processes which take values in a finite time interval $[0, T]$. For every such stochastic process $\eta$ and any time $t$ this determines three $\sigma$-subalgebras of $\mathcal{F}$:

1. **Past:** the $\sigma$-algebra of pre-images of Borel sets in $\mathbb{R}^d$ for $0 < s < t$,

2. **Present:** the $\sigma$-algebra generated by mappings of $\eta(t)$, and

3. **Future:** the $\sigma$-algebra generated by pre-images of Borel sets in $\mathbb{R}^d$ for $t < s < T$.

Taken together, these define a family of non-decreasing $\sigma$-subalgebras $\mathcal{B}_t$ for $t \in [0, T)$, the reason being our countinuous time MDPs also terminate within a finite time. The *conditional expectation* with respect to a $\sigma$-subalgebra is an orthogonal projection from $L^2(\Omega, \mathcal{F}, P)$ to $L^2(\Omega, \mathcal{F}_0, P)$ over the Hilbert space of square integrable random variables in $L^2(\Omega, \mathcal{F}, P)$. The same projection extends to the $L^1$ space of integrable random variables. Therefore, for every $\eta \in L^1(\Omega, \mathcal{F}, P)$ the orthogonal projection to $L^1(\Omega, \mathcal{F}_0, P)$ is then denoted by $\mathbb{E}[\eta|\mathcal{F}_0]$. A more detailed treatment can be found in the textbook by Karatzas & Shreve (2014). We present the definition a martingale process and a filtration, as stated in Section 3.2 by Øksendal (1987).

**Definition A.1.** *A filtration on $(\Omega, \mathcal{F})$ is a family $\mathcal{B} = \{\mathcal{B}_t\}_{0 \leq t < T}$ of $\sigma$-algebras $\mathcal{B}_t \subset \mathcal{F}$ such that*
$$0 \leq s < t \implies \mathcal{B}_s \subset \mathcal{B}_t,$$
*i.e. $\mathcal{B}_t$ is increasing. A d-dimensional stochastic process $\eta(t), t \in [0, T)$ on $(\Omega, \mathcal{F}, P)$ is called a martingale with respect to a filtration $\mathcal{B}$ and $P$ if*

1. *$\eta(t)$ is $\mathcal{B}_t$ measurable for all $t$*

2. *$\mathbb{E}[|\eta(t)|] < \infty$ for all $t$, and*

3. *$\mathbb{E}[\eta(t)|\mathcal{B}_s] = \eta(s)$ for all $s \leq t$.*

### A.1  WEAK SOLUTION

Consider the time-invariant stochastic differential equation (SDE):
$$dX_t = b(X_t)\, dt + \sigma(X_t)\, dW_t, \quad X_0 = x_0,$$
where $b : \mathbb{R}^d \to \mathbb{R}^d$ is the drift coefficient, $\sigma : \mathbb{R}^d \to \mathbb{R}^{d \times m}$ is the diffusion coefficient, $x_0$ is a given initial probability distribution on $\mathbb{R}^d$.

We say that this SDE admits a weak solution in the sense of probability if there exists a probability space $(\Omega, \mathcal{F}, \mathbb{P})$, a filtration $(\mathcal{F}_t)_{t \in [0,T]}$ satisfying the usual conditions, an $m$-dimensional $(\mathcal{F}_t)$-Brownian motion $W = (W_t)_{t \in [0,T]}$, an $\mathbb{R}^d$-valued, $(\mathcal{F}_t)$-adapted process $X = (X_t)_{t \in [0,T]}$, such that: $X_0 = x_0$ a.s., For all $t \in [0, T]$, the process $X$ satisfies the SDE:
$$X_t = X_0 + \int_0^t b(X_s)\, ds + \int_0^t \sigma(X_s)\, dW_s, \quad \mathbb{P}\text{-a.s.}$$

# B  FORMULATION OF EXPLORATORY DYNAMICS

Control theory and its applications are in continuous-time (Kushner & Dupuis, 2001). Therefore, these applications and theories are related to the study of continuous-time stochastic differential equations (SDEs) (Oksendal, 2013; Karatzas & Shreve, 2014). Despite the fact that all real-world physical processes, such as robotic control, financial processes, or control of chemical processes, are continuous in time, they are simulated in discrete time: due to the inherent time discretized nature of computer clocks. This motivated the study of *numerical schemes* for SDEs i.e., discrete-time processes that converge, in some sense, to continuous-time processes (Kloeden & Platen, 1992; Kushner & Dupuis, 2001). Despite their connections, there is a marked difference between control and RL: RL does not assume agent's awareness of the underlying dynamics of the environment but control does. This also means that one fundamental aspect of RL: exploration is absent in control theory. This means an absence of numerical scheme for SDEs with stochastic exploratory policies. We bridge this gap, in context of control affine systems, and posit an open problem. One trivial way to obtain time-dependent randomization of actions is by adding another Wiener process to the feedback policy which increases the number of dimensions of the numerical simulation. We present a result for a policy with added noise that converges to stochastic dynamics with exploration **without adding an additional dimension to the numerical simulation**.

For fixed $\Delta t > 0$, any deterministic policy $\pi$ consider the following:

$$s_{t_n}^{\Delta t, \pi} = s_0 + \sum_{j=1}^{n-1} \left( g(s_{t_j}^{\Delta t, \pi}) + h(s_{t_j})(\pi(s_{t_j}^{\Delta t, \pi}) + \Delta B_j) \right) \Delta t + \sigma(s_{t_j}^{\Delta t, \pi}) \Delta W_j, \tag{8}$$

$$\text{where } \Delta W_j \sim \mathcal{N}(0, \Delta t), \Delta B_j \sim \mathcal{N}(0, 1/\Delta t), \Delta t = t_j - t_{j-1} \forall j \in \mathbb{N}.$$

We obtain the following result for $d_s = 1, d_a = 1$, which is a common approach in numerical methods control theory results (Kushner & Dupuis, 2001): to derive and present results in one dimension for simplicity and tractability. The higher dimensional results follow. We will use $b(x, \pi(x))$ to denote the coefficient of $dt : g(\tilde{s}_t^\pi) + h\tilde{s}_t^\pi)\pi(\tilde{s}_t^\pi)$ in short.

**Lemma B.1.** *Suppose that $g, h, \sigma$ and $\pi$ are Lipschitz continuous and satisfy linear growth condition:*

$$||g(x)|| \leq K_g(1 + |x|), ||h(x)|| \leq K_h(1 + |x|), ||\pi(x)|| \leq K_\pi(1 + |x|),$$

*then $s_t^{\Delta t, \pi} \to s_t$ weakly where $s_t^\pi$ is solution to the SDE:*

$$ds_t^\pi = (g(s_t^\pi) + h(s_t^\pi)\pi(s_t^\pi))dt + h(s_t^\pi)dw_t' + \sigma(s_t^\pi)dw_t. \tag{9}$$

*Moreover, the solution to this SDE has the same pathwise distribution as the following SDE:*

$$d\tilde{s}_t^\pi = (g(\tilde{s}_t^\pi) + h(\tilde{s}_t^\pi)\pi(\tilde{s}_t^\pi))dt + \sqrt{h(\tilde{s}_t^\pi)^2 + \sigma(\tilde{s}_t^\pi)^2}dw_t. \tag{10}$$

*Proof.* The first part of the Lemma can be proved using the popular and standard Martingale approach detailed in Chapter 11 by Stroock & Varadhan (2006). For some function $f \in C^3(\mathbb{R}^{d_s})$ denote by $f_n = (f(s_{t_n}^{\Delta t, \pi}))$, where we will be omitting the superscript $\Delta t, \pi$ in $f_l$ for brevity as it is implied, similarly we omit $\pi$ in the superscript when its obvious from the context. Consider the following Taylor expansion of this stochastic processes:

$$f_n = f(s_{t_{n-1}}^{\Delta t} + \left( g(s_{t_{n-1}}^{\Delta t}) + h(s_{t_{n-1}})(\pi(s_{t_{n-1}}^{\Delta t}) + \Delta B_j) \right) \Delta t + \sigma(s_{t_{n-1}}^{\Delta t}) \Delta W_j)$$

$$= f(s_{t_{n-1}}^{\Delta t} + \left( g(s_{t_{n-1}}^{\Delta t}) + h(s_{t_{n-1}})(\pi(s_{t_{n-1}}^{\Delta t})) \right) \Delta t + h(s_{t_{n-1}}) \Delta W_{n-1}' + \sigma(s_{t_{n-1}}^{\Delta t}) \Delta W_{n-1}),$$

where $\Delta' W_j = \Delta t \mathcal{N}(0, 1/\Delta t) = \mathcal{N}(0, \Delta t)$, independent of $\Delta W$. Consider the following Taylor expansion:

$$f_n = f(s_{t_{n-1}}^{\Delta t}) + \Delta t f'(s_{t_{n-1}}^{\Delta t}) \left( g(s_{t_{n-1}}^{\Delta t}) + h(s_{t_{n-1}})\pi(s_{t_{n-1}}^{\Delta t}) \right) + \Delta W_{n-1}' f'(s_{t_{n-1}}^{\Delta t}) h(s_{t_{n-1}})\pi(s_{t_{n-1}}^{\Delta t})$$

$$+ \Delta W_{n-1} f'(s_{t_{n-1}}^{\Delta t})\sigma(s_{t_{n-1}}) + \frac{1}{2} f''(s_{t_{n-1}}^{\Delta t}) \left( \left( h(s_{t_{n-1}})\pi(s_{t_{n-1}}^{\Delta t}) \right)^2 + \sigma(s_{t_{n-1}})^2 \right) + O(\Delta t^2)).$$

Now consider the processes:

$$\mu_n - \mu_{n-1} := f(s_{t_n}^{\Delta t}) - f(s_{t_{n-1}}^{\Delta t}) - \mathbb{E}\left[f(s_{t_n}^{\Delta t}) - f(s_{t_{n-1}}^{\Delta t})\right]$$

$$\nu_n - \nu_{n-1} := \mathbb{E}\left[f(s_{t_n}^{\Delta t}) - f(s_{t_{n-1}}^{\Delta t})\right],$$

where the expectation is conditioned over the canonical filtration. For the sake of brevity in this proof, we make the following substitution:

$$a(x) := g(x) + h(x)\,\pi(x).$$

Expanding these terms we obtain:

$$\mu_n = \sum_{j=1}^{n} f'\left(s_{t_{j-1}}^{\Delta t}\right)\left(\sigma\left(s_{t_{j-1}}^{\Delta t}\right)\Delta W_{j-1} + h\left(s_{t_{j-1}}^{\Delta t}\right)\Delta W'_{j-1}\right)$$

$$\nu_n = \sum_{j=1}^{n} \left[f'\left(s_{t_{j-1}}^{\Delta t}\right)a\left(s_{t_{j-1}}^{\Delta t}\right) + \tfrac{1}{2} f''\left(s_{t_{j-1}}^{\Delta t}\right)\left(h^2 + \sigma^2\right)\left(s_{t_{j-1}}^{\Delta t}\right)\right]\Delta t$$

Here $\mu$ is the martingale part and $\nu$ is the remainder. Now we rewrite the indices $n$ for fixed $\Delta t$ such that the integer index is determined by the time. We work on a uniform grid $t_k := k\,\Delta t$ for $k = 0, 1, \ldots, N$ with $\Delta t = T/N$. Define, for $0 \le t < T$,

$$n(t) := \min\{n \in \{1,\ldots,N\} : t < t_n\} = 1 + \lfloor t/\Delta t \rfloor, \qquad \text{and set } n(T) := N.$$

Then $t \in [\,t_{n(t)-1},\, t_{n(t)}\,)$, and the *left gridpoint before* $t$ is

$$\lfloor t \rfloor_{\Delta t} := t_{n(t)-1}.$$

We work on a filtered probability space $(\Omega, \mathcal{F}, (\mathcal{F}_t)_{t\ge0}, \mathbb{P})$ satisfying the usual conditions and carry two independent standard Brownian motions $W = (W_t)_{t\ge0}$ and $W' = (W'_t)_{t\ge0}$ with $W_0 = W'_0 = 0$. On the grid $t_k = k\Delta t$ we set the discrete noises to be the Brownian increments:

$$\Delta W_k := W_{t_{k+1}} - W_{t_k}, \qquad \Delta W'_k := W'_{t_{k+1}} - W'_{t_k},$$

so that $\Delta W_k, \Delta W'_k \overset{\text{i.i.d.}}{\sim} \mathcal{N}(0, \Delta t)$, independent of each other and of $\mathcal{F}_{t_k}$.

Following this notation, we rewrite $\mu$ and $\nu$ as:

$$\widehat{\mu}(t) := \sum_{j=1}^{n(t)-1} f'(s_{t_{j-1}}^{\Delta t})\left[\sigma(s_{t_{j-1}}^{\Delta t})\,\Delta W_{j-1} + h(s_{t_{j-1}}^{\Delta t})\,\Delta W'_{j-1}\right],$$

$$\widehat{\nu}(t) := \sum_{j=1}^{n(t)-1} \left(f'(s_{t_{j-1}}^{\Delta t})\,a(s_{t_{j-1}}^{\Delta t}) + \tfrac{1}{2}f''(s_{t_{j-1}}^{\Delta t})\,(\sigma^2 + h^2)(s_{t_{j-1}}^{\Delta t})\right)\Delta t$$

$$+ \left(f'(s_{t_{n(t)-1}}^{\Delta t})\,a(s_{t_{n(t)-1}}^{\Delta t}) + \tfrac{1}{2}f''(s_{t_{n(t)-1}}^{\Delta t})\,(\sigma^2 + h^2)(s_{t_{n(t)-1}}^{\Delta t})\right)\theta(t)$$

where $n(t)$ is the unique index with $t \in [t_{n(t)-1}, t_{n(t)})$. Then for all $t \in [0,T]$ and $\theta(t) := t - t_{n(t)-1} = t - (n(t)-1)\Delta t \in [0, \Delta t)$.,

$$f(s_t^{\Delta t}) = f(s_0) + \widehat{\mu}(t) + \widehat{\nu}(t) + r^{\Delta t}(t), \qquad \mathbb{E}\big[|r^{\Delta t}(t)|\big] = O(\Delta t^2).$$

We now show increment bounds for $\widehat{\mu}$ and $\widehat{\nu}$. Fix $p \ge 2$. We want to show that there exists $C_{p,T} < \infty$, independent of $\Delta t \le 1$, such that for all $0 \le s < t \le T$,

$$\mathbb{E}\left|\widehat{\mu}(t) - \widehat{\mu}(s)\right|^p \le C_{p,T}\,|t-s|^{p/2}, \tag{11}$$

$$\mathbb{E}\left|\widehat{\nu}(t) - \widehat{\nu}(s)\right|^p \le C_{p,T}\,|t-s|^p. \tag{12}$$

For the martingale part, apply the Burkholder–Davis–Gundy inequality to the increment $\widehat{\mu}(t) - \widehat{\mu}(s)$ to obtain

$$\mathbb{E}\left|\widehat{\mu}(t) - \widehat{\mu}(s)\right|^p \le C_p\,\mathbb{E}\left(\langle\widehat{\mu}\rangle_t - \langle\widehat{\mu}\rangle_s\right)^{p/2},$$

where the quadratic variation over $(s, t]$ is

$$\langle \widehat{\mu} \rangle_t - \langle \widehat{\mu} \rangle_s = \sum_{j:\, (t_{j-1}, t_j] \subset (s,t]} \left( f'(s^{\Delta t}_{t_{j-1}}) \right)^2 (h^2 + \sigma^2)(s^{\Delta t}_{t_{j-1}}) \Delta t \; + \; \text{(partial-step terms)}.$$

Using global linear growth of $h, g, \sigma$ and uniform moment bounds for $s^{\Delta t}$ yields $\mathbb{E}\big[ \langle \widehat{\mu} \rangle_t - \langle \widehat{\mu} \rangle_s \big] \leq C_T |t - s|$, hence equation 11.

For the finite-variation part, by definition

$$\widehat{\nu}(t) - \widehat{\nu}(s) = \int_s^t \left( f'(s^{\Delta t}_{\lfloor u \rfloor_{\Delta t}})\, a(s^{\Delta t}_{\lfloor u \rfloor_{\Delta t}}) + \tfrac{1}{2} f''(s^{\Delta t}_{\lfloor u \rfloor_{\Delta t}}) (h^2 + \sigma^2)(s^{\Delta t}_{\lfloor u \rfloor_{\Delta t}}) \right) du,$$

where $\lfloor u \rfloor_{\Delta t}$ is the left gridpoint before $u$. Using linear growth and the uniform moment bounds gives $\mathbb{E}\, |\, \widehat{\nu}(t) - \widehat{\nu}(s)\, |^p \leq C_{p,T}|t - s|^p$, which is equation 12.

Therefore, For any $p > 2$ there exists $C_{p,T}$ such that

$$\sup_{\Delta t \leq 1} \mathbb{E}\big|\, f(s^{\Delta t}_t) - f(s^{\Delta t}_s)\, \big|^p \; \leq \; C_{p,T}\, |t - s|^{p/2}, \qquad 0 \leq s < t \leq T.$$

Hence, by the Kolmogorov continuity theorem, for every $\alpha \in \left(0, \frac{1}{2} - \frac{1}{p}\right)$ the family $\{f(s^{\Delta t}_\cdot)\}_{\Delta t}$ admits modifications with sample paths that are $\alpha$-Hölder continuous on $[0, T]$, uniformly in $\Delta t$. In particular, any weak limit is supported on $C([0, T])$.

Finally, we have that $s^{\Delta t}_t$ converges weakly to a stochastic process, $s_t$ such that $f(s_t) - \int_0^t L f(s_l) dl$ where $L = (\sigma(x)^2 + h(x)^2)\frac{1}{2}\frac{\partial}{\partial x^2} + a(x)\frac{\partial}{\partial x}$ is Martingale. This implies, as is common in the martingale approach, that such a process $s_t$ is uniquely identified, in the sense of probability, by the solution to Equation 9. Finally, this equation has the same probability on the way to the solution of SDE in Equation 10 because it has the same generator $L$, which implies the same law or the pathwise distribution (see chapter 5, 6 by Stroock & Varadhan (2006)). $\qquad\square$

Therefore, to simulate the dynamics of exploratory agents, we can simulate the solutions to SDE 10. In this setting, we assume that the agent has access to the magnitude of time discretization for exploration, that is, the scalar $\Delta t$.

### B.1  Open Problem: Numerical Scheme for General Continuous-Time Processes with Exploration

While the above numerical scheme holds for control affine SDEs, it need not hold for more general control problems. Specifically, when the function $b(s, a)$ (from Equation 1) has higher-order nonzero derivatives, greater than order 1, in $a$, then we do not have the ability to obtain a stochastic process such as $B \sim \mathcal{N}(0, 1/\Delta t)$ with a simplified multiplicative structure. This implies that the expression $\Delta t B$ is not the only expression that contains $B$ in the Taylor expansion of the proof for Lemma 3.1 instead higher-order terms such as $\Delta B^2$ appear and lead to local changes approaching infinity as $\Delta t \to 0$. Solving this problem is an avenue for future research.

## C  Itô–Taylor Expansion

A common method to analyze smooth functions is using the Taylor series. Taylor series of a function, $f$, centered around a fixed point, say $x$, is an analytical formula which is a summation of powers of the increment, $\delta$, multiplied higher-order derivatives: $f(x + \delta) = \sum_{i=0}^\infty \frac{f^{(i)}(x)}{i!} \delta^i$, where $f^{(i)}$ is the $i$-th derivative. In deep RL, the random variable we are interested in is the state as a function of time parameterized by a two-layer linearized NN. For stochastic dynamics, the *vanilla* Taylor expansion is not sufficient; this is because the $dw_t$ term in dynamics (Equation 4) also contributes to the increments. Therefore, we utilize the Itô–Taylor expansion which accounts for this stochastic increment. Following the textbook by Kloeden & Platen (1992), we first define the multiple stochastic integrals, coefficient functions, and hierarchical sets of indices to define the Itô–Taylor expansion.

## C.1 MULTIPLE STOCHASTIC INTEGRALS

To extend the Taylor expansion to SDEs, we require *multiple stochastic integrals*, which generalize the deterministic integrals in the classical Taylor series. These integrals appear naturally when the Itô calculus is applied to SDEs. We define these using the setting of 1D SDEs. For a multi-index $\alpha = (j_1, j_2, \ldots, j_k)$, where each $j_i \in \{0, 1\}$ with $j_i = 0$ corresponding to a time increment and $j_i = 1$ corresponding to a Wiener process $W_t^{(j_i)}$, the *multiple stochastic integral* is defined as:

$$I_\alpha[f] = \int_0^t \int_0^{l_k} \cdots \int_0^{l_2} f(l_1) \, dZ_{l_1}^{(j_1)} \cdots dZ_{l_k}^{(j_k)},$$

where each $dZ_s^{(j)}$ is either $ds$ if $j = 0$ or $dw_s$ if $j = 1$. These integrals encapsulate both drift and diffusion effects in the stochastic process and are key building blocks of the Itô –Taylor expansion. We provide definitions in the context of a smooth function $f$.

## C.2 COEFFICIENT FUNCTIONS

Each multiple stochastic integral is multiplied by a *coefficient function* derived from the original function $f$ and its partial derivatives with respect to time and state variables. Consider an SDE of the form:

$$dx_t = b(x_t)dt + \sigma(x_t)dw_t,$$

we define the differential operators as follows:

$$L^0 f(x, t) := \frac{\partial f(x, t)}{\partial t} + b(x)\frac{\partial f(x, t)}{\partial x} + \frac{1}{2}\sigma(x)^2\frac{\partial^2 f(x, t)}{\partial x^2},$$

$$L^1 f(x, t) := \sigma(x)\frac{\partial f(x, t)}{\partial x}.$$

The *coefficient function* corresponding to a multi-index $\alpha = (j_1, j_2, \ldots, j_k)$ is defined recursively as:

$$f_\alpha(x) := L^{j_1} L^{j_2} \cdots L^{j_k} f(x).$$

These recursively applied operators capture the evolving influence of both deterministic and stochastic components on the function $f$, and are essential to systematically construct terms in the Itô–Taylor expansion. For a 1D example and letting $f(x) = x$, we have $f_{(0,1)} = b\sigma' + \frac{1}{2}\sigma^2\sigma''$. For example, for the SDE defined by a two-layer linearized NN, which combines the formulation in Section 3 with the setting of Section 5, the differential operators are defined as:

$$L^0 f(x) = \left(g(x) + h(x)\left(f(s; W^0) + \Phi(x, W^0)W\right)\right)\frac{\partial f(x)}{\partial x} + \frac{1}{2}\left(\sigma(x)^2 + h(x)^2\right)\frac{\partial^2 f(x)}{\partial x^2},$$

$$L^1 f(x) = \sqrt{\sigma(x)^2 + h(x)^2}\frac{\partial f(x)}{\partial x}.$$

## C.3 HIERARCHICAL SETS

We also define the successor and predecessor of a multi-index $\alpha$ as follows. Let $\alpha = (j_1, j_2, \ldots, j_k)$. The *predecessor* of $\alpha$ (denoted $\alpha^-$) is obtained by removing the last entry: $\alpha^- := (j_1, j_2, \ldots, j_{k-1})$. Then the $\alpha^+$ multi-index is defined as the multi-index obtained by deleting all the components that are equal to 0. For $\alpha = (1, 1, 0, 1)$, we have $\alpha^+ = (1, 1, 1)$.

These constructions allow us to define coefficient functions and multiple stochastic integrals recursively using a tree-like structure over multi-indices. For example, for $\alpha = (0, 1)$, we have $I_\alpha[f] = \int_0^t I_{\alpha^-}[f]dt = \int_0^t \int_0^l f \, dw_l dt$

To organize and truncate the infinite Itô–Taylor expansion, we define *hierarchical sets* of multi-indices that determine which terms to include based on their *order*. Let $\alpha = (j_1, j_2, \ldots, j_k)$ be a multi-index and define the length of $\alpha$ as:

$$|\alpha| := |\alpha|_0 + |\alpha|_W,$$

where $|\alpha|_0$ denotes the number of zero entries (corresponding to time increments $dt$), and $|\alpha|_w$ is the number of non-zero entries (corresponding to stochastic increments $dw_t$). Let $k_0(\alpha)$ be the number of zeros before the first nonzero component (or the total length if all are zero). For $i = 1, \ldots, |\alpha^+|$, define $k_i(\alpha)$ as the number of components between the $i$th and $(i + 1)$th non-zero components (or up to the end if $i = |\alpha^+|$). For example, if $\alpha = (0, 1, 2, 0)$, then $\alpha^+ = (1, 2)$, $|\alpha^+| = 2$, and

$$k_0(\alpha) = 1, \quad k_1(\alpha) = 0, \quad k_2(\alpha) = 1.$$

For a desired strong or weak approximation order $p$, we define the hierarchical set:

$$\mathcal{A}_p := \{\alpha \,|\, |\alpha| \le p\},$$
$$\text{and } \alpha^- \in \mathcal{A}_p \text{ for each } \alpha \in \backslash\{v\},$$

where $v$ is the multi-index of length zero. The set of all hierarchical sets is $\mathcal{M} = \cup_{p \in \mathbb{N}} \mathcal{A}_p$.

## C.4   TRUNCATED ITÔ–TAYLOR EXPANSION

Equipped with definitions, we can provide the following result of the Itô –Taylor expansion for a smooth $f$.

**Theorem C.1** (Itô–Taylor Expansion (Kloeden & Platen, 1992, Theorem 5.5.1)). *Let $\rho$ and $\tau$ be two stopping times such that*

$$t_0 \le \rho(\omega) \le \tau(\omega) \le T \quad w.p.1.$$

*Let $\mathcal{A} \subset \mathcal{M}$ be a hierarchical set, and let $f : \mathbb{R}^d \to \mathbb{R}$. Then the Itô –Taylor expansion holds:*

$$f(\tau, X_\tau) = \sum_{\alpha \in \mathcal{A}} I_\alpha \left[f_\alpha(X_\rho)\right]_{\rho,\tau} + \sum_{\alpha \in \mathcal{B}(\mathcal{A})} I_\alpha \left[f_\alpha(X.)\right]_{\rho,\tau},$$

*provided that there exist all derivatives of all orders of $f$, drift $a$, diffusion $b$, and the required multiple Itô integrals exist.*

Therefore, for $\kappa = 0, 1, \ldots$ and $f(x) = x$ the truncated Itô –Taylor expansion is:

$$X_t^\kappa = \sum_{\alpha \in \Lambda_\kappa} I_\alpha[f_\alpha(X_0)]_{0,t}, \tag{13}$$

where $\Lambda_\kappa = \{\alpha \in \mathcal{M} : |\alpha| \le \kappa\}$ is the hierarchical set of all multi-indices of size less than or equal to $\kappa$. We refer to the other terms, not included in the expansion, as the remainder terms. We denote by $\mathcal{B}_\kappa = \{\alpha \in \mathcal{M} : |\alpha| \le \kappa, |\alpha|_0 = |\alpha|\}$ the set of all multi-indices of size $\kappa$ with all elements set to 0 and therefore correspond to the deterministic part of the series We use the notation $\Omega_k = \Lambda_\kappa - \mathcal{B}_\kappa$ to denote the non-deterministic part of the Itô -Taylor expansion. We denote by $\Xi_k$ the set of all multi-indices of size $k$, i.e. $\Xi_k = \{\alpha \in \mathcal{M} : |\alpha| = \kappa\}$

### C.5 EXAMPLE ITÔ –TAYLOR EXPANSION

As an example, letting $I_\alpha = [1]_\alpha$, and omitting the arguments of the functions (which is $X_0$ in this case) the Itô –Taylor expansion for $\kappa = 3$ is:

$$X_t = X_0 + b\,I_{(0)} + \sigma\,I_{(1)} + \left(bb' + \frac{1}{2}\sigma^2 b''\right) I_{(0,0)}$$

$$+ \left(b\sigma' + \frac{1}{2}\sigma^2\sigma''\right) I_{(0,1)} + b\sigma' I_{(1,0)} + \sigma\sigma' I_{(1,1)}$$

$$+ \left[b\left(bb' + (b')^2 + \sigma\sigma'' + \frac{1}{2}\sigma^2 b''\right) + \frac{1}{2}\sigma^2\left(bb'' + 3b'b''\right)\right.$$

$$\left. + \left((\sigma')^2 + \sigma\sigma''\right) b'' + 2\sigma\sigma' b''' + \frac{1}{4}\sigma^4 b^{(4)}\right] I_{(0,0,0)}$$

$$+ \left[b\left(\sigma'b' + b\sigma'' + \sigma\sigma'\sigma'' + \frac{1}{2}\sigma^2\sigma'''\right) + \frac{1}{2}\sigma^2\left(b''\sigma' + 2b'\sigma''\right.\right.$$

$$\left.\left. + b\sigma''' + \left((\sigma')^2 + \sigma\sigma''\right)\sigma'' + 2\sigma\sigma'\sigma''' + \frac{1}{2}\sigma^2\sigma^{(4)}\right)\right] I_{(0,0,1)}$$

$$+ \left[b(\sigma'b' + \sigma b'') + \frac{1}{2}\sigma^2\left(\sigma''b' + 2\sigma'b'' + b\sigma'''\right)\right] I_{(0,1,0)}$$

$$+ \left[b\left((\sigma')^2 + \sigma\sigma''\right) + \frac{1}{2}\sigma^2\left(\sigma''\sigma' + 2\sigma\sigma'' + \sigma\sigma'''\right)\right] I_{(0,1,1)}$$

$$+ \sigma\left(bb'' + (b')^2 + \sigma\sigma'b'' + \frac{1}{2}\sigma^2 b'''\right) I_{(1,0,0)}$$

$$+ \sigma\left(b\sigma'' + \sigma'b' + \sigma\sigma'\sigma'' + \frac{1}{2}\sigma^2\sigma'''\right) I_{(1,0,1)}$$

$$+ \sigma(\sigma'b' + b''\sigma) I_{(1,1,0)} + \sigma\left((\sigma')^2 + \sigma\sigma''\right) I_{(1,1,1)} + R,$$

where $R$ denotes the remainder.

## D  LINEAR TWO-TIMESCALE SYSTEM

Suppose that you have the following SDE controlled by a parameter $\theta(\tau)$:

$$dX_t^\tau = -\theta(\tau)X_t^\tau\,dt + \sigma\,dw_t \quad \text{and} \quad \frac{d\theta(\tau)}{d\tau} = f(\theta).$$

Consider the following update: $\theta(\tau + \eta) = \theta(\tau) + \eta f(\theta_\tau)$. Consider the Itô -Taylor expansion for the $X_t(\theta)$ (following the expansion in Section C.5):

$$X_t(\theta) = X_0 - \theta X_0 I_{(0)} + \sigma I_{(1)} + \theta^2 X_0 I_{(0,0)} + 0 \cdot I_{(0,1)} + 0 \cdot I_{(1,0)} + 0 \cdot I_{(1,1)}$$
$$+ \left[-\theta X_0\left(\theta^2 X_0 + \theta^2\right)\right] I_{(0,0,0)} + 0 \cdot I_{(0,0,1)}$$
$$+ 0 \cdot I_{(0,1,0)} + 0 \cdot I_{(0,1,1)} + \sigma\theta^2 I_{(1,0,0)} + 0 \cdot I_{(1,0,1)} + 0 \cdot_{(1,1,0)} + 0 \cdot I_{(1,1,1)} + R$$
$$= X_0 - \theta X_0 I_{(0)} + \sigma I_{(1)} + \theta^2 X_0 I_{(0,0)} - \left[\theta X_0\left(\theta^2 X_0 + \theta^2\right)\right] I_{(0,0,0)} + \sigma\theta^2 I_{(1,0,0)} + R,$$

of which the leading expression, minus $R$, is denoted by $X_t^3(\theta)$. Further, consider the expression $X_t^3(\theta + \eta\theta) - X_t^3(\theta)$:

$$X_t^3(\theta + \eta f(\theta)) - X_t^3(\theta) = -\eta f(\theta)X_0 I_{(0)} + \left((\theta + \eta f(\theta))^2 - \theta^2\right) X_0 I_{(0,0)}$$
$$- X_0\left[(\theta + \eta f(\theta))\left((\theta + \eta f(\theta))^2 X_0 + (\theta + \eta f(\theta))^2\right) - \theta\left(\theta^2 X_0 + \theta^2\right)\right] I_{(0,0,0)}$$
$$+ \sigma\left[(\theta + \eta f(\theta))^2 - \theta^2\right] I_{(1,0,0)}$$
$$= -\eta f(\theta)X_0 I_{(0)} + 2\theta\eta f(\theta)X_0 I_{(0,0)} - 3\theta^2\eta f(\theta)(X_0 + 1)I_{(0,0,0)} + 2\sigma\eta f(\theta)\theta I_{(1,0,0)} + O(\eta^2)$$

### D.1 ANALYTIC SOLUTION TO TWO-TIMESCALE SDEs

Suppose that you have the SDE and the ODE as described above:

$$dX_t^\tau = -\theta(\tau)X_t^\tau \, dt + \sigma \, dw_t \quad \text{and} \quad \frac{d\theta(\tau)}{d\tau} = f(\theta(\tau)).$$

Let $Y_t^\tau := \frac{\partial X_t^\tau}{\partial \tau}$ and, therefore, heuristically:

$$\frac{\partial dX_t^\tau}{\partial \tau} = - X_t^\tau f(\theta(\tau))dt - Y_t^\tau \theta(\tau)dt.$$

Therefore, solving the following ODE with the random variable $X^\tau$ gives the solution to $Y_t^\tau$:

$$dY_t^\tau = -X_t^\tau f(\theta(\tau))dt - Y_t^\tau \theta(\tau)dt, \quad \text{with } Y(0,\tau) = 0. \tag{14}$$

The two-dimensional SDE can be written as:

$$d\begin{bmatrix} X_t^\tau \\ Y_t^\tau \end{bmatrix} = \begin{bmatrix} -\theta(\tau) & 0 \\ -f(\theta(\tau)) & -\theta(\tau) \end{bmatrix} \begin{bmatrix} X_t^\tau \\ Y_t^\tau \end{bmatrix} dt + \begin{bmatrix} \sigma \\ 0 \end{bmatrix} dw_t.$$

We simplify the notation by omitting $\tau$ since it is fixed and also denote $Z_t = \begin{bmatrix} X_t \\ Y_t \end{bmatrix}$:

$$d\begin{bmatrix} X_t \\ Y_t \end{bmatrix} = \begin{bmatrix} -\theta & 0 \\ -f(\theta) & -\theta \end{bmatrix} \begin{bmatrix} X_t \\ Y_t \end{bmatrix} dt + \begin{bmatrix} \sigma \\ 0 \end{bmatrix} dw_t$$
$$dZ_t = F(\theta)Z_t dt + \sigma' dw_t.$$

Consider $L^0, L^1$ operators for this two-dimensional SDE are as follows:

$$L^0 = -\theta X \frac{\partial}{\partial X} - (f(\theta)X + \theta Y)\frac{\partial}{\partial Y} + \frac{1}{2}\sigma^2 \frac{\partial^2}{\partial X^2}$$

$$L^1 = \sigma \frac{\partial}{\partial X}.$$

Here we ignore the $\partial/\partial t$ term because the dynamics are time invariant. Expanding the SDE using the Itô -Taylor expansion for $Z$ letting $f(Z) = Z$, we obtain the following:

$$
\begin{aligned}
Z_t =& Z_0 + L^0 f I_{(0)} + L^1 f I_{(1)} + L^0 L^0 f I_{(0,0)} + L^0 L^1 f I_{(0,1)} + L^1 L^0 f I_{(1,0)} + R \\
=& Z_0 + \begin{bmatrix} -\theta X_0 \\ -f(\theta)X_0 - \theta Y_0 \end{bmatrix} I_{(0)} + \begin{bmatrix} \sigma \\ 0 \end{bmatrix} I_{(1)} + \begin{bmatrix} \theta^2 X_0 \\ \theta f(\theta)X_0 + \theta^2 Y_0 \end{bmatrix} I_{(0,0)} + \begin{bmatrix} -\sigma\theta \\ -\sigma f(\theta) \end{bmatrix} I_{(1,0)} \\
& + \begin{bmatrix} 0 \\ 0 \end{bmatrix} I_{(0,1)} + \begin{bmatrix} 0 \\ 0 \end{bmatrix} I_{(1,1)} + R \\
=& \begin{bmatrix} X_0 \\ Y_0 \end{bmatrix} + \begin{bmatrix} -\theta X_0 \\ -f(\theta)X_0 - \theta Y_0 \end{bmatrix} I_{(0)} + \begin{bmatrix} \sigma \\ 0 \end{bmatrix} I_{(1)} + \begin{bmatrix} \theta^2 X_0 \\ 2\theta f(\theta)X_0 + \theta^2 Y_0 \end{bmatrix} I_{(0,0)} + \begin{bmatrix} -\sigma\theta \\ -\sigma f(\theta) \end{bmatrix} I_{(1,0)} + R \\
=& \begin{bmatrix} X_0 \\ 0 \end{bmatrix} + \begin{bmatrix} -\theta X_0 \\ -f(\theta)X_0 \end{bmatrix} I_{(0)} + \begin{bmatrix} \sigma \\ 0 \end{bmatrix} I_{(1)} + \begin{bmatrix} \theta^2 X_0 \\ 2\theta f(\theta)X_0 \end{bmatrix} I_{(0,0)} + \begin{bmatrix} -\sigma\theta \\ -\sigma f(\theta) \end{bmatrix} I_{(1,0)} + R
\end{aligned}
$$

Consider the expansion for the third term here:

$$
\begin{aligned}
\bar{Z}_t^3 =& L^0 L^0 L^0 f I_{(0,0,0)} + L^0 L^0 L^1 f I_{(0,0,1)} + L^0 L^1 L^0 f I_{(0,1,0)} + L^1 L^0 L^0 f I_{(1,0,0)} \\
& + L^0 L^1 L^1 f I_{(0,1,1)} + L^1 L^0 L^1 f I_{(1,0,1)} + L^1 L^1 L^0 f I_{(1,1,0)} + L^1 L^1 L^0 f I_{(1,1,1)} \\
=& \begin{bmatrix} -\theta^3 X_0 \\ -2\theta^2 f(\theta)X_0 - \theta^2 f(\theta)X_0 - \theta^3 Y_0 \end{bmatrix} I_{(0,0,0)} + \begin{bmatrix} 0 \\ 0 \end{bmatrix} I_{(0,0,1)} + \begin{bmatrix} 0 \\ 0 \end{bmatrix} I_{(0,1,0)} + \begin{bmatrix} \sigma\theta^2 \\ \sigma\theta f(\theta) \end{bmatrix} I_{(1,0,0)} \\
& + \begin{bmatrix} 0 \\ 0 \end{bmatrix} I_{(0,1,1)} + \begin{bmatrix} 0 \\ 0 \end{bmatrix} I_{(1,0,1)} + \begin{bmatrix} 0 \\ 0 \end{bmatrix} I_{(1,1,0)} + \begin{bmatrix} 0 \\ 0 \end{bmatrix} I_{(1,1,1)} \\
=& \begin{bmatrix} -\theta^3 X_0 \\ -3\theta^2 f(\theta)X_0 \end{bmatrix} I_{(0,0,0)} + \begin{bmatrix} \sigma\theta^2 \\ 2\sigma\theta f(\theta) \end{bmatrix} I_{(1,0,0)}
\end{aligned}
$$

Combining the two expressions, we obtain the following:

$$Z_t = \begin{bmatrix} X_0 \\ 0 \end{bmatrix} + \begin{bmatrix} -\theta X_0 \\ -f(\theta)X_0 \end{bmatrix} I_{(0)} + \begin{bmatrix} \sigma \\ 0 \end{bmatrix} I_{(1)} + \begin{bmatrix} \theta^2 X_0 \\ 2\theta f(\theta)X_0 \end{bmatrix} I_{(0,0)} + \begin{bmatrix} -\sigma\theta \\ -\sigma f(\theta) \end{bmatrix} I_{(1,0)}$$
$$+ \begin{bmatrix} -\theta^3 X_0 \\ -3\theta^2 f(\theta)X_0 \end{bmatrix} I_{(0,0,0)} + \begin{bmatrix} \sigma\theta^2 \\ 2\sigma\theta f(\theta) \end{bmatrix} I_{(1,0,0)} + R.$$

The expressions for $Y$ closely mirror the expression for $X$ given that $\theta$ changes according to $d\theta(\tau)/d\tau = f(\theta)$. This insight can be used to reason about the changes, in gradient step, for the more general control affine problem with linearized neural network where the Itô -Taylor expansion is polynomial in $\Delta W^\tau$.

## E CRITIC FORMULATION

To evaluate the changes in the random variables of interest we present the following update rules in the actor-critic framework. The value estimate parameterized by a linearized neural network can be denoted by:

$$F_v(s,t;U,B) = \frac{1}{\sqrt{n}} \sum_{\kappa=1}^{n} B_\kappa \varphi(U_\kappa \cdot [s,t]),$$
$$v^{\text{lin}}(s,t;U) = F_v(s,t;U^0) + \Psi(s,t;U^0)(U - U^0),$$

similar to Equation 7. Here $U = [U_1, \ldots, U_n] \in \mathbb{R}^{n(d_s+1)}$ and $B = [B_1, \ldots, B_n] \in \mathbb{R}^{1 \times n}$ and $\varphi$ is the smooth activation function, $\tanh$. We denote by $[s,t]$ the concatenation of the state and time variables. The parameters are initialized in the same way as described in Section 5. $B^0, U^0$ remain fixed after initialization. The matrix $\Psi$ is defined as:

$$\Psi(s,t;U^0) = \frac{1}{\sqrt{n}} \left[ B_1^0 \varphi'(U_1^0 \cdot [s,t])[s,t]^\top, \ldots, B_n^0 \varphi'(U_n^0 \cdot [s,t])[s,t]^\top \right] \in \mathbb{R}^{1 \times n(d_s+1)}.$$

Based on this formulation of the linearized NN we can define the gradient updates for the value estimate and the policy as follows:

$$\widehat{\mathbb{G}}(U,W) = \int_0^T e^{-\beta l} \Psi(\tilde{s}_l^{\pi^W}, l; U^0)^\top \left[ \partial_t v^{\text{lin}}(l, \tilde{s}_l^{\pi^W}; U) + r(\tilde{s}_l^{\pi^W}) - \beta v^{\text{lin}}(l, \tilde{s}_l^{\pi^W}; U) \right], \quad (15)$$

$$\widehat{\mathcal{G}}(U,W) = \int_0^T e^{-\beta l} \Phi(\tilde{s}_l^{\pi^W}; W^0)^\top \left[ \partial_t v^{\text{lin}}(l, \tilde{s}_l^{\pi^W}; U) + r(\tilde{s}_l^{\pi^W}) - \beta v^{\text{lin}}(l, \tilde{s}_l^{\pi^W}; U) \right] dl. \quad (16)$$

In this setting there are two time-dependent random variables whose changes we track over gradient steps: $v_\tau^{\text{lin}}(t, \tilde{s}_{l,\tau}), s_{t,\tau}$. We will use the shorthand $v_{t,\tau}$ to denote the first term. We also omit the superscript $n$ for $s$, denoting the state under the gradient updates described above for a linearized NN of width $n$.

# F   EXPRESSION FOR CHANGE IN VALUE ESTIMATE OVER GRADIENT STEP

To better analyze and understand the change in the value estimate, we evaluate the changes in $v_{t,\tau}$ over gradient steps. For the learning rate $\eta$, consider the difference:

$$
\begin{aligned}
v_{t,\tau+\eta} - v_{t,\tau} =& F(s_{t,\tau+\eta}, t; U^0) + \Psi(s_{t,\tau+\eta}, t; U^0)(U^{\tau+\eta} - U^0) \\
& - \left( F(\tilde{s}_{t,\tau}, t; U^0) + \Psi(s_{t,\tau}, t; U^0)(U^\tau - U^0) \right) \\
=& \frac{1}{\sqrt{n}} \sum_{\kappa=1}^{n} B_\kappa^0 \left( \varphi(U_\kappa^0 \cdot [s_{t,\tau+\eta}, t]) - \varphi(U_\kappa^0 \cdot [s_{\tau,t}, t]) \right) \\
& + \left( \Psi(s_{t,\tau+\eta}, t; U^0) U^{\tau+\eta} - \Psi(\tilde{s}_{t,\tau}, t; U^0) U^\tau \right) - \left( \Psi(s_{t,\tau+\eta}, t; U^0) - \Psi(\tilde{s}_{t,\tau}, t; U^0) \right) U^0 \\
=& \frac{1}{\sqrt{n}} \sum_{\kappa=1}^{n} B_\kappa^0 \Bigg( \varphi'(U_\kappa^0 \cdot [\tilde{s}_{t,\tau}, t])[s_{t,\tau+\eta} - \tilde{s}_{t,\tau}, 0]^\top U_\kappa^0 \\
& + \frac{1}{2} \varphi''(U_\kappa^0 \cdot [\tilde{s}_{t,\tau}, 0]) \left( [s_{t,\tau+\eta} - \tilde{s}_{t,\tau}, 0]^\top U_{\kappa,1}^0 \right)^2 \Bigg) + O(\eta^2) \\
& + \frac{1}{\sqrt{n}} \sum_{\kappa=1}^{n} B_\kappa^0 \left( \varphi'(U_\kappa^0 \cdot [s_{t,\tau+\eta}, t])[s_{t,\tau+\eta}, t]^\top U_\kappa^{\tau+\eta} - \varphi'(U_\kappa^0 \cdot [s_{\tau,t}, t])[\tilde{s}_{t,\tau}, t]^\top U_\kappa^\tau \right) \\
& - \frac{1}{\sqrt{n}} \sum_{\kappa=1}^{n} B_\kappa^0 \left( \varphi'(U_\kappa^0 \cdot [s_{t,\tau+\eta}, t])[s_{t,\tau+\eta}, t]^\top - \varphi'(U_\kappa^0 \cdot [s_{\tau,t}, t])[\tilde{s}_{t,\tau}, t]^\top \right) U_\kappa^0
\end{aligned}
\tag{17}
$$

We use $\Delta \tilde{s}_{t,\tau} = s_{t,\tau+\eta} - \tilde{s}_{t,\tau}$ as the shorthand. We further analyze the two expressions above.

$$
\begin{aligned}
& \frac{1}{\sqrt{n}} \sum_{\kappa=1}^{n} B_\kappa^0 \left( \varphi'(U_\kappa^0 \cdot [s_{t,\tau+\eta}, t])[s_{t,\tau+\eta}, t]^\top U_\kappa^{\tau+\eta} - \varphi'(U_\kappa^0 \cdot [s_{\tau,t}, t])[\tilde{s}_{t,\tau}, t]^\top U_\kappa^\tau \right) \\
=& \frac{1}{\sqrt{n}} \sum_{\kappa=1}^{n} B_\kappa^0 \Bigg( \varphi'(U_\kappa^0 \cdot [s_{t,\tau+\eta}, t])[\Delta \tilde{s}_{t,\tau}, t]^\top U_\kappa^\tau \\
& + \eta \varphi'(U_\kappa^0 \cdot [s_{t,\tau+\eta}, t])[\Delta \tilde{s}_{t,\tau}, t]^\top \widehat{\mathbb{G}}(U^\tau, W^\tau)_\kappa + \\
& + \varphi'(U_\kappa^0 \cdot [s_{t,\tau+\eta}, t])[\tilde{s}_{t,\tau}, t]^\top U_\kappa^\tau + \eta \varphi'(U_\kappa^0 \cdot [s_{t,\tau+\eta}, t])[\tilde{s}_{t,\tau}, t]^\top \widehat{\mathbb{G}}(U^\tau, W^\tau)_\kappa \\
& - \varphi'(U_\kappa^0 \cdot [s_{\tau,t}, t])[\tilde{s}_{t,\tau}, t]^\top U_\kappa^\tau \Bigg).
\end{aligned}
$$

Taylor expanding on these expressions individually we obtain:

$$
\begin{aligned}
=&\frac{1}{\sqrt{n}}\sum_{\kappa=1}^{n} B_\kappa^0 \Bigg( \Big( \big(\varphi'(U_\kappa^0\cdot[\tilde{s}_{t,\tau},t])+\varphi''(U_\kappa^0\cdot[\tilde{s}_{t,\tau},t])[\Delta\tilde{s}_{t,\tau},0]^\top U_\kappa^0 \\
&+\frac{1}{2}\varphi'''(U_\kappa^0\cdot[\tilde{s}_{t,\tau},t])\big([\Delta\tilde{s}_{t,\tau},0]^\top U_\kappa^0\big)^2+O(\eta^2)\big)[\tilde{s}_{t,\tau},t]^\top U_\kappa^\tau \\
&+\big(\eta\varphi'(U_\kappa^0\cdot[\tilde{s}_{t,\tau},t])[\Delta\tilde{s}_{t,\tau},t]^\top\widehat{\mathbb{G}}(U^\tau,W^\tau)_\kappa \\
&+\eta\varphi''(U_\kappa^0\cdot[\tilde{s}_{t,\tau},t])\big(U_\kappa^0\cdot[\Delta\tilde{s}_{t,\tau},0]\big)[\Delta\tilde{s}_{t,\tau},t]^\top\widehat{\mathbb{G}}(U^\tau,W^\tau)_\kappa+O(\eta^2)\big) \\
&+\big(\eta\varphi'(U_\kappa^0\cdot[\tilde{s}_{t,\tau},t])[\tilde{s}_{t,\tau},t]^\top\widehat{\mathbb{G}}(U^\tau,W^\tau)_\kappa \\
&+\eta\varphi'(U_\kappa^0\cdot[\tilde{s}_{t,\tau},t])\big([\Delta\tilde{s}_{t,\tau},0]^\top U_\kappa^0\big)[\tilde{s}_{t,\tau},t]^\top\widehat{\mathbb{G}}(U^\tau,W^\tau)_\kappa \\
&+\frac{\eta}{2}\varphi''(U_\kappa^0\cdot[\tilde{s}_{t,\tau},t])\big([\Delta\tilde{s}_{t,\tau},0]^\top U_\kappa^0\big)^2\mathbb{G}(U^\tau,W^\tau)_\kappa+O(\eta^2)\big)-\varphi'(U_\kappa^0\cdot[s_{\tau,t},t])[\tilde{s}_{t,\tau},t]^\top U_\kappa^\tau \Bigg) \\[6pt]
=&\frac{1}{\sqrt{n}}\sum_{\kappa=1}^{n} B_\kappa^0 \Bigg( \Big( \big(\varphi''(U_\kappa^0\cdot[\tilde{s}_{t,\tau},t])[\Delta\tilde{s}_{t,\tau},0]^\top U_\kappa^0 \\
&+\frac{1}{2}\varphi'''(U_\kappa^0\cdot[\tilde{s}_{t,\tau},t])\big([\Delta\tilde{s}_{t,\tau},0]^\top U_\kappa^0\big)^2+O(\eta^2)\big)[\tilde{s}_{t,\tau},t]^\top U_\kappa^\tau \\
&+\big(\eta\varphi'(U_\kappa^0\cdot[\tilde{s}_{t,\tau},t])[\Delta\tilde{s}_{t,\tau},t]^\top\widehat{\mathbb{G}}(U^\tau,W^\tau)_\kappa \\
&+\eta\varphi''(U_\kappa^0\cdot[\tilde{s}_{t,\tau},t])\big(U_\kappa^0\cdot[\Delta\tilde{s}_{t,\tau},0]\big)[\Delta\tilde{s}_{t,\tau},t]^\top\widehat{\mathbb{G}}(U^\tau,W^\tau)_\kappa+O(\eta^2)\big) \\
&+\big(\eta\varphi'(U_\kappa^0\cdot[\tilde{s}_{t,\tau},t])[\tilde{s}_{t,\tau},t]^\top\widehat{\mathbb{G}}(U^\tau,W^\tau)_\kappa \\
&+\eta\varphi'(U_\kappa^0\cdot[\tilde{s}_{t,\tau},t])\big([\Delta\tilde{s}_{t,\tau},0]^\top U_\kappa^0\big)[\tilde{s}_{t,\tau},t]^\top\widehat{\mathbb{G}}(U^\tau,W^\tau)_\kappa \\
&+\frac{\eta}{2}\varphi''(U_\kappa^0\cdot[\tilde{s}_{t,\tau},t])\big([\Delta\tilde{s}_{t,\tau},0]^\top U_\kappa^0\big)^2\widehat{\mathbb{G}}(U^\tau,W^\tau)_\kappa+O(\eta^2)\big)\Big)\Bigg) \\[6pt]
=&\frac{1}{\sqrt{n}}\sum_{\kappa=1}^{n} B_\kappa^0 \Bigg( \Big( \big(\varphi''(U_\kappa^0\cdot[\tilde{s}_{t,\tau},t])[\Delta\tilde{s}_{t,\tau},0]^\top U_\kappa^0 \\
&+\frac{1}{2}\varphi'''(U_\kappa^0\cdot[\tilde{s}_{t,\tau},t])\big([\Delta\tilde{s}_{t,\tau},0]^\top U_\kappa^0\big)^2+O(\eta^2)\big)[\tilde{s}_{t,\tau},t]^\top U_\kappa^\tau \\
&+\big(\eta\varphi'(U_\kappa^0\cdot[\tilde{s}_{t,\tau},t])[\Delta\tilde{s}_{t,\tau},t]^\top\widehat{\mathbb{G}}(U^\tau,W^\tau)_\kappa+O(\eta^2)\big) \\
&+\big(\eta\varphi'(U_\kappa^0\cdot[\tilde{s}_{t,\tau},t])[\tilde{s}_{t,\tau},t]^\top\widehat{\mathbb{G}}(U^\tau,W^\tau)_\kappa \\
&+\eta\varphi'(U_\kappa^0\cdot[\tilde{s}_{t,\tau},t])\big([\Delta\tilde{s}_{t,\tau},0]^\top U_\kappa^0\big)[\tilde{s}_{t,\tau},t]^\top\widehat{\mathbb{G}}(U^\tau,W^\tau)_\kappa+O(\eta^2)\big)\Bigg).
\end{aligned}
$$

$$\tag{18}$$

Now, consider the second term in the summation above (Equation 17) for $v_{t,\tau+\eta} - v_{t,\tau}$:

$$\frac{1}{\sqrt{n}} \sum_{\kappa=1}^{n} B_\kappa^0 \left( \varphi'(U_\kappa^0 \cdot [s_{t,\tau+\eta}, t])[s_{t,\tau+\eta}, t]^\top - \varphi'(U_\kappa^0 \cdot [s_{\tau,t}, t])[\tilde{s}_{t,\tau}, t]^\top \right) U_\kappa^0$$

$$= \frac{1}{\sqrt{n}} \sum_{\kappa=1}^{n} B_\kappa^0 \left( \varphi'(U_\kappa^0 \cdot [\tilde{s}_{t,\tau}, t])[s_{t,\tau+\eta}, t]^\top + \varphi''(U_\kappa^0 \cdot [\tilde{s}_{t,\tau}, t])[\Delta\tilde{s}_{t,\tau}, 0]^\top U_\kappa^0 [s_{t,\tau+\eta}, t]^\top \right.$$

$$\left. + O(\eta^2) - \varphi'(U_\kappa^0 \cdot [s_{\tau,t}, t])[\tilde{s}_{t,\tau}, t]^\top \right) U_\kappa^0$$

$$= \frac{1}{\sqrt{n}} \sum_{\kappa=1}^{n} B_\kappa^0 \left( \varphi'(U_\kappa^0 \cdot [\tilde{s}_{t,\tau}, t])[\tilde{s}_{t,\tau}, t]^\top + \varphi'(U_\kappa^0 \cdot [\tilde{s}_{t,\tau}, t])[\Delta\tilde{s}_{t,\tau}, 0]^\top \right.$$

$$\left. + \varphi'(U_\kappa^0 \cdot [\tilde{s}_{t,\tau}, t])[\Delta\tilde{s}_{t,\tau}, 0]^\top + O(\eta^2) - \varphi'(U_\kappa^0 \cdot [s_{\tau,t}, t])[\tilde{s}_{t,\tau}, t]^\top \right) U_\kappa^0$$

$$= \frac{1}{\sqrt{n}} \sum_{\kappa=1}^{n} B_\kappa^0 \left( \varphi'(U_\kappa^0 \cdot [\tilde{s}_{t,\tau}, t])[\Delta\tilde{s}_{t,\tau}, 0]^\top + \varphi''(U_\kappa^0 \cdot [\tilde{s}_{t,\tau}, t])\Delta\tilde{s}_{t,\tau} U_{\kappa,1}^0 [\tilde{s}_{t,\tau}, t]^\top \right.$$

$$\left. + O(\eta^2) \right) U_\kappa^0.$$

We further simplify the expression for the second last in the expansion of equation 18. To do so we denote by $q_{l,\tau} = \partial_t v^{\text{lin}}(l, s_{l,\tau}; U) + r(s_{l,\tau}) - \beta v^{\text{lin}}(l, s_{l,\tau}; U)$.

$$\frac{1}{\sqrt{n}} \sum_{\kappa=1}^{n} \eta B_\kappa^0 \left( \varphi'(U_\kappa^0 \cdot [\tilde{s}_{t,\tau}, t])[\tilde{s}_{t,\tau}, t]^\top \widehat{\mathbb{G}}(U^\tau, W^\tau)_\kappa \right)$$

$$= \frac{1}{\sqrt{n}} \sum_{\kappa=1}^{n} \eta B_\kappa^0 \left( \varphi'(U_\kappa^0 \cdot [\tilde{s}_{t,\tau}, t]) \left( \int_0^T e^{-\beta l} \frac{1}{\sqrt{n}} B_\kappa^0 \varphi'(U_\kappa^0 \cdot [\tilde{s}_{l,\tau}, l])(\tilde{s}_{l,\tau}\tilde{s}_{t,\tau} + lt)q(s, l; U)dl \right) \right)$$

$$= \eta \int_0^T e^{-\beta l} \frac{1}{n} \sum_{\kappa=1}^{n} (B_\kappa^0)^2 \varphi'(U_\kappa^0 \cdot [\tilde{s}_{t,\tau}, t]) \varphi'(U_\kappa^0 \cdot [\tilde{s}_{l,\tau}, l])(\tilde{s}_{l,\tau}\tilde{s}_{t,\tau} + lt)q_{l,\tau}dl$$

### F.1 SUMMARY STATISTICS AND POLYNOMIAL EXPRESSION OF STATE VARIABLE

Consider the random variable defined by:

$$Y_{t,\tau} = \frac{1}{\sqrt{n}} \sum_{\kappa=1}^{n} B_\kappa^0 U_{\kappa,1}^0 \varphi'(U_\kappa^0 \cdot [\tilde{s}_{t,\tau}, t]). \tag{19}$$

We prove that as $n \to \infty$ and for $\tau = O(\sqrt{n})$ it converges to a Gaussian variable conditioned on $\tilde{s}_{t,\tau}$. To do so, we first need to isolate the dependence of $U_\kappa^0$ on $\tilde{s}_{t,\tau}$, for any $\kappa$. Consider the following expansion for $\tilde{s}_{t,\tau}$:

$$\tilde{s}_{t,\tau} = \sum_{k=0}^{\infty} \sum_{\alpha \in \Xi_k} I_\alpha [f_{\alpha,\tau}(s_0)]_{0,t},$$

where $f(x) = x$ and $f_{\alpha,\tau}$ correspond to the coefficient function (as defined in Section C.2) for the dynamics defined by:

$$d\tilde{s}_{t,\tau} = \left( g(\tilde{s}_{t,\tau}) + h(\tilde{s}_{t,\tau}) \left( F_\pi(s; W^0) + \Phi(\tilde{s}_{t,\tau}; W^0)(W^\tau - W^0) \right) \right),$$

which are the dynamics of the agent under a linearized policy at gradient time $\tau$, meaning with parameters $W^\tau$. This dependency arises from $W^\tau - W^0$, since the stochastic gradient-based update in Equation 16. We define by $\mathcal{C}_i$ the set of all possible indices: $\mathbf{c} = \{c_0, c_1, \ldots, c_i\}$ where $c_j \in \mathbb{Z}^+$

with and $0 \leq c_j \leq i$ as $\mathcal{C}_i$. These $c_j$'s correspond to the order of the derivative of $\Phi$ in the Itô -Taylor series of $\tilde{s}_{t,\tau}$. We also define the set of all integers that form the exponents of these derivative terms: $\mathbf{m} = \{m_0, m_1, \ldots, m_i\}$ where $m_j \in \mathbb{Z}^+$ and $0 \leq m_j \leq i$ as $\mathcal{M}_i$. We denote by $\Delta W^\tau = W^\tau - W^0$ The state variable, written as Itô -Taylor series, can be split into two: $\bar{s}_{\tau,t}$ which depends on $\Delta W^\tau$ and the other part $\tilde{s}_{\tau,\tau} - \bar{s}_{\tau,t}$ which is independent of the weights $\Delta W^\tau$. To show the independence of $\tilde{s}_{t,\tau}$ with respect to $U_\kappa^0$, we analyze the variable $\tilde{s}_{t,\tau}$. This variable, $\tilde{s}_{t,\tau}$, can be rewritten as follows.

$$
\tilde{s}_{t,\tau} = \sum_{i=1}^{\infty} \sum_{\alpha \in \Xi_i} \left( \mathbb{B}_\alpha + \sum_{\mathbf{c} \in \mathcal{C}_i, \mathbf{m} \in \mathcal{M}_i} \mathbb{C}_{\mathbf{c},\mathbf{m},\alpha} \Pi_{j=1}^i \left( F_\pi^{(c_j)}(s_0; W^0) + \Phi^{(c_j)}(s_0; W^0) \Delta W^\tau \right)^{m_j} \right) I_\alpha[1]_{0,t},
$$
(20)

where the coefficient $\mathbb{C}_{\mathbf{c},\mathbf{m},\alpha}$ is determined by the iterative application based on coefficient functions of the differential operator described in Section C.2. Also note that $\Xi_i$ is the set of all multi-indices of size $i$ (see Section C.2). We therefore analyze the expression:

$$
\begin{aligned}
\Delta W^\tau &= \sum_{i=1}^{\lfloor \tau/\eta \rfloor} \int_0^T \eta \widehat{\mathcal{G}}(U^{(i-1)\eta}, W^{(i-1)\eta}) \\
&= \sum_{i=1}^{\lfloor \tau/\eta \rfloor} \int_0^T e^{-\beta l} \Phi(\tilde{s}_l^{\pi^{W^{(i-1)\eta}}}; W^0)^\top \left[ \partial_t v^{\lin}(l, \tilde{s}_l^{\pi^{W^{(i-1)\eta}}}; U^{(i-1)\eta}) \right. \\
&\qquad\qquad \left. + r(\tilde{s}_l^{\pi^{W^{(i-1)\eta}}}) - \beta v^{\lin}(l, \tilde{s}_l^{\pi^{W^{(i-1)\eta}}}; U^{(i-1)\eta}) \right] dl.
\end{aligned}
$$

To obtain our results, we will use a version of the Berry–Esséen theorem (Theorem 2.2.14 in the textbook by Tao (2012)).

**Theorem F.1** (Berry–Esséen theorem, less weak form). *Let $X$ have mean zero, unit variance, and finite third moment, and let $F$ be any smooth function, bounded in magnitude by $1$, and Lipschitz. Let*

$$
Z_n := \frac{X_1 + \cdots + X_n}{\sqrt{n}}, \quad \text{where } X_1, \ldots, X_n \text{ are i.i.d. copies of } X.
$$

*Then we have*

$$
\mathbb{E}F(Z_n) = \mathbb{E}F(G) + O\left( \frac{1}{\sqrt{n}} \mathbb{E}|X|^3 \left( 1 + \sup_{x \in \mathbb{R}} |F'(x)| \right) \right),
$$

*where $G \equiv \mathcal{N}(0,1)$.*

To show the independence of random variables: $U_{\kappa,1}^0, \tilde{s}_{t,\tau}$ in the limit $n \to \infty$, we fix $\kappa = n$ without loss of generality and we start with the case of $\tau = \eta$, which means a single gradient update.

$$
\begin{aligned}
\Delta W^\eta &= \eta \int_0^T \widehat{\mathcal{G}}(U^0, W^0) \\
&= \eta e^{-\beta l} \Phi(\tilde{s}_l^{\pi^{W^0}}; W^0)^\top \left[ \quad \partial_t v^{\lin}(l, \tilde{s}_l^{\pi^{W^0}}; U^0) \right. \\
&\qquad\qquad \left. + r(\tilde{s}_l^{\pi^{W^0}}) - \beta v^{\lin}(l, \tilde{s}_l^{\pi^{W^0}}; U^0) \right] dl \\
&= \eta \int_0^T e^{-\beta l} \Phi(\tilde{s}_l^{\pi^{W^0}}; W^0)^\top \left[ \partial_t F_v(\tilde{s}_l^{\pi^{W^0}}, l; U^0, B^0) \right. \\
&\qquad\qquad \left. + r(\tilde{s}_l^{\pi^{W^0}}) - \beta F_v(\tilde{s}_l^{\pi^{W^0}}, l; U^0, B^0) \right] dl.
\end{aligned}
$$

Substituting this into a general expression with $\Delta W$ as a multiplicative factor within the summation of $\tilde{s}_{t,\tau}$ above (Equation 20).

$$\Phi^{(c)}(s_0; W^0)\Delta W^\eta = \eta \Phi^{(c)}(s_0; W^0)\widehat{\mathcal{G}}(U^0, W^0)$$

$$= \eta \int_0^T e^{-\beta l} \Phi^{(c)}(s_0; W^0)\Phi(\tilde{s}_l^{\pi^{W^0}}; W^0)^\top \left[\partial_t v^{\lin}(l, \tilde{s}_l^{\pi^{W^0}}; U^0)\right.$$

$$\left. + r(\tilde{s}_l^{\pi^{W^0}}) - \beta v^{\lin}(l, \tilde{s}_l^{\pi^{W^0}}; U^0)\right] dl.$$

## F.2 Single Gradient Step Dynamics in Variable

The dependence on $B_n^0, U_n^0$ in this expression arises from the expression inside the square brackets:
$\partial_t v^{\lin}(l, \tilde{s}_l^{\pi^{W^0}}; U^0) - \beta v^{\lin}(l, \tilde{s}_l^{\pi^{W^0}}; U^0)$.

$$\partial_t v^{\lin}(l, \tilde{s}_l^{\pi^{W^0}}; U^0) - \beta v^{\lin}(l, \tilde{s}_l^{\pi^{W^0}}; U^0)$$

$$= \frac{1}{\sqrt{n}}\sum_{\kappa=1}^n B_\kappa^0 U_{\kappa,2}^0 \varphi'(U_\kappa \cdot [\tilde{s}_l^{\pi^{W^0}}, l]) - \beta \frac{1}{\sqrt{n}}\sum_{\kappa=1}^n B_\kappa \varphi(U_\kappa^0 \cdot [\tilde{s}_l^{\pi^{W^0}}, l]),$$

Now substitute this expression in the polynomial expansion of $\tilde{s}_{t,\tau}$ (Equation 20).

$$s_{t,\eta} = \sum_{i=1}^\infty \sum_{\alpha\in\Xi_i} \mathbb{B}_\alpha + \left(\sum_{\mathbf{c}\in\mathcal{C}_i, \mathbf{m}\in\mathcal{M}_i} \mathbb{C}_{\mathbf{c},\mathbf{m},\alpha}\Pi_{j=1}^i \left(F_\pi^{(c_j)}(s_0; W^0) + \eta\Phi^{(c_j)}(s_0; W^0)\Delta W^\tau\right)^{m_j} I_\alpha[1]_{0,t}\right),$$

$$= \sum_{i=1}^\infty \sum_{\alpha\in\Xi_i} \mathbb{B}_\alpha + \left(\sum_{\mathbf{c}\in\mathcal{C}_i, \mathbf{m}\in\mathcal{M}_i} \mathbb{C}_{\mathbf{c},\mathbf{m},\alpha}\Pi_{j=1}^i \left(F_\pi^{(c_j)}(s_0; W^0) + \eta\int_0^T e^{-\beta l}\Phi^{(c_j)}(s_0; W^0)\Phi(\tilde{s}_l^{\pi^{W^0}}; W^0)^\top\right.\right.$$

$$\left.\left. \left[\partial_t v^{\lin}(l, \tilde{s}_l^{\pi^{W^0}}; U^0) + r(\tilde{s}_l^{\pi^{W^0}}) - \beta v^{\lin}(l, \tilde{s}_l^{\pi^{W^0}}; U^0)\right] dl\right)^{m_j} I_\alpha[1]_{0,t}\right).$$

We substittue the summation for $\Phi()$:

$$s_{t,\eta} = \sum_{i=1}^\infty \sum_{\alpha\in\Xi_i} \mathbb{B}_\alpha + \left(\sum_{\mathbf{c}\in\mathcal{C}_i, \mathbf{m}\in\mathcal{M}_i} \mathbb{C}_{\mathbf{c},\mathbf{m},\alpha}\Pi_{j=1}^i \left(F_\pi^{(c_j)}(s_0; W^0) + \eta\int_0^T e^{-\beta l}\Phi^{(c_j)}(s_0; W^0)\Phi(\tilde{s}_l^{\pi^{W^0}}; W^0)^\top\right.\right.$$

$$\left[r(\tilde{s}_l^{\pi^{W^0}}) + \right.$$

$$\left.\frac{1}{\sqrt{n}}\sum_{\kappa=1}^n B_\kappa^0\left(U_{\kappa,2}^0\varphi'(U_\kappa^0 \cdot [\tilde{s}_l^{\pi^{W^0}}, l]) - \beta\varphi(U_\kappa^0 \cdot [\tilde{s}_l^{\pi^{W^0}}, l])\right)\right] dl\Big)^{m_j}$$

$$\left. I_\alpha[1]_{0,t}\right)$$

$$= \sum_{i=1}^\infty \sum_{\alpha\in\Xi_i} \mathbb{B}_\alpha + \left(\sum_{\mathbf{c}\in\mathcal{C}_i, \mathbf{m}\in\mathcal{M}_i} \mathbb{C}_{\mathbf{c},\mathbf{m},\alpha}\Pi_{j=1}^i \left(F_\pi^{(c_j)}(s_0; W^0) + \eta\int_0^T e^{-\beta l}\Phi^{(c_j)}(s_0; W^0)\Phi(\tilde{s}_l^{\pi^{W^0}}; W^0)^\top\right.\right.$$

$$\left[r(\tilde{s}_l^{\pi^{W^0}}) + \right.$$

$$\left.\frac{1}{\sqrt{n}}\sum_{\kappa=1}^{n-1} B_\kappa^0\left(U_{\kappa,2}^0\varphi'(U_\kappa^0 \cdot [\tilde{s}_l^{\pi^{W^0}}, l]) - \beta\varphi(U_\kappa^0 \cdot [\tilde{s}_l^{\pi^{W^0}}, l])dl\right)\right.\Big)^{m_j}$$

$$+ \int_0^T \frac{\eta m_j}{\sqrt{n}} B_\kappa^0\left(U_{\kappa,2}^0\varphi'(U_\kappa^0 \cdot [\tilde{s}_l^{\pi^{W^0}}, l]) - \beta\varphi(U_\kappa^0 \cdot [\tilde{s}_l^{\pi^{W^0}}, l]\right) dl$$

$$\left. + O(\eta^2 n^{-1/2})\right]$$

$$\left. I_\alpha[1]_{0,t}\right).$$

Here, the first four equalities are a result of successive substitution, and the last equality is a result of Taylor expansion of the polynomial under the exponent $m_j$. To simplify the expression, we denote by $K(j, W^0, U^0, n, \eta)$ the expression in large brackets $\Big()$ with the exponent $m_j$ i.e.

$$
K(j, W^0, U^0, n, \eta) = \Bigg( F_\pi^{(c_j)}(s_0; W^0) + \eta \int_0^T e^{-\beta l} \Phi^{(c_j)}(s_0; W^0) \Phi(\tilde{s}_l^{\pi^{W^0}}; W^0)^\top
$$
$$
\Big[ r(\tilde{s}_l^{\pi^{W^0}})
$$
$$
+ \frac{1}{\sqrt{n}} \sum_{\kappa=1}^{n-1} B_\kappa^0 \Big( U_{\kappa,2}^0 \varphi'(U_\kappa \cdot [\tilde{s}_l^{\pi^{W^0}}, l]) - \beta \varphi(U_\kappa \cdot [\tilde{s}_l^{\pi^{W^0}}, l]) dl \Big] \Bigg)^{m_j}.
$$

Also, note that this expression, $K$, is independent of $U_n^0, B_n^0$. We further denote by $s_{t,\eta}^{n-1}$ the expression for $s_{t,\eta}$ up to the $n$-th term in the expression above:

$$
s_{t,\eta} = \sum_{i=1}^\infty \sum_{\alpha \in \Xi_i} \mathbb{B}_\alpha + \Bigg[ \sum_{\mathbf{c} \in \mathcal{C}_i, \mathbf{m} \in \mathcal{M}_i} \mathbb{C}_{\mathbf{c},\mathbf{m},\alpha} \Pi_{j=1}^i \Bigg( K(j, W^0, U^0, n, \eta)
$$
$$
+ \frac{\eta m_j}{\sqrt{n}} B_n^0 \Big( \int_0^T U_{n,2}^0 \varphi'(U_n \cdot [\tilde{s}_l^{\pi^{W^0}}, l]) - \beta \varphi(U_n \cdot [\tilde{s}_l^{\pi^{W^0}}, l]) dl \Big) + O(\eta^2 n^{-1/2}) \Bigg] I_\alpha[1]_{0,t}
$$
$$
= \sum_{i=1}^\infty \sum_{\alpha \in \Xi_i} \mathbb{B}_\alpha + \Bigg[ \sum_{\mathbf{c} \in \mathcal{C}_i, \mathbf{m} \in \mathcal{M}_i} \mathbb{C}_{\mathbf{c},\mathbf{m},\alpha} \Big( \Pi_{j=1}^i K(j, W^0, U^0, n, \eta) \Big)
$$
$$
+ \frac{\eta B_n^0}{\sqrt{n}} \sum_{j=1}^i m_j \Bigg( \Big( \int_0^T U_{n,2}^0 \varphi'(U_n \cdot [\tilde{s}_l^{\pi^{W^0}}, l]) - \beta \varphi(U_n \cdot [\tilde{s}_l^{\pi^{W^0}}, l] \Big) \Pi_{\substack{j'=1 \\ j' \neq j}}^i K(j', W^0, U^0, l, n, \eta) dl \Bigg)
$$
$$
+ O(\eta^2 n^{-1/2}) \Bigg] I_\alpha[1]_{0,t}
$$
$$
= \sum_{i=1}^\infty \Bigg[ \sum_{\alpha \in \Xi_i} \mathbb{B}_\alpha + \sum_{\mathbf{c} \in \mathcal{C}_i, \mathbf{m} \in \mathcal{M}_i} \mathbb{C}_{\mathbf{c},\mathbf{m},\alpha} \Big( \Pi_{j=1}^i K(j, W^0, U^0, n, \eta) \Big)
$$
$$
+ \frac{\eta B_\kappa^0}{\sqrt{n}} \sum_{i=1}^\infty \sum_{\alpha \in \Xi_i} \sum_{\mathbf{c} \in \mathcal{C}_i, \mathbf{m} \in \mathcal{M}_i} \mathbb{C}_{\mathbf{c},\mathbf{m},\alpha} \Bigg( \Big( \int_0^T U_{n,2}^0 \varphi'(U_n \cdot [\tilde{s}_l^{\pi^{W^0}}, l]) - \beta \varphi(U_n \cdot [\tilde{s}_l^{\pi^{W^0}}, l] \Big)
$$
$$
\Pi_{\substack{j'=1 \\ j' \neq j}}^i K(j', W^0, U^0, l, n, \eta) dl \Bigg)
$$
$$
+ O(\eta^2 n^{-1/2}) \Bigg] I_\alpha[1]_{0,t},
$$

(21)

where the first equality is through substitution and the second is obtained using a polynomial expansion and suppressing all $O(\eta^2)$ terms. We will denote by $K_{-j,-n}$ the product $m_j \Pi_{\substack{j'=1 \\ j' \neq j}}^i K(j', W^0, U^0, n, \eta)$ and omit other variables as they are apparent from the context. There-fore, we can substitute these expressions into the primary variable of interest in this section that is

$B_n^0 U_{n,1}^0 \varphi'(U_n^0 \cdot [s_{t,\eta}, t]).$

$B_n^0 U_{n,1}^0 \varphi'(U_n^0 \cdot [s_{t,\eta}, t]) =$

$$B_n^0 U_{n,1}^0 \varphi'\left(U_{n,1}^0 \sum_{i=1}^\infty \sum_{\alpha \in \Xi_i} \mathbb{B}_\alpha + \left[\sum_{\mathbf{c} \in \mathcal{C}_i, \mathbf{m} \in \mathcal{M}_i} \mathbb{C}_{\mathbf{c},\mathbf{m},\alpha} \left(\Pi_{j=1}^i K(j, W^0, U^0, l, n, \eta)\right),\right.\right.$$

$$\left.\left. + \frac{\eta B_\kappa^0}{\sqrt{n}} \sum_{j=1}^i \left(\int_0^T \left(U_{n,2}^0 \varphi'(U_\kappa^0 \cdot [\tilde{s}_l^{\pi^{W^0}}, l]) - \beta\varphi(U_n^0 \cdot [\tilde{s}_l^{\pi^{W^0}}, l])\right) K_{-j,-n} dl\right)\right.\right.$$

$$\left.\left. + O(\eta^2 n^{-1/2})\right] I_\alpha[1]_{0,t} + U_{n,2}^0 t\right)$$

which upon being Taylor expanded results in:

$B_n^0 U_{n,1}^0 \varphi'\left(U_n^0 \cdot [s_{t,\eta}, t]\right)$

$$= B_n^0 U_{n,1}^0 \Bigg( \varphi'(U_{n,1}^0 R_{\eta,t,-n} + U_{n,2}^0 t)$$

$$+ \varphi''(U_{n,1}^0 R_{\eta,t,-n} + U_{n,2}^0 t)\left[\frac{\eta B_\kappa^0}{\sqrt{n}} \sum_{i=1}^\infty \sum_{\alpha \in \Xi_i} \mathbb{B}_\alpha + \sum_{\mathbf{c} \in \mathcal{C}_i} \sum_{\mathbf{m} \in \mathcal{M}_i} \sum_{j=1}^i\right.$$

$$\left.\mathbb{C}_{\mathbf{c},\mathbf{m},\alpha} \left(\int_0^T \left(U_{n,2}^0 \varphi'(U_n^0 \cdot [\tilde{s}_l^{\pi^{W^0}}, l]) - \beta\varphi(U_n^0 \cdot [\tilde{s}_l^{\pi^{W^0}}, l]) K_{-j,-n} dl\right)\right] I_\alpha[1]_{0,t}$$

$$+ O(\eta^2 n^{-1/2}) \Bigg).$$

(22)

where the expression $R_{t,\eta,-n}$ that is used to denote all terms that do not involve $B_n^0, U_{n,1}^0$ in the summation of $s_{t,\eta}$:

$$R_{t,\eta,-n} = \sum_{i=1}^\infty \sum_{\alpha \in \Xi_i} \mathbb{B}_\alpha + \left[\sum_{\mathbf{c} \in \mathcal{C}_i, \mathbf{m} \in \mathcal{M}_i} \mathbb{C}_{\mathbf{c},\mathbf{m},\alpha} \left(\Pi_{j=1}^i K(j, W^0, U^0, n, \eta)\right)\right] I_\alpha[1]_{0,t}. \quad (23)$$

Note that $R_{t,\eta,-n}$ converges to $s_{t,\eta}$ as $n \to \infty$. Substituting the general expression for $B_\kappa^0 U_{\kappa,1}^0 \varphi'(U_\kappa^0 \cdot [s_{t,\eta}, t])$ (following Equation 22) into the expression for $Y_{t,\eta}$ (Equation 19) we obtain the following:

$$Y_{t,\eta} = \frac{1}{\sqrt{n}} \sum_{\kappa=1}^n B_\kappa^0 U_{\kappa,1}^0 \varphi'(U_\kappa^0 \cdot [s_{t,\eta}, t])$$

$$= \frac{1}{\sqrt{n}} \sum_{\kappa=1}^n B_\kappa^0 U_{\kappa,1}^0 \varphi'(U_{\kappa,1}^0 R_{t,\eta,-\kappa} + U_{\kappa,2}^0 t)$$

$$+ \frac{\eta}{n} \sum_{\kappa=1}^n \left(B_\kappa^0\right)^2 U_{\kappa,1}^0 \varphi''(U_{\kappa,1}^0 R_{\eta,t,-\kappa} + U_{\kappa,2}^0 t) \sum_{i=1}^\infty \sum_{\alpha \in \Xi_i} \mathbb{B}_\alpha + \sum_{\mathbf{c} \in \mathcal{C}_i} \sum_{\mathbf{m} \in \mathcal{M}_i} \sum_{j=1}^i$$

(24)

$$\left[\mathbb{C}_{\mathbf{c},\mathbf{m},\alpha} \left(\int_0^T \left(U_{\kappa,2}^0 \varphi'(U_\kappa^0 \cdot [\tilde{s}_l^{\pi^{W^0}}, l]) - \beta\varphi(U_\kappa^0 \cdot [\tilde{s}_l^{\pi^{W^0}}, l])\right) K_{-j,-\kappa} dl\right)\right] I_\alpha[1]_{0,t}$$

$$+ O(\eta^2 n^{-1}) I_\alpha[1]_{0,t}.$$

Equation equation 24 provides a decomposition for $s_{t,\eta}$ after one gradient step. Since the expression in the second summation above is not necessarily mean 0, we have the summation of $n$ terms multiplied with $\frac{\eta}{n}$ of order $O(\eta)$.

### F.3 Two Gradient Step Dynamics

Since we already have the result for a single gradient step, which form the base case of our argument in the inductive proof, we derive the expression for an additional gradient step. The reason behind doing so is to ensure that none of the error terms "explode" over multiple gradient steps. Consider the following expansion:

$$
\begin{aligned}
Y_{t,2\eta} =& \frac{1}{\sqrt{n}} \sum_{\kappa=1}^{n} B_\kappa^0 U_{\kappa,1}^0 \varphi'(U_\kappa^0 \cdot [s_{t,2\eta}, t]) \\
=& \frac{1}{\sqrt{n}} \sum_{\kappa=1}^{n} B_\kappa^0 U_{\kappa,1}^0 \varphi'(U_{\kappa,1}^0 R_{t,2\eta,-\kappa} + U_{\kappa,2}^0 t) \\
&+ \frac{\eta}{n} \sum_{\kappa=1}^{n} \left(B_\kappa^0\right)^2 U_{\kappa,1}^0 \varphi''(U_{\kappa,1}^0 R_{2\eta,t,-\kappa} + U_{\kappa,2}^0 t) \sum_{i=1}^{\infty} \sum_{\alpha \in \Xi_i} \mathbb{B}_\alpha + \sum_{\mathbf{c} \in \mathcal{C}_i} \sum_{\mathbf{m} \in \mathcal{M}_i} \sum_{j=1}^{i} \\
&\left[ \mathbb{C}_{\mathbf{c},\mathbf{m},\alpha} \left( \int_0^T \left( U_{\kappa,2}^0 \varphi'(U_\kappa \cdot [\tilde{s}_{l,\eta}, l]) - \beta \, \varphi(U_\kappa \cdot [\tilde{s}_{l,\eta}, l]) \right) K_{-j,-\kappa} dl \right) \right] I_\alpha[1]_{0,t} \\
&+ O(\eta^2 n^{-1}) I_\alpha[1]_{0,t}.
\end{aligned}
\tag{25}
$$

We further expand the expression for $R_{t,2\eta,-\kappa}$.

$$
R_{t,2\eta,-\kappa} = \sum_{i=1}^{\infty} \sum_{\alpha \in \Xi_i} \mathbb{B}_\alpha + \left[ \sum_{\mathbf{c} \in \mathcal{C}_i, \mathbf{m} \in \mathcal{M}_i} \mathbb{C}_{\mathbf{c},\mathbf{m},\alpha} \left( \Pi_{j=1}^{i} K(j, W^0, U^0, \kappa, 2\eta) \right) \right] I_\alpha[1]_{0,t}.
\tag{26}
$$

Upon substituting the value for $K(j, W^0, U^0, l, n, 2\eta)$, we obtain the following.

$$
\begin{aligned}
K(j, W^0, U^0, n, 2\eta) =& \left( F_\pi^{(c_j)}(s_0; W^0) + \eta \int_0^T e^{-\beta u} \Phi^{(c_j)}(s_0; W^0) \Phi(\tilde{s}_{u,\eta}; W^0)^\top \right. \\
& \left[ r(\tilde{s}_{u,\eta}) \right. \\
& \left. \left. + \frac{1}{\sqrt{n}} \sum_{\kappa=1}^{n-1} B_\kappa^0 \left( U_{\kappa,2}^0 \varphi'(U_\kappa^0 \cdot [\tilde{s}_{u,\eta}, u]) - \beta \varphi(U_\kappa^0 \cdot [\tilde{s}_{u,\eta}, u]) du \right] \right)^{m_j} \right..
\end{aligned}
\tag{27}
$$

Once again using the shorthand $K_{-j,-\kappa} = m_j \Pi_{\substack{j'=1 \\ j' \neq j}}^{i} K(j', W^0, U^0, n, 2\eta)$. We first decompose the first expression.

$$
\begin{aligned}
& \Phi^{(c_j)}(s_0; W^0) \Phi(\tilde{s}_{u,\eta}; W^0)^\top \\
=& \frac{1}{n} \sum_{\kappa=1}^{n} \left[ C_\kappa^0 \tilde{s}_{u,\eta} \varphi'(W_\kappa^0 \tilde{s}_{u,\eta}) \times \right. \\
& \left. \left( (W_\kappa^0)^{c_j} \tilde{s}_0 C_\kappa^0 \varphi^{(c_j+1)}(W_\kappa^0 \tilde{s}_0) + c_j C_\kappa^0 (W_\kappa^0)^{c_j-1} \varphi^{(c_j)}(W_\kappa^0 \tilde{s}_0) \right) \right] \\
=& \frac{1}{n} \sum_{\kappa=1}^{n} \left[ C_\kappa^0 \left( R_{\eta,u,-\kappa} \varphi'(W_\kappa^0 R_{\eta,u,-\kappa}) \right. \right. \\
& + \varphi''(W_\kappa^0 R_{\eta,u,-\kappa}) \frac{\eta B_\kappa^0}{\sqrt{n}} \sum_{j=1}^{i} \left( \int_0^T \left( U_{\kappa,2}^0 \varphi'(U_\kappa^0 \cdot [\tilde{s}_{0,u}, u]) - \beta \varphi(U_\kappa^0 \cdot [\tilde{s}_{0,u}, u]) \right) K_{-j,-\kappa} du \right) \\
& \left. + O(\eta^2 n^{-1/2}) \right] I_\alpha[1]_{0,l} \bigg) \bigg) \times \\
& \left. \left( (W_\kappa^0)^{c_j} \tilde{s}_{0,\eta} C_\kappa^0 \varphi^{(c_j+1)}(W_\kappa^0 \tilde{s}_{0,\eta}) + c_j C_\kappa^0 (W_\kappa^0)^{c_j-1} \varphi^{(c_j)}(W_\kappa^0 \tilde{s}_{0,\eta}) \right) \right],
\end{aligned}
$$

where we utilize the Taylor expression as in Equation 22 and the substitution in Equation 23 for $\kappa$ instead of $n$. Therefore, we can ignore the expressions, other than $R_{\eta,l,-n}\varphi'(W_\kappa^0 R_{\eta,l,-n})$ because they have the leading term of $O(\eta n^{-3/2})$ while the summation is over the $n$ variables. Decomposing the other expressions in Equation 27:

$$
\begin{aligned}
r(\tilde{s}_{l,\eta}) =& r(R_{\eta,l,-\kappa}) \\
&+ r'(R_{\eta,l,-\kappa})\frac{\eta B_\kappa^0}{\sqrt{n}}\sum_{i=1}^\infty \sum_{\alpha\in\Xi_i}\mathbb{B}_\alpha + \sum_{\mathbf{c}\in\mathcal{C}_i}\sum_{\mathbf{m}\in\mathcal{M}_i}\sum_{j=1}^i \\
&\mathbb{C}_{\mathbf{c},\mathbf{m},\alpha}\left(\int_0^T \left(U_{\kappa,2}^0\varphi'(U_\kappa^0\cdot[\tilde{s}_{u,0},u]) - \beta\varphi(U_\kappa^0\cdot[\tilde{s}_{u,0},u])\right)K_{-j,-\kappa}du\right)\Bigg]I_\alpha[1]_{0,l} \\
&+ O(\eta^2 n^{-1/2}).
\end{aligned}
$$

For succinctness, we denote by

$$
\phi^{(c_j)}(s_0, W_\kappa^0, C_\kappa^0) = C_\kappa^0\left((W_\kappa^0)^{c_j}\tilde{s}_0\varphi^{(c_j+1)}(W_\kappa^0\tilde{s}_0) + c_j(W_\kappa^0)^{c_j-1}\varphi^{(c_j)}(W_\kappa^0\tilde{s}_0)\right). \tag{28}
$$

Therefore, following a similar expansion for $\varphi', \varphi$ we can expand the expression in Equation 27.

$$
\begin{aligned}
&K(j,W^0,U^0,n,2\eta) \\
&= \left(F_\pi^{(c_j)}(s_0; W^0) + \eta\int_0^T e^{-\beta u}\Phi^{(c_j)}(s_0; W^0)\Phi(\tilde{s}_{l,\eta}; W^0)^\top\right. \\
&\qquad\left[r(\tilde{s}_{u,\eta})\right. \\
&\qquad\left.\left.+\frac{1}{\sqrt{n}}\sum_{\kappa=1}^{n-1}B_\kappa^0\left(U_{\kappa,2}^0\varphi'(U_\kappa^0\cdot[\tilde{s}_{u,\eta},u]) - \beta\varphi(U_\kappa^0\cdot[\tilde{s}_{u,\eta},u])du\right]\right)^{m_j}\right. \\
&= \left(F_\pi^{(c_j)}(s_0; W^0) + \eta\int_0^T e^{-\beta u}\left[\left(\frac{1}{n}\sum_{\kappa=1}^{n-1}C_\kappa^0 R_{\eta,u,-n}\varphi'(W_\kappa^0 R_{\eta,u,-n})\phi^{(c_j)}(s_0,W_\kappa^0,C_\kappa^0)\right) + O(\eta n^{-1})\right]\right. \\
&\qquad\times\left[r(R_{\eta,u,-n}) + \frac{1}{\sqrt{n}}\sum_{\kappa'=1}^{n-1}B_{\kappa'}^0\left(U_{\kappa',2}^0\varphi'(U_{\kappa'}^0\cdot[R_{\eta,u,-n},u]) - \beta\varphi(U_{\kappa'}^0\cdot[R_{\eta,u,-n},u])\right)\right. \\
&\qquad+\left(r'(R_{\eta,u,-n}) + \frac{1}{\sqrt{n}}\sum_{\kappa'=1}^{n-1}B_{\kappa'}^0\left(U_{\kappa',2}^0\varphi''(U_{\kappa'}^0\cdot[R_{\eta,u,-n},u]) - \beta\varphi'(U_{\kappa'}^0\cdot[R_{\eta,u,-n},u]\right)\right. \\
&\qquad\frac{\eta B_n^0}{\sqrt{n}}\sum_{i=1}^\infty\sum_{\alpha\in\Xi_i}\mathbb{B}_\alpha + \sum_{\mathbf{c}\in\mathcal{C}_i}\sum_{\mathbf{m}\in\mathcal{M}_i}\sum_{j=1}^i\mathbb{C}_{\mathbf{c},\mathbf{m},\alpha}\left(\int_0^T\left(U_{n,2}^0\varphi'(U_n^0\cdot[\tilde{s}_{l,0},l]) - \beta\varphi(U_n^0\cdot[\tilde{s}_{l,0},l])\right)K_{-j,-n}dl\right) \\
&\qquad\left.\left.+ O(\eta^2 n^{-1/2})\right]du\right)^{m_j} \\
&= \left(F_\pi^{(c_j)}(s_0; W^0) + \eta\int_0^T e^{-\beta u}\left[\left(\frac{1}{n}\sum_{\kappa=1}^{n-1}C_\kappa^0 R_{\eta,u,-n}\varphi'(W_\kappa^0 R_{\eta,u,-n})\phi^{(c_j)}(s_0,W_\kappa^0,C_\kappa^0)\right)\right]\right. \\
&\qquad+\left[r(R_{\eta,u,-n}) + \frac{1}{\sqrt{n}}\sum_{\kappa'=1}^{n-1}B_{\kappa'}^0\left(U_{\kappa',2}^0\varphi'(U_{\kappa'}^0\cdot[R_{\eta,u,-n},u]) - \beta\varphi(U_{\kappa'}^0\cdot[R_{\eta,u,-n},u])\right)\right]du \\
&\qquad\left.+ O(\eta^2 n^{-1/2})\right)^{m_j}.
\end{aligned}
$$

We note that in the expression above, every term except $O(\eta^2 n^{-1/2})$ is independent of $U_n^0, B_n^0$. We substitute this expression for $K(j, W^0, U^0, n, 2\eta)$ into $R_{t,2\eta,-n}$ (Equation 26):

$$
\begin{aligned}
R_{t,2\eta,-n} = &\sum_{i=1}^{\infty} \sum_{\alpha \in \Xi_i} \mathbb{B}_\alpha + \Bigg[ \sum_{\mathbf{c} \in \mathcal{C}_i, \mathbf{m} \in \mathcal{M}_i} \mathbb{C}_{\mathbf{c},\mathbf{m},\alpha} \Pi_{j=1}^i \bigg( F_\pi^{(c_j)}(s_0; W^0) \\
&+ \eta \int_0^T e^{-\beta u} \left[ \Big(\frac{1}{n} \sum_{\kappa=1}^{n-1} C_\kappa^0 R_{\eta,u,-n} \varphi'(W_\kappa^0 R_{\eta,u,-n}) \phi^{(c_j)}(s_0, W_\kappa^0, C_\kappa^0) \Big) \right] \\
&+ \left[ r(R_{\eta,u,-n}) + \frac{1}{\sqrt{n}} \sum_{\kappa'=1}^{n-1} B_{\kappa'}^0 \Big( U_{\kappa',2}^0 \varphi'(U_{\kappa'}^0 \cdot [R_{\eta,u,-n}, u]) - \beta \varphi(U_{\kappa'}^0 \cdot [R_{\eta,u,-n}, u]) \Big) \right] du \\
&+ O(\eta^2 n^{-1/2}) \bigg)^{m_j} \Bigg] I_\alpha[1]_{0,t} \\
= &\sum_{i=1}^{\infty} \sum_{\alpha \in \Xi_i} \mathbb{B}_\alpha + \Bigg[ \sum_{\mathbf{c} \in \mathcal{C}_i, \mathbf{m} \in \mathcal{M}_i} \mathbb{C}_{\mathbf{c},\mathbf{m},\alpha} \Pi_{j=1}^i \bigg( F_\pi^{(c_j)}(s_0; W^0) \\
&+ \eta \int_0^T e^{-\beta u} \left[ \Big(\frac{1}{n} \sum_{\kappa=1}^{n-1} C_\kappa^0 R_{\eta,u,-n} \varphi'(W_\kappa^0 R_{\eta,u,-n}) \phi^{(c_j)}(s_0, W_\kappa^0, C_\kappa^0) \Big) \right] \\
&+ \left[ r(R_{\eta,u,-n}) + \frac{1}{\sqrt{n}} \sum_{\kappa'=1}^{n-1} B_{\kappa'}^0 \Big( U_{\kappa',2}^0 \varphi'(U_{\kappa'}^0 \cdot [R_{\eta,u,-n}, u]) - \beta \varphi(U_{\kappa'}^0 \cdot [R_{\eta,u,-n}, u]) \Big) \right] du \bigg)^{m_j} \\
&+ \sum_{j=1}^i m_j O(\eta^2 n^{-1/2}) K_{-j,-n} \Bigg] I_\alpha[1]_{0,t}
\end{aligned}
$$

Substituting this into the expression for $Y_{t,2\eta}$ we obtain a summation over i.i.d. terms while separating out the terms with dependent terms, similar to the one-step update in Equation 24.

$$
\begin{aligned}
Y_{t,2\eta} = &\frac{1}{\sqrt{n}} \sum_{\kappa=1}^n B_\kappa^0 U_{\kappa,1}^0 \varphi'(U_{\kappa,1}^0 R_{t,2\eta,-\kappa,-\kappa} + U_{\kappa,2}^0 t) \\
&+ \frac{\eta}{n} \sum_{\kappa=1}^n \left(B_\kappa^0\right)^2 U_{\kappa,1}^0 \varphi''(U_{\kappa,1}^0 R_{2\eta,t,-\kappa} + U_{\kappa,2}^0 t) \sum_{i=1}^{\infty} \sum_{\alpha \in \Xi_i} \mathbb{B}_\alpha + \sum_{\mathbf{c} \in \mathcal{C}_i} \sum_{\mathbf{m} \in \mathcal{M}_i} \sum_{j=1}^i \\
&\left[ \mathbb{C}_{\mathbf{c},\mathbf{m},\alpha} \left( \int_0^T \Big( U_{\kappa,2}^0 \varphi'(U_\kappa \cdot [\tilde{s}_{l,\eta}, l]) - \beta \, \varphi(U_\kappa \cdot [\tilde{s}_{l,\eta}, l]) \Big) K_{-j,-\kappa} dl \right) \right] I_\alpha[1]_{0,t} \\
&+ \frac{\eta^2}{n} \sum_{\kappa=1}^n O(1),
\end{aligned}
$$

where $R_{t,2\eta,-n,-n}$ is the expression as follows:

$$
\begin{aligned}
R_{t,2\eta,-n,-n} = &\sum_{i=1}^{\infty} \sum_{\alpha \in \Xi_i} \mathbb{B}_\alpha + \Bigg[ \sum_{\mathbf{c} \in \mathcal{C}_i, \mathbf{m} \in \mathcal{M}_i} \mathbb{C}_{\mathbf{c},\mathbf{m},\alpha} \Pi_{j=1}^i \bigg( F_\pi^{(c_j)}(s_0; W^0) \\
&+ \eta \int_0^T e^{-\beta u} \left[ \Big(\frac{1}{n} \sum_{\kappa=1}^{n-1} C_\kappa^0 R_{\eta,u,-n} \varphi'(W_\kappa^0 R_{\eta,u,-n}) \phi^{(c_j)}(s_0, W_\kappa^0, C_\kappa^0) \Big) \right] \\
&+ \left[ r(R_{\eta,u,-n}) + \frac{1}{\sqrt{n}} \sum_{\kappa'=1}^{n-1} B_{\kappa'}^0 \Big( U_{\kappa',2}^0 \varphi'(U_{\kappa'}^0 \cdot [R_{\eta,u,-n}, u]) - \beta \varphi(U_{\kappa'}^0 \cdot [R_{\eta,u,-n}, u]) \Big) \right] du \bigg)^{m_j} \Bigg],
\end{aligned}
$$

which represents the removal of the dependency on the random variables at index $n$ across two gradient steps, and $R_{t,2\eta,-\kappa,-\kappa}$ follows similarly.

### F.4 GROWTH OF RESIDUAL PART IN STATE VARIABLE

Following the expressions for $s_{t,\eta}, s_{t,2\eta}$ we can prove the following proposition.

**Proposition F.2.** *Given a $\tau$ that is a multiple of $\eta$ and $\kappa \le n$ we find that $\tilde{s}_{t,\tau}$ is the sum of two variables: one independent of $B_\kappa^0, U_\kappa^0$ and another of order $O(\eta n^{-1/2})$.*

*Proof.* We prove this by induction. For the initial gradient step $\tau = \eta$, this is evident from the decomposition in Equation 21. Note that once again we prove for $n$ to simplify the summation and notation and all these results apply for $\kappa$. To prove for general $\tau$ we assume that it holds for $\tau - \eta$ and then prove that this case holds for $\tau$. For $\tau - \eta$, we write $s_{t,\tau-\eta} = \bar{R}_{t,\tau-\eta,-n} + O(\eta n^{-1/2})$, where $\bar{R}_{t,\tau-\eta,-n}$ is the expression independent of $B_n^0, U_n^0$. First, we define $R_{t,\tau,-n}$, following the notation and explanation provided in Section F.2:

$$R_{t,\tau,-n} = \sum_{i=1}^{\infty} \sum_{\alpha \in \Xi_i} \mathbb{B}_\alpha + \left[ \sum_{\mathbf{c} \in \mathcal{C}_i, \mathbf{m} \in \mathcal{M}_i} \mathbb{C}_{\mathbf{c},\mathbf{m},\alpha} \left( \Pi_{j=1}^i K(j, W^0, U^0, n, \tau) \right) \right] I_\alpha[1]_{0,t}, \text{ where}$$

$$K(j, W^0, U^0, n, \tau) = \left( F_\pi^{(c_j)}(s_0; W^0) + \eta \int_0^T e^{-\beta l} \Phi^{(c_j)}(s_0; W^0) \Phi(\tilde{s}_{l,\tau-\eta}; W^0)^\top \right.$$

$$\left[ r(\tilde{s}_{l,\tau-\eta}) \right.$$

$$\left. + \frac{1}{\sqrt{n}} \sum_{\kappa=1}^{n-1} B_\kappa^0 \left( U_{\kappa,2}^0 \varphi'(U_\kappa \cdot [\tilde{s}_{l,\tau-\eta} - \beta\varphi(U_\kappa \cdot [\tilde{s}_{l,\tau-\eta} dl] \right) \right)^{m_j}.$$

Therefore, similar to Section F.3 we can rewrite $K(j, W^0, U^0, n, \tau)$ as:

$K(j, W^0, U^0, n, \tau)$

$$= \left( F_\pi^{(c_j)}(s_0; W^0) + \eta \int_0^T e^{-\beta u} \Phi^{(c_j)}(s_0; W^0) \Phi(\tilde{s}_{l,\tau-\eta}; W^0)^\top \right.$$

$$\left[ r(\tilde{s}_{u,\tau-\eta}) \right.$$

$$\left. + \frac{1}{\sqrt{n}} \sum_{\kappa=1}^{n-1} B_\kappa^0 \left( U_{\kappa,2}^0 \varphi'(U_\kappa^0 \cdot [\tilde{s}_{u,\tau-\eta}, u]) - \beta\varphi(U_\kappa^0 \cdot [\tilde{s}_{u,\tau-\eta}, u]) du \right] \right)^{m_j}$$

$$= \left( F_\pi^{(c_j)}(s_0; W^0) + \eta \int_0^T e^{-\beta u} \left[ \left( \frac{1}{n} \sum_{\kappa=1}^{n-1} C_\kappa^0 \bar{R}_{\eta,u,-n} \varphi'(W_\kappa^0 R_{\eta,u,-n}) \phi^{(c_j)}(s_0, W_\kappa^0, C_\kappa^0) \right) \right] \right.$$

$$+ \left[ r(\bar{R}_{\tau-\eta,u,-n}) + \frac{1}{\sqrt{n}} \sum_{\kappa'=1}^{n-1} B_{\kappa'}^0 \left( U_{\kappa',2}^0 \varphi'(U_{\kappa'}^0 \cdot [\bar{R}_{\tau-\eta,u,-n}, u]) - \beta\varphi(U_{\kappa'}^0 \cdot [\bar{R}_{\tau-\eta,u,-n}, u]) \right) \right] du$$

$$\left. + O(\eta^2 n^{-1/2}) \right)^{m_j}.$$

Substituting this into the expression of $R_{t,\tau,-n}$ above we obtain:

$$R_{t,\tau,-n} = \sum_{i=1}^{\infty} \sum_{\alpha \in \Xi_i} \mathbb{B}_\alpha + \left[ \sum_{\mathbf{c} \in \mathcal{C}_i, \mathbf{m} \in \mathcal{M}_i} \mathbb{C}_{\mathbf{c},\mathbf{m},\alpha} \Pi_{j=1}^i \left( F_\pi^{(c_j)}(s_0; W^0) \right. \right.$$

$$+ \eta \int_0^T e^{-\beta u} \left[ \left( \frac{1}{n} \sum_{\kappa=1}^{n-1} C_\kappa^0 \bar{R}_{\tau-\eta,u,-n} \varphi'(W_\kappa^0 \bar{R}_{\tau-\eta,u,-n}) \phi^{(c_j)}(s_0, W_\kappa^0, C_\kappa^0) \right) \right]$$

$$+ \left[ r(\bar{R}_{\tau-\eta,u,-n}) + \frac{1}{\sqrt{n}} \sum_{\kappa'=1}^{n-1} B_{\kappa'}^0 \left( U_{\kappa',2}^0 \varphi'(U_{\kappa'}^0 \cdot [\bar{R}_{\tau-\eta,u,-n}, u]) - \beta\varphi(U_{\kappa'}^0 \cdot [\bar{R}_{\tau-\eta,u,-n}, u]) \right) \right] du \right)^{m_j}$$

$$\left. + \sum_{j=1}^i m_j O(\eta^2 n^{-1/2}) K_{-j,-n} \right] I_\alpha[1]_{0,t},$$

which we simplify by removing the $I_\alpha[1]_{0,t}$ expression:

$$R_{t,\tau,-n} = \sum_{i=1}^\infty \sum_{\alpha\in\Xi_i} \mathbb{B}_\alpha + \Bigg[ \sum_{\mathbf{c}\in\mathcal{C}_i,\mathbf{m}\in\mathcal{M}_i} \mathbb{C}_{\mathbf{c},\mathbf{m},\alpha}\Pi_{j=1}^i \Bigg( F_\pi^{(c_j)}(s_0;W^0)$$

$$+ \eta \int_0^T e^{-\beta u} \Bigg[ \Big( \frac{1}{n}\sum_{\kappa=1}^{n-1} C_\kappa^0 \bar{R}_{\tau-\eta,u,-n}\varphi'(W_\kappa^0 \bar{R}_{\tau-\eta,u,-n})\phi^{(c_j)}(s_0,W_\kappa^0,C_\kappa^0) \Big) \Bigg] \Bigg]$$

$$+ \Bigg[ r(\bar{R}_{\tau-\eta,u,-n})$$

$$+ \frac{1}{\sqrt{n}}\sum_{\kappa'=1}^{n-1} B_{\kappa'}^0 \Big( U_{\kappa',2}^0\varphi'(U_{\kappa'}^0 \cdot [\bar{R}_{\tau-\eta,u,-n},u]) - \beta\varphi(U_{\kappa'}^0 \cdot [\bar{R}_{\tau-\eta,u,-n},u]) \Big) \Bigg] du \Bigg)^{m_j} I_\alpha[1]_{0,t}$$

$$+ O(\eta^2 n^{-1/2}) \sum_{i=1}^\infty \sum_{\alpha\in\Xi_i} \sum_{j=1}^i \mathbb{C}_{\mathbf{c},\mathbf{m},\alpha} K_{-j,-n} I_\alpha[1]_{0,t}.$$

We denote the part of the expression that does not include the $O(\eta^2 n^{-1/2})$ as $\bar{R}_{\tau,t,-n}$ which is independent of $B_n^0, U_n^0$. Therefore, following the definition of $R_{t,\tau,-n}$ and $\tilde{s}_{t,\tau}$ we can write:

$$\tilde{s}_{t,\tau} = \bar{R}_{\tau,t,-n} + O(\eta^2 n^{-1/2})$$

$$+ \frac{\eta B_\kappa^0}{\sqrt{n}} \sum_{i=1}^\infty \sum_{\alpha\in\Xi_i} \mathbb{B}_\alpha + \sum_{\mathbf{c}\in\mathcal{C}_i,\mathbf{m}\in\mathcal{M}_i} \mathbb{C}_{\mathbf{c},\mathbf{m},\alpha} \Bigg( \Big( \int_0^T U_{n,2}^0\varphi'(U_n \cdot [\tilde{s}_{l,\tau-\eta},l]) - \beta\varphi(U_n \cdot [\tilde{s}_{l,\tau-\eta},l]) \Big)$$

$$\Pi_{\substack{j'=1 \\ j'\neq j}}^i K(j',W^0,U^0,l,n,\eta)dl \Bigg),$$

which proves the statement of the proposition. $\qquad\square$

## G  GENERALIZED LEMMA FOR GRADIENT UPDATES

Using the single-step and two-step which we now use to establish the following lemma. We further assume that $\varphi$ and $\varphi'$ are Lipschitz continuous. To prove this result, we first state a type of Berry-Eseen theorem for Martingales which will be used for proving a conditional central limit theorem in what follows. This is a restatement of the main result, theorem 1, by Haeusler (1988) with .

**Theorem G.1** (Haeusler (1988), simplified version). *Let $(X_k, \mathcal{F}_k)_{k\geq 1}$ be a sequence of square-integrable martingale differences, i.e.*

$$\mathbb{E}[X_k \mid \mathcal{F}_{k-1}] = 0, \qquad \mathbb{E}[X_k^2] < \infty,$$

*such that $\mathcal{F}_0 \subseteq \mathcal{F}_1 \subseteq \mathcal{F}_2 \dots$.*

*Define the partial sums*

$$S_n = \sum_{k=1}^n X_k,$$

*the $\mathcal{F}_0$-conditional variance*

$$D_n^2 := \sum_{k=1}^n \mathbb{E}[X_k^2|\mathcal{F}_0],$$

*and the predictable quadratic variation*

$$\langle S\rangle_n := \sum_{k=1}^n \mathbb{E}[X_k^2 \mid \mathcal{F}_{k-1}].$$

*Let $\nu(x)$ denote the standard normal distribution function. Then there exists a universal constant $C > 0$ such that*

$$\sup_{x\in\mathbb{R}} \left| \mathbb{P}\Big( \frac{S_n}{B_n} \leq x \Big) - \nu(x) \right| \leq C(L_n + N_n),$$

*where*

$$L_n := \frac{1}{B_n^3} \sum_{k=1}^n \mathbb{E}[|X_k|^3], \qquad N_n := \frac{1}{B_n^3} \mathbb{E}\left[\,|\langle S \rangle_n - D_n^2|^{3/2}\right].$$

Equipped with this theorem, we prove the following key lemma.

**Lemma G.2.** *At gradient step $\tau$, which is an integer multiple of $\eta$, a random variable of the form:*

$$Y_{t,\tau} = \frac{1}{\sqrt{n}} \sum_{\kappa=1}^n B_\kappa^0 U_{\kappa,1}^0 \varphi'(U_\kappa^0 \cdot [\tilde{s}_{t,\tau}, t]),$$

*conditioned upon $\tilde{s}_{t,\tau} = s$ is equal in distribution to the sum of a 0 centered Gaussian random variable with variance:*

$$\mathbb{E}\left[B^2 U_1^2 \left(\varphi'(U_1 s + U_2 t)\right)^2\right], \text{ where } U_1, U_2 \sim \mathcal{N}(0,1), B \sim \text{Unif}(-1,1),$$

*up to an error term of order $O(1/\sqrt{n})$.*

*Proof.* We prove this using proposition F.2 for $\tau$, which are integer multiples of $\eta$.

**Bounding the error:** For a fixed scalar $s$ we have that $Y_t(s) = \frac{1}{\sqrt{n}} \sum_{\kappa=1}^n B_\kappa^0 U_{\kappa,1}^0 \varphi'(U_\kappa^0 \cdot [s, t])$ is equal to a Gaussian of variance $\mathbb{E}\left[B^2 U_1^2 \left(\varphi'(U_1 s + U_2 t)\right)^2\right]$ plus $O(1/\sqrt{n})$ by Theorem F.1. The variance will be denoted by $\text{var}(s) = \mathbb{E}\left[B^2 U_1^2 \left(\varphi'(U_1 s + U_2 t)\right)^2\right]$. From proposition F.2 we know that $\tilde{s}_{t,\tau} = \bar{R}_{t,\tau,-\kappa} + O(\eta n^{-1/2})$, where $\bar{R}_{t,\tau,-\kappa}$ is independent of $B_\kappa^0, U_\kappa^0$. Let ,

$$Z_{t,\tau} = \frac{1}{\sqrt{n}} \sum_{\kappa=1}^n B_\kappa^0 U_{\kappa,1}^0 \varphi'(U_\kappa^0 \cdot [R_{t,\tau,-\kappa}, t]) \text{ and } X_\kappa = \frac{1}{\sqrt{n}} B_\kappa^0 U_{\kappa,1}^0 \varphi'(U_\kappa^0 \cdot [\bar{R}_{t,\tau,-\kappa}, t]).$$

Let $\mathcal{F}_0$ be the event $s_{t,\tau} = s$ and the subsequent $\mathcal{F}.$ be defined canonically such that $B_\kappa^0, U_\kappa^0, \bar{R}_{t,\tau,-\kappa}$ are measurable and $\mathcal{F}_0 \subseteq \mathcal{F}_1 \ldots \subseteq \mathcal{F}_\kappa$. Clearly,

$$\mathbb{E}[X_\kappa | \mathcal{F}_{\kappa-1}] = \mathbb{E}\left[B_\kappa^0 | \mathcal{F}_{\kappa-1}\right] \mathbb{E}\left[U_{\kappa,1}^0 \varphi'(U_\kappa^0 \cdot [\bar{R}_{t,\tau,-\kappa}, t]) | \mathcal{F}_{\kappa-1}\right] = 0,$$

since $B_\kappa^0$ is independent of $\bar{R}_{t,\tau,-\kappa}$ (see Equation 23). Now we note that, following the notation in theorem G.1, we have the $\mathcal{F}_0$-conditional variance:

$$D_n^2 := \frac{1}{n} \sum_{\kappa=1}^n \mathbb{E}\left[(B_\kappa^0)^2 (U_{\kappa,1}^0)^2 \varphi(U_\kappa^0 \cdot [\bar{R}_{t,\tau,-\kappa}])^2 | \mathcal{F}_0\right],$$

which is non-zero if $\varphi(U_\kappa^0 \cdot [\bar{R}_{t,\tau,-\kappa}])$ is measurably non-zero. Note that $|D_n - \text{var}(s)|$ is $O(n^{-1/2}\eta)$ because $|\bar{R}_{t,\tau,-\kappa} - s|$ is also $O(n^{-1/2}\eta)$. Further, the predictable quadratic variation is defined as:

$$\langle S \rangle_n := \sum_{k=1}^n \mathbb{E}[X_k^2 \mid \mathcal{F}_{k-1}],$$

and since we have $\varphi'$ is Lipschitz with some Lipschitz constant $C_{\varphi'}$ and $\bar{R}_{t,\tau,-\kappa} - \tilde{s}_{t,\tau} = O(n^{-1/2}\eta)$, we have:

$$\langle S \rangle_n = \mathbb{E}\left[B^2 U_1^2 \left(\varphi'(U_1 \tilde{s}_{t,\tau} + U_2 t)\right)^2\right] + O(\eta),$$

where the expectation is over the randomness of $U_1, U_2, B$. Consequently, we have

$$N_n = O((\eta)^{3/2}).$$

We also have the following:

$$L_n = \frac{1}{D_n^3} \frac{1}{n^{3/2}} \sum_{\kappa=1}^n \mathbb{E}\left[|B_\kappa^0 U_{\kappa,1}^0 \varphi'(U_\kappa^0 \cdot [\tilde{s}_{t,\tau}, t])|^3\right] = O\left(\frac{1}{\sqrt{n}}\right),$$

since $|B_\kappa^0 U_{\kappa,1}^0 \varphi'(U_\kappa^0 \cdot [\tilde{s}_{t,\tau}, t])| = O(1)$. Therefore, by theorem G.1 we have the following:

$$\sup_{x \in \mathbb{R}} \left| \mathbb{P}\left(\frac{S_n}{B_n} \leq x\right) - \nu(x) \right| = O\left(\frac{1}{\sqrt{n}}\right).$$

This proves the statement of the lemma.

$\square$

## H  CONDITIONAL LAW OF LARGE NUMBERS

Using the Martingale CLT above (Theorem G.1), we provide a corollary which will be applied to obtain a conditional law of large numbers, which will be used in subsequent proofs to bound expressions to 0 by $O(1/\sqrt{n})$.

**Theorem H.1** (Conditional LLN via martingale CLT). *Let $(Y_k, \mathcal{F}_k)_{k \geq 1}$ be $\mathcal{F}_k$-adapted with $\mu := \mathbb{E}[Y_k]$ and $\mathbb{E}|Y_k|^3 < \infty$, and fix a sub-$\sigma$-field $\mathcal{F}_0 \subseteq \mathcal{F}_1$. Given the Doob split*

$$Y_k - \mu = X_k + A_k, \qquad X_k := Y_k - \mathbb{E}(Y_k \mid \mathcal{F}_{k-1}), \quad A_k := \mathbb{E}(Y_k \mid \mathcal{F}_{k-1}) - \mu,$$

*and the partial sums*

$$M_n := \sum_{k=1}^n X_k, \qquad R_n := \sum_{k=1}^n A_k, \qquad S_n := \sum_{k=1}^n Y_k = \mu n + M_n + R_n.$$

*We also we define the $\mathcal{F}_0$-conditional variance and the predictable quadratic variation as:*

$$B_n^2(\mathcal{F}_0) := \mathbb{E}[M_n^2 \mid \mathcal{F}_0] = \sum_{k=1}^n \mathbb{E}[X_k^2 \mid \mathcal{F}_0], \qquad \langle M \rangle_n := \sum_{k=1}^n \mathbb{E}[X_k^2 \mid \mathcal{F}_{k-1}].$$

*Under the following assumptions*

*(i) Conditional variance growth: there exists $\sigma^2(\mathcal{F}_0) \in (0, \infty)$ a.s. such that*

$$\frac{B_n^2(\mathcal{F}_0)}{n} \xrightarrow[n \to \infty]{a.s.} \sigma^2(\mathcal{F}_0).$$

*(ii) Conditional Haeusler convergence to 0:*

$$L_n(\mathcal{F}_0) := \frac{1}{B_n^3(\mathcal{F}_0)} \sum_{k=1}^n \mathbb{E}\big[|X_k|^3 \mid \mathcal{F}_0\big] \to 0 \quad a.s.,$$

$$N_n(\mathcal{F}_0) := \frac{1}{B_n^3(\mathcal{F}_0)} \mathbb{E}\big[\,|\langle M \rangle_n - B_n^2(\mathcal{F}_0)|^{3/2} \,\big|\, \mathcal{F}_0\big] \to 0 \quad a.s.$$

*(iii) (Predictable remainder) $\frac{R_n}{\sqrt{n}} \xrightarrow[n \to \infty]{} 0.$*

*We have (conditional LLN with rate in $\mathbb{L}^1$)*

$$\mathbb{E}\left[\left|\frac{1}{n} \sum_{k=1}^n Y_k - \mu\right| \,\middle|\, \mathcal{F}_0\right] \leq \frac{\sqrt{\sigma^2(\mathcal{F}_0)}}{\sqrt{n}} + \frac{C}{n} \qquad a.s.,$$

*for a constant $C$ depending only on $\sup_k \mathbb{E}|Z_k|$. In particular, $\frac{1}{n} \sum_{k=1}^n Y_k \to \mu$ in $\mathbb{L}^1$ (hence in probability) given $\mathcal{F}_0$, with leading rate $O_{a.s.}(n^{-1/2})$.*

*Proof sketch.* Write $\bar{Y}_n - \mu = \frac{M_n}{n} + \frac{R_n}{n}$. Conditionally on $\mathcal{F}_0$,

$$\mathbb{E}\left[\left|\frac{M_n}{n}\right| \,\middle|\, \mathcal{F}_0\right] \leq \frac{1}{n} \left(\mathbb{E}[M_n^2 \mid \mathcal{F}_0]\right)^{1/2} = \frac{B_n(\mathcal{F}_0)}{n} \sim \frac{\sqrt{\sigma^2(\mathcal{F}_0)}}{\sqrt{n}},$$

giving the $1/\sqrt{n}$ term. The coboundary gives $\mathbb{E}(|R_n| \mid \mathcal{F}_0) = O_{\text{a.s.}}(1)$, hence $\mathbb{E}(|R_n|/n \mid \mathcal{F}_0) = O_{\text{a.s.}}(1/n)$, which proves (A).

For (B), we apply the conditional Haeusler bound to $M_n/B_n(\mathcal{F}_0)$ (theorem G.1) and the Stein inequality for bounded $C^1$ test functions to get

$$\left|\mathbb{E}\big[F(M_n/B_n(\mathcal{F}_0)) \mid \mathcal{F}_0\big] - \mathbb{E}[F(G)]\right| \leq C(1 + \|F'\|_\infty)\,(L_n(\mathcal{F}_0) + N_n(\mathcal{F}_0)) = O_{\text{a.s.}}(n^{-1/2}).$$

Finally, replace $M_n/B_n(\mathcal{F}_0)$ by $\sqrt{n}\,(\bar{Y}_n - \mu)/\sqrt{\sigma^2(\mathcal{F}_0)}$: since $B_n(\mathcal{F}_0)/\sqrt{n} \to \sqrt{\sigma^2(\mathcal{F}_0)}$ a.s. by (i) and $R_n/\sqrt{n} \to 0$ in $L^1$ (from (iii)), this perturbation is $o_{\text{a.s.}}(1)$ in the test function bound. $\qquad \square$

# I  SUFFICIENT STATISTICS FOR CHANGE IN VALUE ESTIMATES

In Section F we describe the expression for the change in the value estimates over gradient steps. We state the following lemma summarizing the expression for the change over the gradient step $\eta = O(1/\sqrt{n})$ in the following lemma.

**Lemma I.1.** *The change in the value estimate in a single step of gradient update* $\Delta \tilde{s}_{t,\tau}$ *is as follows:*

$$
\Delta v_{t,\tau} = \Delta \tilde{s}_{t,\tau} \left[ \frac{1}{\sqrt{n}} \sum_{\kappa=1}^{n} B_\kappa^0 U_{\kappa,1}^0 \left( \varphi''(U_\kappa^0 \cdot [\tilde{s}_{t,\tau}, t]) - \varphi''(U_\kappa^0 . [\tilde{s}_{t,\tau}, t]) \left( U_{\kappa,1}^0 \tilde{s}_{t,\tau} + U_{\kappa,2}^0 t \right) \right) \right]
$$
$$
+ \frac{\eta}{\sqrt{n}} \sum_{\kappa=1}^{n} B_\kappa^0 \left( \varphi'(U_\kappa^0 \cdot [\tilde{s}_{t,\tau}, t])[\tilde{s}_{t,\tau}, t]^\top \widehat{\mathbb{G}}(U^\tau, W^\tau)_\kappa \right) + O(\eta^2),
$$
(29)

*and its distribution, conditioned on* $\tilde{s}_{t,\tau}, v_{t,\tau}, v'_{t,\tau}, a_{t,\tau},$ *is a Gaussian with mean:*

$$
\int_0^T e^{-\beta l} \mathbb{E} \left[ B^2 \varphi'(U \cdot [\tilde{s}_{t,\tau}, t]) \varphi'(U \cdot [\tilde{s}_{l,\tau}, l])(\tilde{s}_{l,\tau} \tilde{s}_{t,\tau} + lt) q_{l,\tau} \right] dl, \text{ where}
$$
$$
q_{l,\tau} = \partial_t v_{l,\tau} + r(\tilde{s}_{l,\tau}) - \beta v_{l,\tau} \text{ and } U = [U_1, U_2] \sim \mathcal{N}(0,1), B \sim Unif(-1,1),
$$

*multiplied by* $\eta$ *and variance:*

$$
\mathbb{E} \left[ B^2 U_1^2 \left( \varphi''(U \cdot [\tilde{s}_{t,\tau}, t]) - \varphi''(U.[\tilde{s}_{t,\tau}, t]) \left( U_1 \tilde{s}_{t,\tau} + U_2 t \right) \right)^2 \right]
$$
$$
\text{where } U = [U_1, U_2] \sim \mathcal{N}(0,1), B \sim Unif(-1,1),
$$

*multiplied by* $(\Delta \tilde{s}_{t,\tau})^2$ *up to an error of* $O(1/\sqrt{n})$.

*Proof.* The derivation of the expression in equation 29 is provided in Section F and we obtain it here subsuming all the terms that are of order $O(\eta^2)$. The argument for the Gaussian variance, under the conditioned variables described above, is derived from Lemma G.2.

To prove the mean converging at rate $O(1/\sqrt{n})$, we use Theorem H.1. The expression above in Equation 29 can be re-written as:

$$
\frac{\eta}{\sqrt{n}} \sum_{\kappa=1}^{n} B_\kappa^0 \left( \varphi'(U_\kappa^0 \cdot [\tilde{s}_{t,\tau}, t])[\tilde{s}_{t,\tau}, t]^\top \widehat{\mathbb{G}}(U^\tau, W^\tau)_\kappa \right)
$$
$$
= \eta \int_0^T e^{-\beta l} \frac{1}{n} \sum_{\kappa=1}^{n} (B_\kappa^0)^2 \varphi'(U_\kappa^0 \cdot [\tilde{s}_{t,\tau}, t]) \varphi'(U_\kappa^0 \cdot [\tilde{s}_{l,\tau}, l])(\tilde{s}_{l,\tau} \tilde{s}_{t,\tau} + lt) q_{l,\tau} dl.
$$
(30)

Therefore, in the notation of Theorem H.1 and the leave one out notation of Lemma G.2 we write:

$$
Y_\kappa = \int_0^T e^{-\beta l} (B_\kappa^0)^2 \varphi'(U_\kappa^0 \cdot [\bar{R}_{t,\tau,-\kappa}, t]) \varphi'(U_\kappa^0 \cdot [\bar{R}_{l,\tau,-\kappa}, l])(\bar{R}_{l,\tau,-\kappa} \bar{R}_{t,\tau,-\kappa} + lt) q_{l,\tau}.
$$

We propose $\mu$ (for LLN) as above and therefore the assumptions hold as follows:

1. The growth of conditional variance assumption holds because the squared difference between $\mu$ and $Y_\kappa$ is bounded $|Y_\kappa - \mu| = O(1/\sqrt{n})$ and therefore their sum increased to power 2 divided by n is $O(1)$.

2. Conditional Haeusler convergence to 0: once again since $|Y_\kappa - \mu| = O(1/\sqrt{n})$ and $|\mathbb{E}[Y_\kappa | \mathcal{F}_{\kappa-1}] - \mu = O(1/\sqrt{n})$ and therefore $|Y_\kappa - \mu|^3 = O(n^{-3/2})$ we have that:

$$
\frac{\sum_{\kappa=1}^{n} |Y_\kappa - \mathbb{E}[Y_\kappa | \mathcal{F}_{\kappa-1}]|^3}{O(1)} \xrightarrow{0}, \text{a.s.}
$$

and also

$$
\frac{\mathbb{E}\left[ |\langle M \rangle_n - B_n^2(\mathcal{F}_0)|^{3/2} \mid \mathcal{F}_0 \right]}{O(1)} \to 0.
$$

3. Let

$$\mu(s_{t,\tau}) = \int_0^T e^{-\beta l} \mathbb{E}\left[ B^2 \varphi'(U \cdot [\tilde{s}_{t,\tau}, t]) \varphi'(U \cdot [\tilde{s}_{l,\tau}, l]) (\tilde{s}_{l,\tau} \tilde{s}_{t,\tau} + lt) q_{l,\tau} \right] dl$$

Since all $Y_\kappa - \mu(s_{t,\tau})$ has a leading $(B_\kappa^0)^3$ leading product term and is of order $O(1/\sqrt{n})$ we know that it is centered around zero and therefore, $R_n/\sqrt{n} \to 0$.

Therefore, we have that expression in Equation 30 converges to $\mu(s_{t,\tau})$ at the rate $O(1/\sqrt{n})$ in its cdf. $\qquad\square$

We provide a similar lemma for the variable $v'_{t,\tau} = \partial_t v_{t,\tau}$.

**Lemma I.2.** *The change in the value estimate in a single step of gradient update $\Delta \tilde{s}_{t,\tau}$ is as follows:*

$$\Delta \partial_t v_{t,\tau} = \Delta \tilde{s}_{t,\tau} \left[ \frac{1}{\sqrt{n}} \sum_{\kappa=1}^n B_\kappa^0 U_{\kappa,1}^0 U_{\kappa,2}^0 \left( \varphi'''(U_\kappa^0 \cdot [\tilde{s}_{t,\tau}, t]) - \varphi'''(U_\kappa^0 . [\tilde{s}_{t,\tau}, t]) \left( U_{\kappa,1}^0 \tilde{s}_{t,\tau} + U_{\kappa,2}^0 t \right) \right) \right]$$
$$+ \frac{\eta}{\sqrt{n}} \sum_{\kappa=1}^n B_\kappa^0 U_{\kappa,2}^0 \left( \varphi''(U_\kappa^0 \cdot [\tilde{s}_{t,\tau}, t]) [\tilde{s}_{t,\tau}, t]^\top \widehat{\mathbb{G}}(U^\tau, W^\tau)_\kappa \right) + O(\eta^2),$$

(31)

*and its distribution, conditioned on $\tilde{s}_{t,\tau}, v_{t,\tau}, v'_{t,\tau}, a_{t,\tau}$ and $\Delta \tilde{s}_{t,\tau}$, is a Gaussian with mean:*

$$\int_0^T e^{-\beta l} \mathbb{E}\left[ B^2 U_2 \varphi''(U \cdot [\tilde{s}_{t,\tau}, t]) \varphi''(U \cdot [\tilde{s}_{l,\tau}, l]) (\tilde{s}_{l,\tau} \tilde{s}_{t,\tau} + lt) q_{l,\tau} \right] dl, \text{ where}$$
$$q_{l,\tau} = \partial_t v_{l,\tau}^{lin} + r(\tilde{s}_{l,\tau}) - \beta v_{l,\tau}^{lin} \text{ and } U = [U_1, U_2] \sim \mathcal{N}(0,1), B \sim \text{Unif}(-1,1),$$

*multiplied by $\eta$ and variance:*

$$\mathbb{E}\left[ B^2 U_1^2 U_2^2 \left( \varphi'''(U \cdot [\tilde{s}_{l,\tau}, t]) - \varphi'''(U . [\tilde{s}_{t,\tau}, t]) (U_1 \tilde{s}_{t,\tau} + U_2 t) \right)^2 \right]$$
$$\text{where } U = [U_1, U_2] \sim \mathcal{N}(0,1), B \sim \text{Unif}(-1,1),$$

*multiplied by $(\Delta \tilde{s}_{t,\tau})^2$ with an additional error of $O(1/\sqrt{n})$.*

*Proof.* The proof is based on taking a partial derivative with respect to $t$ in Equation 29 and the rest follows from from the conditional law of large numbers and CLT. $\qquad\square$

## J    CHANGE IN ACTION OVER GRADIENT STEP

We define by $a_{t,\tau}$ the action chosen by the agent at time $t$ and the gradient step $\tau$. Formally, it is defined as $a_{t,\tau} = F^{\text{lin}}(\tilde{s}_{t,\tau}; W^\tau) = F_\pi(\tilde{s}_{t,\tau}; W^0) + \Phi(\tilde{s}_{t,\tau}; W^0) \Delta W^\tau$. Now, similar to the value

estimate in Section F, consider the change in this variable:

$$
\begin{aligned}
a_{t,\tau+\eta} - a_{t,\tau} =& F_\pi(s_{t,\tau+\eta}; W^0) + \Phi(s_{t,\tau+\eta}; W^0)(W^{\tau+\eta} - W^0) \\
& - F_\pi(\tilde{s}_{t,\tau}; W^0) + \Phi(\tilde{s}_{t,\tau}; W^0)(W^\tau - W^0) \\
=& \frac{1}{\sqrt{n}} \sum_{\kappa=1}^n C_\kappa^0 \left( \varphi(W_\kappa^0 s_{t,\tau+\eta}) - \varphi(W_\kappa^0 \tilde{s}_{t,\tau}) \right) \\
& + \left( \Phi(s_{t,\tau+\eta}; W^0)W^{\tau+\eta} - \Phi(s_{t,\tau}; W^0)W^\tau \right) - \left( \Phi(s_{t,\tau+\eta}; W^0) - \Phi(s_{t,\tau}; W^0) \right) W^0 \\
=& \frac{1}{\sqrt{n}} \sum_{\kappa=1}^n C_\kappa^0 \left( \varphi'(W_\kappa^0 \tilde{s}_{t,\tau})(s_{t,\tau+\eta} - \tilde{s}_{t,\tau})W_\kappa^0 \right. \\
& \left. + \frac{1}{2}\varphi''(W_\kappa^0 \tilde{s}_{t,\tau}) \left( (s_{t,\tau+\eta} - \tilde{s}_{t,\tau})W_\kappa^0 \right)^2 \right) + O(\eta^2) \\
& + \frac{1}{\sqrt{n}} \sum_{\kappa=1}^n C_\kappa^0 \left( \varphi'(W_\kappa^0 s_{t,\tau+\eta})s_{t,\tau+\eta}W_\kappa^{\tau+\eta} - \varphi'(W_\kappa^0 s_{\tau,t})\tilde{s}_{t,\tau}W_\kappa^\tau \right) \\
& - \frac{1}{\sqrt{n}} \sum_{\kappa=1}^n C_\kappa^0 \left( \varphi'(W_\kappa^0 s_{t,\tau+\eta})s_{t,\tau+\eta} - \varphi'(W_\kappa^0 s_{\tau,t})\tilde{s}_{t,\tau} \right) W_\kappa^0.
\end{aligned}
$$

Similar to the previous section, consider the first expression in the summation above: $W_\kappa^0 C_\kappa^0 \varphi'(W_\kappa^0 \tilde{s}_{t,\tau})$. To evaluate the sum of these variables in infinite width limit we have to separate the dependence of $\tilde{s}_{t,\tau}$ on $W_\kappa^0, C_\kappa^0$, we observe the following about $s_{t,0}$ and $\kappa = n$:

$$
\begin{aligned}
s_{t,0} =& \sum_{i=1}^\infty \sum_{\alpha \in \Xi_i} \left( \mathbb{B}_\alpha + \sum_{\mathbf{c} \in \mathcal{C}_i, \mathbf{m} \in \mathcal{M}_i} \mathbb{C}_{\mathbf{c},\mathbf{m},\alpha} \Pi_{j=1}^i \left( F_\pi^{(c_j)}(s_0; W^0) \right)^{m_j} \right) I_\alpha[1]_{0,t} \\
=& \sum_{i=1}^\infty \sum_{\alpha \in \Xi_i} \left( \mathbb{B}_\alpha + \sum_{\mathbf{c} \in \mathcal{C}_i, \mathbf{m} \in \mathcal{M}_i} \mathbb{C}_{\mathbf{c},\mathbf{m},\alpha} \Pi_{j=1}^i \left( \frac{1}{\sqrt{n}} \sum_{\kappa=1}^n C_\kappa^0 \varphi^{(c_j)}(s_0 W_\kappa^0) \right)^{m_j} \right) I_\alpha[1]_{0,t},
\end{aligned}
$$

where we can factor out the expression for $\kappa = n$ as follows:

$$
\begin{aligned}
s_{t,0} =& \sum_{i=1}^\infty \sum_{\alpha \in \Xi_i} \left( \mathbb{B}_\alpha + \sum_{\mathbf{c} \in \mathcal{C}_i, \mathbf{m} \in \mathcal{M}_i} \mathbb{C}_{\mathbf{c},\mathbf{m},\alpha} \Pi_{j=1}^i \left( \frac{1}{\sqrt{n}} \sum_{\kappa=1}^{n-1} C_\kappa^0 \varphi^{(c_j)}(s_0 W_\kappa^0) \right)^{m_j} \right. \\
& \left. + \sum_{\mathbf{c} \in \mathcal{C}_i, \mathbf{m} \in \mathcal{M}_i} \mathbb{C}_{\mathbf{c},\mathbf{m},\alpha} \sum_{j=1}^i \frac{1}{\sqrt{n}} C_n^0 \varphi^{(c_j)}(s_0 W_n^0) K_{-j,-n} + O(n^{-2}) \right) I_\alpha[1]_{0,t},
\end{aligned}
$$

where similar to the previous sections we have the following notation:

$$
K_{-j,-n} = m_j \Pi_{\substack{j'=1 \\ j' \neq j}}^i \left( \frac{1}{\sqrt{n}} \sum_{\kappa=1}^{n-1} C_\kappa^0 \varphi^{(c_j)}(s_0 W_\kappa^0) \right)^{m_j},
$$

and the derivative of $c_j$ order of $\varphi$ is:

$$
\varphi^{(c_j)}(s_0 W_n^0) = (W_n^0)^{c_j} \left. \frac{\partial^{c_j} \varphi(x)}{x} \right|_{x=s_0 W_n^0}.
$$

We further denote by $R_{t,0,-n}$:

$$
R_{t,0,-n} = \sum_{i=1}^\infty \sum_{\alpha \in \Xi_i} \left( \mathbb{B}_\alpha + \sum_{\mathbf{c} \in \mathcal{C}_i, \mathbf{m} \in \mathcal{M}_i} \mathbb{C}_{\mathbf{c},\mathbf{m},\alpha} \Pi_{j=1}^i \left( \frac{1}{\sqrt{n}} \sum_{\kappa=1}^{n-1} C_\kappa^0 \varphi^{(c_j)}(s_0 W_\kappa^0) \right)^{m_j} \right) I_\alpha[1]_{0,t}
$$

Using this leave-one-out formulation, we further rewrite the summation as follows.

$$\frac{1}{\sqrt{n}}\sum_{\kappa=1}^{n}C_{\kappa}^{0}W_{\kappa}^{0}\varphi'(W_{\kappa}^{0}s_{t,0}) = \frac{1}{\sqrt{n}}\sum_{\kappa=1}^{n}C_{\kappa}^{0}W_{\kappa}^{0}\varphi'(W_{\kappa}^{0}R_{t,0,-n})$$

$$+\sum_{i=1}^{\infty}\sum_{\alpha\in\Xi_{i}}\sum_{\mathbf{c}\in\mathcal{C}_{i},\mathbf{m}\in\mathcal{M}_{i}}\mathbb{C}_{\mathbf{c},\mathbf{m},\alpha}\sum_{j=1}^{i}\frac{1}{n}\sum_{\kappa=1}^{n}(C_{\kappa}^{0})^{2}\varphi^{(c_{j})}(s_{0}W_{\kappa}^{0})W_{\kappa}^{0}\varphi''(W_{\kappa}^{0}R_{t,0,-n})K_{-j,-\kappa}.$$
(32)

Using the conditional law of large numbers above and the fact that $\varphi^{(c_{j})}(s_{0}W_{n}^{0})$ is odd (for an odd function $\varphi = \tanh$), and together with $W_{\kappa}^{0}\varphi''(W_{\kappa}^{0}R_{t,0,-n})$ which is symmetric in $W_{\kappa}^{0}$, we have that:

$$\frac{1}{n}\sum_{\kappa=1}^{n}(C_{\kappa}^{0})^{2}\varphi^{(c_{j})}(s_{0}W_{\kappa}^{0})W_{\kappa}^{0}\varphi''(W_{\kappa}^{0}R_{t,0,-n})K_{-j,-\kappa} = O\left(\frac{1}{\sqrt{n}}\right),$$

since the sequence inside the summation has mean 0 (by law of large numbers). Also, we note that, given the result that the Itô -Taylor expansion converges, we have:

$$\sum_{i=1}^{\infty}\sum_{\alpha\in\Xi_{i}}\sum_{\mathbf{c}\in\mathcal{C}_{i},\mathbf{m}\in\mathcal{M}_{i}}\mathbb{C}_{\mathbf{c},\mathbf{m},\alpha}\sum_{j=1}^{i}(C_{\kappa}^{0})^{2}\varphi^{(c_{j})}(s_{0}W_{\kappa}^{0})W_{\kappa}^{0}\varphi''(W_{\kappa}^{0}R_{t,0,-n})K_{-j,-\kappa} = O(1).$$

Once again, similar to Lemma 3.1 we can show that the expression in Equation 32 is distributed as Gaussian with mean 0 and variance $\mathbb{E}[C^{2}W^{2}\varphi(Ws)]$ with an additional error term of order $O(1/\sqrt{n})$, conditioned on $s_{t,0} = s$.

Now to show the inductive step, suppose that for a $\tau$ that is an integral multiple of $\eta$ we can express the general expression of Equation 32 as the sum of a Gaussian plus and an error term of order $O(1/\sqrt{n})$. Now we expand it recursively:

$$\frac{1}{\sqrt{n}}\sum_{\kappa=1}^{n}C_{\kappa}^{0}\varphi'(W_{\kappa}^{0}\tilde{s}_{t,\tau}) = \frac{1}{\sqrt{n}}\sum_{\kappa=1}^{n}C_{\kappa}^{0}\varphi'(W_{\kappa}^{0}R_{t,\tau,-\kappa})$$

$$+\sum_{i=1}^{\infty}\sum_{\alpha\in\Xi_{i}}\sum_{\mathbf{c}\in\mathcal{C}_{i},\mathbf{m}\in\mathcal{M}_{i}}\mathbb{C}_{\mathbf{c},\mathbf{m},\alpha}\sum_{j=1}^{i}\frac{1}{n}\sum_{\kappa=1}^{n}(C_{\kappa}^{0})^{2}\varphi^{(c_{j})}(s_{0}W_{\kappa}^{0})W_{\kappa}^{0}\varphi''(W_{\kappa}^{0}R_{t,0,-n})K_{-j,-\kappa}.$$
(33)

To isolate the dependence on $W_{\kappa}^{0}, C_{\kappa}^{0}$ in $K_{-j,-\kappa}$ we expand the expression for $\kappa = n$ as follows:

$$K_{-j,-n} = \Pi_{j=1}^{i}\left(\frac{1}{\sqrt{n}}\sum_{\kappa=1}^{n-1}C_{\kappa}^{0}\varphi^{(c_{j})}(\tilde{s}_{t,\tau}W_{\kappa}^{0}) + \Phi^{(c_{j})}(s_{0};W^{0})W^{\tau}\right)^{m_{j}},$$

$$= \Pi_{j=1}^{i}\left(\frac{1}{\sqrt{n}}\sum_{\kappa=1}^{n-1}C_{\kappa}^{0}\varphi^{(c_{j})}(\tilde{s}_{t,\tau}W_{\kappa}^{0}) + \frac{1}{n}\sum_{\kappa=1}^{n}\phi^{(c_{j})}(s_{0};W_{\kappa}^{0},B_{n}^{0})W_{\kappa}^{\tau}\right)^{m_{j}}$$

$$= K_{-j,-n,-n} + \frac{1}{n}\sum_{j=1}^{i}m_{j}\phi^{(c_{j})}(s_{0};W_{n}^{0},B_{n}^{0})K_{-j,-n}.$$

Therefore, the additional dependence on $W_{n}^{0}, B_{n}^{0}$ in equation 33 is $O(1/n^{2})$. Therefore, we have shown that for a general $\tau$ the inductive argument is valid.

Continuing the decomposition of the expressions in $\Delta a_{t,\tau}$ above, we observe:

$$\frac{1}{\sqrt{n}}\sum_{\kappa=1}^{n}C_{\kappa}^{0}\left(\varphi'(W_{\kappa}^{0}s_{t,\tau+\eta})s_{t,\tau+\eta}W_{\kappa}^{\tau+\eta} - \varphi'(W_{\kappa}^{0}s_{\tau,t})\tilde{s}_{t,\tau}W_{\kappa}^{\tau}\right)$$

$$= \Delta\tilde{s}_{t,\tau}\frac{1}{\sqrt{n}}\sum_{\kappa=1}^{n}C_{\kappa}^{0}\left(\varphi''(W_{\kappa}^{0}\tilde{s}_{t,\tau})W_{\kappa}^{0}\right)$$

$$+\frac{\eta}{\sqrt{n}}\sum_{\kappa=1}^{n}C_{\kappa}^{0}\varphi'(W_{\kappa}^{0}\tilde{s}_{t,\tau})\tilde{s}_{t,\tau}\widehat{\mathcal{G}}(U,W)(U^{\tau},W^{\tau})_{\kappa} + O(\eta^{2}).$$

Further expanding the expression for $\Delta a_{t,\tau}$ we have the following:

$$\frac{1}{\sqrt{n}} \sum_{\kappa=1}^{n} C_\kappa^0 \left( \varphi'(W_\kappa^0 s_{t,\tau+\eta}) s_{t,\tau+\eta} - \varphi'(W_\kappa^0 s_{\tau,t}) \tilde{s}_{t,\tau} \right) W_\kappa^0$$

$$= \frac{1}{\sqrt{n}} \sum_{\kappa=1}^{n} B_\kappa^0 \left( \varphi'(W_\kappa^0 \tilde{s}_{t,\tau}) \Delta \tilde{s}_{t,\tau} + \varphi''(W_\kappa^0 \tilde{s}_{t,\tau}) \Delta \tilde{s}_{t,\tau} W_\kappa^0 \tilde{s}_{t,\tau} \right) W_\kappa^0 + O(\eta^2).$$

We also express the mean term which includes the gradient vector as follows:

$$\frac{\eta}{\sqrt{n}} \sum_{\kappa=1}^{n} C_\kappa^0 \varphi'(W_\kappa^0 \tilde{s}_{t,\tau}) \tilde{s}_{t,\tau} \widehat{\mathcal{G}}(U, W)(U^\tau, W^\tau)_\kappa + O(\eta^2)$$

$$= \eta \int_0^T \frac{1}{n} \sum_{\kappa=1}^{n} (C_\kappa^0)^2 \varphi'(W_\kappa^0 \tilde{s}_{t,\tau}) \varphi'(W_\kappa^0 \tilde{s}_{l,\tau}) (\tilde{s}_{l,\tau} \tilde{s}_{t,\tau}) q_{l,\tau} dl.$$

## K    SUFFICIENT STATISTICS FOR CHANGE IN ACTION

In Section J we derive the change in the action variable: $a_{t,\tau}$ over the gradient step. Here we provide a lemma, analogous to Lemma I.1 but for $a_{t,\tau}$, summarizing the sufficient statistics required to track the change in the action variable.

**Lemma K.1.** *The change in the action variable in a single step of gradient update $\Delta a_{t,\tau}$ is as follows:*

$$\Delta a_{t,\tau} = \Delta \tilde{s}_{t,\tau} \frac{1}{\sqrt{n}} \sum_{\kappa=1}^{n} C_\kappa^0 W_\kappa^0 \left( \varphi''(W_\kappa^0 \tilde{s}_{t,\tau}) - \varphi''(W_\kappa^0 \tilde{s}_{t,\tau}) W_\kappa^0 \tilde{s}_{t,\tau} \right)$$

$$+ \eta \int_0^T \frac{1}{n} \sum_{\kappa=1}^{n} (C_\kappa^0)^2 \varphi'(W_\kappa^0 \tilde{s}_{t,\tau}) \varphi'(W_\kappa^0 \tilde{s}_{l,\tau}) (\tilde{s}_{l,\tau} \tilde{s}_{t,\tau}) q_{l,\tau} dl + O(\eta^2) \tag{34}$$

*and its distribution, conditioned on $\tilde{s}_{t,\tau}, v_{t,\tau}, v'_{t,\tau}, a_{t,\tau}$ and $\Delta \tilde{s}_{t,\tau}$, is a Gaussian with mean:*

$$\int_0^T e^{-\beta l} \mathbb{E} \left[ C^2 \varphi'(W \tilde{s}_{t,\tau}) \varphi'(W \tilde{s}_{l,\tau}) \tilde{s}_{l,\tau} \tilde{s}_{t,\tau} q_{l,\tau} \right] dl, \text{ where}$$

$$q_{l,\tau} = \partial_t v_{l,\tau}^{lin} + r(\tilde{s}_{l,\tau}) - \beta v_{l,\tau}^{lin} \text{ and } W \sim \mathcal{N}(0,1), C \sim Unif(-1,1),$$

*multiplied by $\eta$ and variance:*

$$\mathbb{E} \left[ C^2 W^2 \left( \varphi''(W \tilde{s}_{t,\tau}) - \varphi''(W \tilde{s}_{t,\tau}) W \tilde{s}_{t,\tau} \right)^2 \right]$$

$$\text{where } W \sim \mathcal{N}(0,1), C \sim Unif(-1,1),$$

*multiplied by $(\Delta \tilde{s}_{t,\tau})^2$ up to an error of $O(1/\sqrt{n})$.*

*Proof.* The result and proof are similar to that of Lemma I.1. The proof of Gaussian variance follows from the derivation and demonstration of the inductive step in Section J (see the discussion around Equation 32) and the combination of the conditional law of large numbers (Theorem H.1) and the conditional CLT (Theorem G.1). The proof of the Gaussian mean originates from the conditional law of large numbers (Theorem H.1). $\qquad\square$

A corollary of this lemma for $\partial_x F_\pi^{\text{lin}}$, which we utilize to estimate the change in the state variable, is also presented below. We denote by $a'_{t,\tau} = \partial_x F_\pi^{\text{lin}}$.

**Lemma K.2.** *The change in the action variable in a single step of gradient update $\Delta a_{t,\tau}$ is as follows:*

$$\Delta a'_{t,\tau} = \Delta \tilde{s}_{t,\tau} \frac{1}{\sqrt{n}} \sum_{\kappa=1}^{n} C_\kappa^0 (W_\kappa^0)^2 \left( \varphi'''(W_\kappa^0 \tilde{s}_{t,\tau}) - \varphi'''(W_\kappa^0 \tilde{s}_{t,\tau}) W_\kappa^0 \tilde{s}_{t,\tau} \right)$$

$$+ \eta \int_0^T \frac{1}{n} \sum_{\kappa=1}^{n} (C_\kappa^0)^2 W_\kappa^0 \varphi''(W_\kappa^0 \tilde{s}_{t,\tau}) \varphi'(W_\kappa^0 \tilde{s}_{l,\tau}) (\tilde{s}_{l,\tau} \tilde{s}_{t,\tau}) q_{l,\tau} dl + O(\eta^2) \tag{35}$$

*and its distribution, conditioned on $\tilde{s}_{t,\tau}, v_{t,\tau}, v'_{t,\tau}, a_{t,\tau}$ and $\Delta\tilde{s}_{t,\tau}$, is a Gaussian with mean:*

$$\int_0^T e^{-\beta l} \mathbb{E}\left[C^2 W \varphi''(W\tilde{s}_{t,\tau}) \varphi'(W\tilde{s}_{l,\tau}) \tilde{s}_{l,\tau} \tilde{s}_{t,\tau} q_{l,\tau}\right] dl, \text{ where}$$

$$q_{l,\tau} = \partial_t v_{l,\tau}^{lin} + r(\tilde{s}_{l,\tau}) - \beta v_{l,\tau}^{lin} \text{ and } W \sim \mathcal{N}(0,1), C \sim Unif(-1,1),$$

*multiplied by $\eta$ and variance:*

$$\mathbb{E}\left[C^2 W^4 \left(\varphi'''(W\tilde{s}_{t,\tau}) - \varphi'''(W\tilde{s}_{t,\tau})W\tilde{s}_{t,\tau}\right)^2\right]$$
$$\text{and } W \sim \mathcal{N}(0,1), C \sim Unif(-1,1),$$

*multiplied by $(\Delta\tilde{s}_{t,\tau})^2$ up to an error of $O(1/\sqrt{n})$.*

*Proof.* The proof proceeds by taking the partial with respect to $s$ in equation 34 and then a similar application of Theorems G.1 and H.1. $\qquad\square$

## L    CHANGE IN STATE OVER GRADIENT STEP

We analyze and better understand how the state variable changes over gradient steps. Similarly to Section F and following the notation in Section F.1, we consider the difference for the learning rate $\eta$, i.e. $\Delta\tilde{s}_{t,\tau} = s_{t,\tau+\eta} - \tilde{s}_{t,\tau}$ and present the following Lemma.

**Lemma L.1.** *Define $Z_{l,t,\tau}$ as:*

$$Z_{t,l,\tau} = Y_{t,\tau} \int_0^t Y_{u,\tau}^{-1} h(\tilde{s}_{u,\tau}) \mathcal{C}_{u,l,\tau} du, \text{ with } Y_{t,\tau} \text{ is solution to}$$
$$dY_{t,\tau} = (a_{t,\tau} + a'_{t,\tau})Y_{t,\tau} dt + \sigma'(\tilde{s}_{t,\tau}) Y_{t,\tau} dw_t$$
$$\mathcal{C}_{u,l,\tau} = \mathbb{E}\left[C^2 \varphi'(\tilde{s}_{l,\tau}W)\varphi'(\tilde{s}_{t,\tau}W)\right], \text{ with } C \sim Unif(-1,1), W \sim \mathcal{N}(0,1).$$

*In addition, define $\mathbb{Z}_{t,\tau} = \int_0^t Z_{t,l,\tau}(v'_{l,\tau} + r(\tilde{s}_{l,\tau}) - \beta v'_{l,\tau})dl$. The change in the state variable, $\Delta\tilde{s}_{t,\tau}$, conditioned on $v_{t,\tau}, a_{t,\tau}, a'_{t,\tau}, \partial_t v_{t,\tau}$, is as follows:*

$$\Delta\tilde{s}_{t,\tau} = \eta\mathbb{Z}_{t,\tau} - M_{t,\tau} + G_{t,\tau} + O(1/n), \text{ where}$$
$$M_{t,\tau} = \tilde{s}_{t,\tau} - s_0 - \int_0^t (g(\tilde{s}_{u,\tau}) + h(\tilde{s}_{u,\tau})a_{u,\tau})du,$$

*and $G_{t,\tau}$ is a random variable and the martingale component of $x_{t,\tau}$, following a similar dynamics to $\tilde{s}_{t,\tau}$:*

$$dx_{t,\tau} = (g(x_{t,\tau} + h(x_{t,\tau})a_{t,\tau})dt + \tilde{\sigma}(x_{t,\tau})dw'_t,$$

*where $w'_t$ is an independent Wiener process and therefore $Z_{t,\tau} = x_{t,\tau} - \mathbb{E}[x_{t,\tau}]$, where the expectation is over the random process $w'_t$.*

*Proof.* Using the Itô -Taylor expansion of the state variable at time $t$ (Equation 20) and taking the difference for $\tau$ and $\tau + \eta$.

$$\Delta\tilde{s}_{t,\tau} = \sum_{i=1}^{\infty} \sum_{\alpha \in \Xi_i} \sum_{\mathbf{c} \in \mathcal{C}_i, \mathbf{m} \in \mathcal{M}_i} \mathbb{C}_{\mathbf{c},\mathbf{m},\alpha} \left( \Pi_{j=1}^i \left( F_\pi^{(c_j)}(s_0; W^0) + \eta\Phi^{(c_j)}(s_0; W^0)\Delta W^{\tau+\eta} \right)^{m_j} I_\alpha[1]_{0,t} \right.$$

$$\left. - \Pi_{j=1}^i \left( F_\pi^{(c_j)}(s_0; W^0) + \eta\Phi^{(c_j)}(s_0; W^0)\Delta W^\tau \right)^{m_j} I'_\alpha[1]_{0,t} \right)$$

$$+ \sum_{i=1}^{\infty} \sum_{\alpha \in \Xi_i} \mathbb{B}_\alpha(I_\alpha[1]_{0,t} - I'_\alpha[1]_{0,t})$$

Consider the deterministic parts of $\Delta \tilde{s}_{t,\tau}$, with respect to $I$ or the deterministic part of the transition where we have $I_\alpha[1]_{0,t} = I'_\alpha[1]_{0,t}$.

$$
\overline{\Delta s}_{t,\tau} = \sum_{i=1}^\infty \sum_{\alpha \in \Xi_i \cap \mathcal{B}_i} \sum_{\mathbf{c} \in \mathcal{C}_i, \mathbf{m} \in \mathcal{M}_i} \mathbb{C}_{\mathbf{c},\mathbf{m},\alpha} \left( \Pi_{j=1}^i \left( F_\pi^{(c_j)}(s_0; W^0) + \Phi^{(c_j)}(s_0; W^0)\Delta W^{\tau+\eta} \right)^{m_j} \right.
$$
$$
\left. - \Pi_{j=1}^i \left( F_\pi^{(c_j)}(s_0; W^0) + \Phi^{(c_j)}(s_0; W^0)\Delta W^{\tau} \right)^{m_j} \right) I'_\alpha[1]_{0,t}
$$
$$
= \sum_{i=1}^\infty \sum_{\alpha \in \Xi_i \cap \mathcal{B}_i} \sum_{\mathbf{c} \in \mathcal{C}_i, \mathbf{m} \in \mathcal{M}_i} \mathbb{C}_{\mathbf{c},\mathbf{m},\alpha} \left( \sum_{j=1}^i \eta m_j \Phi^{(c_j)}(s_0; W^0)\widehat{\mathcal{G}}(U^\tau, W^\tau)K_{-j} + O(\eta^2) \right) I'_\alpha[1]_{0,t},
$$

where $K_{-j} = \Pi_{\substack{j'=1 \\ j' \neq j}}^i \left( F_\pi^{(c_j)}(s_0; W^0) + \Phi^{(c_j)}(s_0; W^0)\Delta W^\tau \right)^{m_j}$. The stochastic part of $\Delta \tilde{s}_{t,\tau}$ is then written as:

$$
\widetilde{\Delta s}_{t,\tau} = \sum_{i=1}^\infty \sum_{\alpha \in \Xi_i \cap \Omega_i} \sum_{\mathbf{c} \in \mathcal{C}_i, \mathbf{m} \in \mathcal{M}_i} \mathbb{C}_{\mathbf{c},\mathbf{m},\alpha} \left( \quad \Pi_{j=1}^i \left( F_\pi^{(c_j)}(s_0; W^0) + \Phi^{(c_j)}(s_0; W^0)\Delta W^{\tau+\eta} \right)^{m_j} \right.
$$
$$
\left. - \Pi_{j=1}^i \left( F_\pi^{(c_j)}(s_0; W^0) + \Phi^{(c_j)}(s_0; W^0)\Delta W^{\tau} \right)^{m_j} \right) I'_\alpha[1]_{0,t}
$$
$$
+ \sum_{i=1}^\infty \sum_{\alpha \in \Xi_i \cap \Omega_i} \mathbb{B}_\alpha + \sum_{\mathbf{c} \in \mathcal{C}_i, \mathbf{m} \in \mathcal{M}_i} \mathbb{C}_{\mathbf{c},\mathbf{m},\alpha} \Pi_{j=1}^i \left( F_\pi^{(c_j)}(s_0; W^0) + \Phi^{(c_j)}(s_0; W^0)\Delta W^{\tau} \right)^{m_j} (I_\alpha[1]_{0,t} - I'_\alpha[1]_{0,t}).
$$

Simplifying these expressions as in the deterministic analog, we obtain the following.

$$
\Delta S_{t,\tau} = \sum_{i=1}^\infty \sum_{\alpha \in \Xi_i} \sum_{\mathbf{c} \in \mathcal{C}_i, \mathbf{m} \in \mathcal{M}_i} \mathbb{C}_{\mathbf{c},\mathbf{m},\alpha} \left( \sum_{j=1}^i \eta m_j \Phi^{(c_j)}(s_0; W^0)\widehat{\mathcal{G}}(U^\tau, W^\tau)K_{-j} + O(\eta^2) \right) I'_\alpha[1]_{0,t}
$$
$$
+ \sum_{i=1}^\infty \sum_{\alpha \in \Xi_i \cap \Omega_i} \mathbb{B}_\alpha + \sum_{\mathbf{c} \in \mathcal{C}_i, \mathbf{m} \in \mathcal{M}_i} \mathbb{C}_{\mathbf{c},\mathbf{m},\alpha} \Pi_{j=1}^i \left( F_\pi^{(c_j)}(s_0; W^0) + \Phi^{(c_j)}(s_0; W^0)\Delta W^{\tau} \right)^{m_j}
$$
$$
\times (I_\alpha[1]_{0,t} - I'_\alpha[1]_{0,t}).
$$
$$\tag{36}$$

We seek to simplify the expression which emerges in both of these terms:

$$
\Phi^{(c_j)}(s_0; W^0)\widehat{\mathcal{G}}(U^\tau, W^\tau) = \int_0^T e^{-\beta l} \Phi^{(c_j)}(s_0; W^0)\Phi(\tilde{s}_l^{\pi^W}; W^0)^\top \left[ \partial_t v_{l,\tau} + r(\tilde{s}_{l,\tau}) - \beta v_{l,\tau} \right] dl,
$$
$$
= \int_0^T \frac{1}{n} \left( \sum_{\kappa=1}^n C_\kappa^0 \tilde{s}_{l,\tau} \varphi'(W_\kappa^0 \tilde{s}_{l,\tau}) \phi^{(c_j)}(s_0, W_\kappa^0, C_\kappa^0) \right) dl
$$
$$
\times \int_0^T e^{-\beta l} \left[ \partial_t v_{l,\tau} + r(\tilde{s}_{l,\tau}) - \beta v_{l,\tau} \right] dl,
$$

where $\phi$ is as defined in equation 28:

$$
\phi^{(c_j)}(s_0, W_\kappa^0, C_\kappa^0) = C_\kappa^0 \left( (W_\kappa^0)^{c_j} \tilde{s}_0 \varphi^{(c_j+1)}(W_\kappa^0 \tilde{s}_0) + c_j (W_\kappa^0)^{c_j-1} \varphi^{(c_j)}(W_\kappa^0 \tilde{s}_0) \right).
$$

To simplify these expressions, we first provide a result for the polynomial expansion, using the Itô-Taylor series (Section C). Consider the differential of $s_t(W)$ with respect to $W$ and we define:

$$
\tilde{s}_{t,\tau} = \nabla_W s_t(W)|_{W=W_\tau},
$$

which can be defined as the solution to:

$$\mathrm{d}S_{t,\tau} = \left(A_{t,\tau}\, S_{t,\tau} + h(\tilde{s}_{t,\tau})\, \Phi(\tilde{s}_{t,\tau}; W^0)\right) dt + \sigma'(\tilde{s}_{t,\tau})\, S_{t,\tau}\, dw_t, \qquad S_0 = 0,$$

$$A_{t,\tau} = g'(\tilde{s}_{t,\tau}) + h'(\tilde{s}_{t,\tau})\big(F_\pi(\tilde{s}_{t,\tau}) + \Phi(\tilde{s}_{t,\tau}; W^0)\Delta W^\tau\big) + h(\tilde{s}_{t,\tau})\Big(F'_\pi(\tilde{s}_{t,\tau}) + \Phi'(\tilde{s}_{t,\tau}; W^0)\Delta W^\tau\Big).$$

Given that we have the entire path: $\tilde{s}_{t,\tau}$ and $a_{t,\tau}$ to condition upon, we can "reconstruct" the driving Brownian motion $w_t$ as follows.

$$dw_t = \frac{d\tilde{s}_{t,\tau} - g(\tilde{s}_{t,\tau}) + h(\tilde{s}_{t,\tau})a_{t,\tau}dt}{\tilde{\sigma}(\tilde{s}_{t,\tau})}$$

$$w_t = \int_0^t \frac{d\tilde{s}_{u,\tau} - g(\tilde{s}_{u,\tau}) + h(\tilde{s}_{u,\tau})a_{u,\tau}dt}{\tilde{\sigma}(\tilde{s}_{u,\tau})} du,$$

because we have a solution with strong uniqueness due to the Lipschitz assumption on the dynamics (see theorem 5.2.5 in the textbook by Karatzas & Shreve (2014)) and $\tilde{\sigma} \neq 0$. Now we further define $Z_{t,l,\tau} = S_{t,\tau}\Phi(s_l; W^0)^\top$ for some fixed $s$, and therefore the corresponding ODE as:

$$dZ_{t,l,\tau} = \left(A_{t,l,\tau}\, Z_{t,l,\tau} + h(\tilde{s}_{t,\tau})\, \Phi(\tilde{s}_{t,\tau}; W^0)\, \Phi(\tilde{s}_{l,\tau}; W^0)^\top\right) dt + \sigma'(\tilde{s}_{t,\tau})\, Z_{t,l,\tau}\, dw_t.$$

To solve for $Z_{t,l,\tau}$, we define $Y_{t,l,\tau}$ as the solution to the equation:

$$dY_{t,\tau} = A_{t,\tau}Y_{t,\tau}dt + \sigma'(\tilde{s}_{t,\tau})\, Y_{t,\tau}dw_t, \qquad Y_0 = 1,$$

$$Z_{t,l,\tau} = Y_{t,\tau} \int_0^t Y_{u,\tau}^{-1}\, h(\tilde{s}_{u,\tau})\, \Phi(\tilde{s}_{u,\tau}; W^0)\Phi(\tilde{s}_{l,\tau}; W^0)^\top du.$$

We also note that

$$S_t = \sum_{i=1}^\infty \sum_{\alpha \in \Xi_i} \sum_{\mathbf{c} \in \mathcal{C}_i, \mathbf{m} \in \mathcal{M}_i} \mathbb{C}_{\mathbf{c},\mathbf{m},\alpha} \left(\sum_{j=1}^i \eta m_j \Phi^{(c_j)}(s_0; W^0) K_{-j}\right) I'_\alpha[1]_{0,t},$$

which is the same as the expression in equation 36, barring the $O(\eta^2)$ error. For intuition of this gradient of the state variable with respect to the parameters see the example in Section D which corresponds to the first expression in equation. Therefore, to solve for $Y_{t,\tau}$ we have access to all $A_{t,\tau}, \tilde{s}_{t,\tau}, w_{t,\tau}$ are all known and $\Phi(\tilde{s}_{t,\tau})\,\Phi(\tilde{s}_{l,\tau})^\top$ depends only on $\tilde{s}_{l,\tau}$ and $\tilde{s}_{t,\tau}$ Therefore, we can now rewrite the change in state at gradient step $\tau$ as follows:

$$\Delta\tilde{s}_{t,\tau} = \eta \int_0^T Z_{t,l,\tau} \left(\partial_t v_{l,\tau} + r(\tilde{s}_{l,\tau}) - \beta v_{l,\tau}\right) dl + O(\eta^2)$$

$$+ \sum_{i=1}^\infty \sum_{\alpha \in \Xi_i \cap \Omega_i} \mathbb{B}_\alpha$$

$$+ \sum_{\mathbf{c} \in \mathcal{C}_i, \mathbf{m} \in \mathcal{M}_i} \mathbb{C}_{\mathbf{c},\mathbf{m},\alpha} \Pi_{j=1}^i \left(F_\pi^{(c_j)}(s_0; W^0) + \Phi^{(c_j)}(s_0; W^0)\Delta W^\tau\right)^{m_j} \left(I_\alpha[1]_{0,t} - I'_\alpha[1]_{0,t}\right) \tag{37}$$

Finally, note the remaining expression from that of $\Delta\tilde{s}_{t,\tau}$:

$$\sum_{i=1}^\infty \sum_{\alpha \in \Xi_i \cap \Omega_i} \mathbb{B}_\alpha + \sum_{\mathbf{c} \in \mathcal{C}_i, \mathbf{m} \in \mathcal{M}_i} \mathbb{C}_{\mathbf{c},\mathbf{m},\alpha} \quad \Pi_{j=1}^i \left(F_\pi^{(c_j)}(s_0; W^0) + \Phi^{(c_j)}(s_0; W^0)\Delta W^\tau\right)^{m_j} \left(I_\alpha[1]_{0,t} - I'_\alpha[1]_{0,t}\right),$$

which can be decomposed into two parts:

$$M_{t,\tau} = \sum_{i=1}^\infty \sum_{\alpha \in \Xi_i \cap \Omega_i} \mathbb{B}_\alpha + \sum_{\mathbf{c} \in \mathcal{C}_i, \mathbf{m} \in \mathcal{M}_i} \mathbb{C}_{\mathbf{c},\mathbf{m},\alpha} \Pi_{j=1}^i \left(F_\pi^{(c_j)}(s_0; W^0) + \Phi^{(c_j)}(s_0; W^0)\Delta W^\tau\right)^{m_j} I'_\alpha[1]_{0,t}$$

$$G_{t,\tau} = \sum_{i=1}^\infty \sum_{\alpha \in \Xi_i \cap \Omega_i} \mathbb{B}_\alpha + \sum_{\mathbf{c} \in \mathcal{C}_i, \mathbf{m} \in \mathcal{M}_i} \mathbb{C}_{\mathbf{c},\mathbf{m},\alpha} \Pi_{j=1}^i \left(F_\pi^{(c_j)}(s_0; W^0) + \Phi^{(c_j)}(s_0; W^0)\Delta W^\tau\right)^{m_j} I_\alpha[1]_{0,t},$$

where $M_{t,\tau} = \tilde{s}_{t,\tau} - \mathbb{E}[\tilde{s}_{t,\tau}]$, where the expectation is about the randomness of the stochastic process $\tilde{s}_{t,\tau}$, and is therefore the martingale component of $\tilde{s}_{t,\tau}$, conditioned on $\tilde{s}_{t,\tau}$.

$$M_{t,\tau} = \tilde{s}_{t,\tau} - s_0 - \int_0^t (g(\tilde{s}_{u,\tau}) + h(\tilde{s}_{u,\tau})a_{u,\tau})du$$

Similarly, $G_{t,\tau}$ is the martingale part of an independent instantiation of the process $\tilde{s}_{t,\tau}$, because we do not condition on $I_\alpha[0,t]$. Putting all these different components together in Equation 37 we have the statement of the Lemma. $\qquad\square$

## M   MAIN RESULT: PUTTING IT ALL TOGETHER

In the previous sections we derive the sufficient statistics and evolution equations for the value estimate (Section F), action (Section J), and state (Section L). Putting all these components together, we can derive a closed system and summary statistics required to describe the gradient dynamics of the actor critic algorithm described in Sections 4, E under the assumptions of Section 3. We prove the main result for gradient time $\tau$, which is an integer multiple of $\eta$ and a 1-dimensional system i.e. $d_s = d_a = 1$ in finite time $T < 1$.

*Proof.* The expression for $\Delta s_{t,\tau}$ comes from the lemma L.1. The expression for $\Delta a_{t,\tau}, \Delta a'_{t,\tau}$ follows from the lemmas K.1, K.2. The expression for the change in $\Delta v_{t,\tau}, \Delta v'_{t,\tau}$ follows from the lemmas I.1, I.2. We conclude the proof by stating that the system is closed up to an error of $O(1/\sqrt{n})$: the changes in all these variables for a single gradient step depend only on each other. $\qquad\square$

## N   CODE FOR LINEARIZED ACTOR AND CRITIC

We use the cleanrl repository to simplify our implementation in section **??**. The code for the actor and critic are modified as follows. First we present LQR environment code block:

Listing 1: Environment construction for continuous-time LQR control.

```
register(
    id="LQRd-v1",
    entry_point="custom_envs.lqr_d_env:LQRdEnv",
    nondeterministic=True,
)

def make_env(env_id, seed, idx, capture_video, run_name,
             A, B, Q, R, Qf,
             dt=0.02, T=1.0,
             Sigma=None, x0_mean=None, x0_std=0.0,
             u_max=20.0, exp_noise=0.05):

    def thunk():
        kwargs = dict(
            A=A,
            B=B,
            Q=Q,
            R=R,
            Qf=Qf,
            dt=dt,
            T=T,
            Sigma=Sigma,
            x0_mean=x0_mean,
            x0_std=x0_std,
            u_max=u_max,
            seed=seed,
            exp_noise=exp_noise,
        )
```

```
30
31            if capture_video and idx == 0:
32                env = gym.make(env_id, render_mode="rgb_array", **kwargs)
33                env = gym.wrappers.RecordVideo(env, f"videos/{run_name}")
34            else:
35                env = gym.make(env_id, **kwargs)
36
37            env = gym.wrappers.RecordEpisodeStatistics(env)
38            env.action_space.seed(seed)
39            return env
40
41        return thunk
42
43    # Example: scalar LQR with diagonal A, B, Q, R
44    d = args.data_dim
45    m = args.action_dim
46
47    A = -0.5 * np.eye(d)
48    B = np.eye(d, m)
49    Q = args.reward_scale * np.eye(d)
50    R = np.zeros((m, m))
51    Qf = np.zeros((d, d))
52    Sigma = args.process_noise * np.eye(d)
53
54    lqr_env = make_env(
55        "LQRd-v1",
56        seed=123 + args.seed,
57        idx=1,
58        x0_mean=1.0,
59        x0_std=0.0,
60        capture_video=False,
61        run_name="test_run",
62        A=A, B=B, Q=Q, R=R, Qf=Qf,
63        dt=0.02, T=1.0, Sigma=Sigma,
64        exp_noise=args.exploration_noise,
65        u_max=args.u_max,
66    )
67
68    envs = gym.vector.SyncVectorEnv([lqr_env])
```

The linearized actor and critic code blocks are as follows:

Listing 2: Linearized actor and critic networks for continuous-time AC.

```
1
2   def tanh_gradient(x: torch.Tensor) -> torch.Tensor:
3       y = torch.tanh(x)
4       grad_tanh = 1 - y ** 2
5       return grad_tanh
6
7   class VNetwork(nn.Module):
8
9       def __init__(self, env, width: int = 256):
10          super().__init__()
11          self.width = width
12          self.state_dim = np.array(env.single_observation_space.shape).
                prod() + 1  # +1 for time
13
14          # initial weights
15          self.fc1 = nn.Linear(self.state_dim, self.width, bias=False)
16          nn.init.normal_(self.fc1.weight, mean=0.0, std=1 / self.
                state_dim)
17
18          # linearization copy (trainable)
```

```python
19          self.fc1_copy = nn.Linear(self.state_dim, self.width, bias=
                False)
20          self.fc1_copy.load_state_dict(self.fc1.state_dict())
21
22          self.v_head = nn.Linear(self.width, 1, bias=False)
23          nn.init.normal_(self.v_head.weight, mean=0.0, std=1.0 / np.sqrt
                (self.width))
24
25      def forward(self, x_with_time: torch.Tensor) -> torch.Tensor:
26          # x_with_time has shape (B, state_dim+1)
27          preactivation_init = self.fc1(x_with_time)
28          init_intermediate = tanh_gradient(preactivation_init)
29          intermediate_linear = self.fc1_copy(x_with_time) -
                preactivation_init
30          h = torch.tanh(preactivation_init) + intermediate_linear *
                init_intermediate
31          v = self.v_head(h) / np.sqrt(self.width)
32          return v.squeeze(-1)   # (B,)
33
34  class Actor(nn.Module):
35
36      def __init__(self, env, width: int = 256):
37          super().__init__()
38          self.width = width
39          self.state_dim = np.array(env.single_observation_space.shape).
                prod()
40
41          self.fc1 = nn.Linear(self.state_dim, self.width, bias=False)
42          nn.init.normal_(self.fc1.weight, mean=0.0, std=1 / self.
                state_dim)
43
44          self.fc1_copy = nn.Linear(self.state_dim, self.width, bias=
                False)
45          self.fc1_copy.load_state_dict(self.fc1.state_dict())
46
47          self.fc_mu = nn.Linear(self.width,
48                                 np.prod(env.single_action_space.shape),
49                                 bias=False)
50
51          # action rescaling (Box space)
52          self.register_buffer(
53              "action_scale",
54              torch.tensor(
55                  (env.action_space.high - env.action_space.low) / 2.0,
56                  dtype=torch.float32,
57              ),
58          )
59          self.register_buffer(
60              "action_bias",
61              torch.tensor(
62                  (env.action_space.high + env.action_space.low) / 2.0,
63                  dtype=torch.float32,
64              ),
65          )
66
67      def forward(self, x: torch.Tensor) -> torch.Tensor:
68          preactivation_init = self.fc1(x)
69          init_intermediate = tanh_gradient(preactivation_init)
70          intermediate_linear = self.fc1_copy(x) - preactivation_init
71          h = torch.tanh(preactivation_init) + intermediate_linear *
                init_intermediate
72          out_x = self.fc_mu(h) / np.sqrt(self.width)
73          return out_x  # actions will be clipped by the caller
```

We optimize these using SGD and in an online episodic manner.

## O   LLM USAGE AND REPRODUCIBILITY STATEMENT

For our work, we used large language models for supportive, non-substantive tasks: we rely on them mainly for discovery (e.g., quickly locating references or related concepts), checking grammar and readability in the drafts, and clarifying technical notions when we need a different perspective to aid understanding. Additionally, we use them for code snippets. All core research contributions, proofs, experiments, and arguments are developed independently.

**Reproducibility Statement.** Our work is primarily theoretical, and we have provided complete proofs of all claims in the appendix along with detailed explanations of the assumptions underlying our results. For empirical validation, we include a toy continuous control experiment in the main text (Section 7) with full details of the environment dynamics, parameter initialization, and training setup, ensuring that the experiment can be replicated without ambiguity. Since the empirical component is intentionally simple and illustrative, and the theoretical framework is fully specified with proofs, the results presented in this paper can be readily reproduced using the information provided.

