# OpenReview forum: "From Ticks to Flows: Dynamics of Neural Reinforcement Learning in Continuous Environments"
_ICLR.cc/2026/Conference — ICLR 2026 Poster_

### Official Review · Reviewer_Rcmf · 2025-10-27

**Soundness:** 3
**Presentation:** 3
**Contribution:** 4
**Rating:** 8
**Confidence:** 4

**Summary:**

This article presents a continuous-time formulation of the dynamics of episodic actor-critic reinforcement learning. The goal of the paper is to provide a characterisation of the coupling between the environment and learning dynamics, going beyond the usual separation of these along their independent time scales. The authors build upon a range of existing works which bring stochastic analysis and control tools to bear in the analysis of RL (exploratory dynamics, Itô expansions, numerical schemes *à la* Kushner/Dupuis). Of course, this problem can’t be captured without further assumptions. The author’s assumptions are quite reasonable: smooth control-affine dynamics (which I would say is already above average complexity), linearised 2-layer hyperbolic tangent neural networks for the agent and critic. Combining infinite-width approximations of the neural nets with statistical theorems for the environment allows for the results, which are asymptotic normality characterisations of the changes in the value and the policy. This approach is representative of the ongoing convergence of stochastic analysis/control and Neural Network limits for RL, and is one of its first convincing and tangible applications to reinforcement learning.

**Strengths:**

_Soundness and Clarity_:
- I appreciated the down-to-earth tone of the paper: the results are presented first and foremost for what they rigourously are, without resorting to window dressing. The authors present clear assumptions and clearly discuss their results in the context thereof, which I highlight amongst the broader soundness and rigour of the article.

_Presentation and Clarity_:
- Explanations of both RL methodology and control are quite extensive, and, I think, make the topics approachable to either community.

- Appendices are extensive and pedagogical. I would like to express my appreciation, in particular, for the efforts the authors put into having examples throughout the background, which bridge over to the calculations performed afterwards.

_Contribution, Quality, & Originality_: The approach taken by the authors is a difficult but promising one, and the results are noteworthy both for their inherent quality and the future avenues of research they seem to make plausible.

**Weaknesses:**

_Soundness_:
- The experimental section is quite underwhelming. Section 7.1 presents a clear visual argument and is fine (except the above point about graph legibility), but section 7.2 hardly contributes to the paper in my opinion. Note that I do not consider such an experimental validation necessary.

_Presentation_:
- Figures 2 and 3 are illegible due to the legend and labels being too small.

- The appendices sorely need a table of contents. While the calculations themselves are detailed and well-explained, it is hard to keep track of the bigger picture due to their sheer size. Perhaps the authors would like to consider proof diagrams to better connect the different parts of the appendices.

- There are quite a few typos in the body, the references, and the appendices. Most of them are inconsequential, but they distract the reader in an already dense and, at times, disorienting work.

**Questions:**

I have many *research* questions about the work, but none would affect my evaluation of it. Thus, I will refrain from bothering the authors and reviewers with them.

---

> ### Author Response · Authors · 2025-11-23
> **Response to Reviewer Rcmf**
>
> We are grateful for your kind words and an in depth review of our work. We apologize for the typos and have fixed most of them both in Appendix and the main body. We respond to your concerns below. We have also fixed the font sizes on graphs for legibility.
>
> > section 7.2 hardly contributes to the paper
>
> While we recognize how this might be the case from a theoretical perspective, we believe that demonstrating that the algorithm we have presented works satisfactorily in a toy setting is helpful for the broader community. We have slightly modified the exposition in Section 7.2 to make this point and hope it is to your satisfaction.
>
> > appendices sorely need a table of contents
>
> We have now added a table of content in the Appendix which will help people navigate better.
>
> > consider proof diagrams to better connect the different parts of the appendices
>
> We now provide a proof diagram in the main body of the paper (Figure 2) with clickable links to the lemmas. We hope this is helpful.
>
> Other than the issues mentioned above, we would be very glad to field your research questions in the future and thank you once again for your encouragement and engagement with our work.

---

> > ### Comment · Reviewer_Rcmf · 2025-11-26
> >
> > I have seen the above response and thank the authors for the effort put into the revision.
> > I have made no changes to my review.

---

### Official Review · Reviewer_nz1G · 2025-10-30

**Soundness:** 4
**Presentation:** 4
**Contribution:** 4
**Rating:** 6
**Confidence:** 1

**Summary:**

This paper proposes a continuous-time theoretical framework for neural reinforcement learning (RL), bridging stochastic control and deep RL theory. The authors model actor-critic learning as a two-timescale stochastic process with an environment time and a gradient time and derive stochastic differential equations that describe the infinitesimal evolution of the state distribution under infinite-width linearized neural networks with vanishing learning rates. Theoretical results are complemented by small-scale empirical validation on a linear-quadratic regulator (LQR) environment.

**Strengths:**

- The combination of NTK methods from deep learning to study gradient updates and SDEs from control theory to study environment updates to provide a unified nonparametric framework across two timescales is neat
- Clear theoretical construction by leveraging Ito-Taylor expansions and martingale CLTs to formally connect gradient-time dynamics and environment evolution
- The LQR experiment confirms that the proposed dynamics can be simulated and produce expected theoretical behavior

**Weaknesses:**

- The entire analysis relies on single hidden layer networks, smooth dynamics, and small learning rates. These conditions, while analytically convenient, make the results difficult to generalize to realistic RL systems.
- The key theorems depend on the infinite-width and vanishing learning-rate limits. No discussion is provided on how these approximations break down for practical networks, nor whether finite-width corrections could meaningfully affect dynamics.

**Questions:**

N/A

---

> ### Author Response · Authors · 2025-11-23
> **Response to Reviewer nz1G**
>
> We thank the reviewer for their positive review and for recognizing the strengths of our work. We clarify the two issues they have raised below.
>
> > The entire analysis relies on single hidden layer networks, smooth dynamics, and small learning rates. These conditions, while analytically convenient, make the results difficult to generalize to realistic RL systems
>
> While it is true that this is a purely theoretical model of continuous RL, presenting a non-parametric model of neural actor-critic opens new ideas of research as it does for supervised learning [1]. This is a step towards better understanding why much more complex deep RL systems work in continuous environments, which remains a blackbox. It is a move towards addressing a fundamental gap in our understanding. We now acknowledge that our results are in the “lazy regime” and there are possible ways to extend it to finite width models in the **related works section (Section 8)**. Our long term goal is to arrive at theoretically grounded but also empirically performant RL systems which we now discuss as part of future work in Section 9.
>
> > No discussion is provided on how these approximations break down for practical networks
>
> Thank you for pointing out this is missing. The finite width error here is of the order $O(1/\sqrt{n})$, while this is not ideal it gives us the error rate at which our model reflects a finite width model. We also now acknowledge (in the latest version) in the related work section that this is indeed not a perfect reflection of practice. All theoretical models, such as ours that are based on deep learning theory, have to make some simplifying assumption to keep the problem tractable. As far as how the learning dynamics are affected in the finite width regime, our current assessment is that in practical networks the feature representations in the intermediate layer change a lot over gradient steps and therefore lead to faster learning. This has been noticed for supervised learning [3].  Moreover, adding momentum in SGD and methods like Adam also speed up learning as opposed to online SGD in our case. We discuss these shortcomings in **Section 9 and related work section of the latest version**.
>
> Please take a look and let us know if this is satisfactory and if you'd be able to raise the score.
>
> ---------------------------
>
> References:
>
> [1]  Wide neural networks of any depth evolve as linear models under gradient descent, Jaehoon Lee, Lechao Xiao, Samuel Schoenholz, Yasaman Bahri, Roman Novak, Jascha Sohl-Dickstein, and Jeffrey Pennington. Neruips 2019
>
> [2] Finite Depth and Width Corrections to the Neural Tangent Kernel, Boris Hanin and Mihai Nica, 2020
>
> [3] Feature Learning in Infinite-Width Neural Networks, Greg Yang, Edward J. Hu, 2020

---

> > ### Comment · Reviewer_nz1G · 2025-11-24
> >
> > Thank you for the response! I am not an expert in this domain and I defer to the other reviewers.

---

### Official Review · Reviewer_xDen · 2025-10-31

**Soundness:** 3
**Presentation:** 3
**Contribution:** 3
**Rating:** 6
**Confidence:** 2

**Summary:**

This paper presents a novel theoretical framework for deep RL in continuous environments by modeling the problem as a continuous-time stochastic process, and it also provides a novel nonparametric formulation for studying overparametrized neural actor-critic algorithms.

**Strengths:**

The paper provides a strong theory for continuous time actor-critic model; providing a nonparametric formulation for studying overparameterized neural actor-critic algorithms.

**Weaknesses:**

The numerical experiments are very limitted, and only one-hidden-layer neural networks are considered.

**Questions:**

1. Extension to deeper architectures (multiple hidden layers): this would significantly strengthen the contribution. Also, please discuss the issue and add numerical experiments.

2. Evaluation across diverse environments: please evaluate the method in multiple environments that vary in dynamics and difficulty.

---

> ### Author Response · Authors · 2025-11-23
> **Response to Reviewer xDen**
>
> We thank you for their praise of our work and for considering our contributions to be positive. We address your concerns below and hope to hear back from you.
>
> > The numerical experiments are very limited, and only one-hidden-layer neural networks are considered and Extension to deeper architectures
>
> In the current work we limit our scope to a simplified setting to maintain tractability. In the latest updated version we discuss how a linearized model of deeper NNs can be modeled in a similar manner to the work by Lee et al [1] in the related works section. We also discuss the limitations of our theoretical model and possible extensions in **Section 8 on related work and Section 9 discussion**. We also position our work in light of similar work, extending to deeper architectures and complex RL algorithms is beyond the scope of current work and definitely promising direction of future work.
>
> > Evaluation across diverse environments: please evaluate the method in multiple environments that vary in dynamics and difficulty
>
> Our results are largely theoretical and therefore our gradient based method has to be very simple (online SGD) and might not translate to complex robotic control environments. We have now added **LQR results in higher-dimensional environments in Section 7.2**, where our proposed algorithm learns near optimal policies for $d_s = 2, 8, 32$ in the LQR setting. We point the reviewer to [2] where they show that for a soft actor critic agent, with various algorithmic scaffolds, linearized NNs behave similar to canonical NNs in the complex Cheetah environment. We now state this explicitly in Section 5. This implies that the linearized models reflect the success of canonical NNs. Our simplified algorithm, with simple SGD and not Adam, without target networks, without multiple q-functions etc. can reliably learn in complex MuJoco or Deepmind control environments. Nonetheless, our theory provides a *metaphor* to study RL in continuous states and environments and applies to simple cases such as LQR. This keeps the theory tractable.
>
> We hope this helps address your concerns and look forward to hearing back from you.
>
> ----------------------------------
>
> References:
>
> [1]  Wide neural networks of any depth evolve as linear models under gradient descent, Jaehoon Lee, Lechao Xiao, Samuel Schoenholz, Yasaman Bahri, Roman Novak, Jascha Sohl-Dickstein, and Jeffrey Pennington. Neruips 2019
>
> [2] Saket Tiwari, Omer Gottesman, and George Konidaris. Geometry of neural reinforcement learning in continuous state and action spaces, ICLR 2025

---

### Official Review · Reviewer_mQkp · 2025-10-31

**Soundness:** 4
**Presentation:** 2
**Contribution:** 3
**Rating:** 6
**Confidence:** 1

**Summary:**

A framework for continuous-time actor-critic RL is introduced that accounts for noisy envrionments. Prior work is adapted by including exploratory stochastic dynamics that can be simulated faithfully in discrete time. Linearized single hidden layer neural networks are used for the actor and critic functions, allowing the formulation of a system of equations that captures the evolution of state over environment time and gradient steps up to an error term. Expressions for the gradients with respect to actor and critic parameters in continuous time are derived and used to create an episodic actor-critic algorithm. The algorithm is evaluated empirically on a one-dimensional toy example and superior exploration is shown when compared to an additive Wiener process.

**Strengths:**

The preliminaries are introduced in a succinct manner. The proposed framework appears to be very thorough incorporating all aspects of stochasticity in RL. The proof is lengthy and presumably rigorous. An algorithm listing sheds light on the presented approach.

**Weaknesses:**

The presented math is overwhelming and confusing at times with apparent inconsistencies of upper and lower case in symbols and of argument order.

It is stated early in the paper that "higher dimensional results follow" wich is later restated as "We believe high-dimensional results should follow.".

The empirical evaluation is very limited. Additional settings should be explored including higher-dimensional tasks.
The figures are very small and hard to read.

Minor errors:
- Line 82: "the RL agent can is"
- Line 108: $\Delta W$ suddenly upper case
- Lemma 4.2: First $v(x,t)$ then $v(T,x)$, are the arguments switched?
- Line 226: "The deterministic policy analog of Theorem 2 by Jia & Zhou (2022) gives an expression for **dicrete** time **the** policy gradient (Sutton et al., 1999) in continuous time setting"
- Theorem 4.3: the gradient is first lower case $g(t,x;\theta)$ then upper case $\mathcal G(t,x;\theta)$.
- Line 235: "... by sampling a single trajectory **to** and updating ..."
- Therome 6.1: extra word "estimate"?
- Line 376: "$d_{s}=d_{s}=1$", subscript is repeated

**Questions:**

- Why does the state-value function depend on time?
- You state that "Although Jia & Zhou (2022) provides empirical validation for a similar algorithm, they do not do so for the neural network actor and critic." Can reiterate the novelty in your work compared to theirs?
- Does the presented framework allow for an extension to multi-layer finit-width networks? Can you elaborate on how this might look like?
- A problem with infinite-width neural networks is that the gradients become neglibly small, effectively renouncing the capability for feature learning. Is this of relevance here?

---

> ### Author Response · Authors · 2025-11-23
> **Response to Reviewer mQkp**
>
> Thank you for the positive comments on the thoroughness of our work and the rigor. We also appreciate you taking the time to point out all the typos and sincerely apologize for them, we have fixed these in the newest version. We address all your concerns below.
>
> > Presented math is overwhelming and confusing
>
> We are now explaining the math, to the best of our ability, by tying it to RL in Sections 2, 3 and 4. We have also removed some notation and statements in Section 4 which are not essential to our work. We have also fixed the typos (especially the confusion with the arguments of value function). We hope this is helpful. Unfortunately, we are restricted in what we can meaningfully explain.
>
> > “higher dimensional results follow” and “We believe high-dimensional results should follow”
>
> The first one was for equivalence of exploratory dynamics (Lemma 3.1) and we anticipate the higher-dimensional results would be similar and easy to prove as is common in control [3,4]. We do not provide these as of now. We have rephrased it to: “We anticipate that higher dimensional results follow similarly”.
>
> The second one was for our main result. After thinking through our main result (Theorem 6.1) for higher-dimensional systems we believe that it is fairly non-trivial and should be subject to future work. We now **note this in section 6** and also in the discussion.
>
> > empirical evaluation is very limited. Additional settings should be explored including higher-dimensional tasks
>
> We have added **high-dimensional LQR results in Section 7.2**. The same algorithm achieves near-optimal performance under a linearized neural-network parameterization for $d_s = 2, 8, 32$. Tiwari et al. 2025 [4] similarly show that linearized networks – for a soft actor–critic setup – recover the performance of standard neural networks at large widths; we note this in Section 5.
>
> We are not able to scale the simplified version of our algorithm, without any scaffolding such as target networks, Adam, or critic stabilization techniques, to complex MuJoCo or DeepMind Control environments. Our main contribution is theoretical, and we expect that introducing theoretically tractable online episodic models will open the doors to empirical advances.
>
> > Why does the state-value function depend on time?
>
> For finite horizon RL the value of an agent being at state $s$ at time $t$ depends on the time. To understand this consider the following discrete time example:
> $$ V(s, t) = \mathbb E \left [ \sum_{i= t}^{T} \gamma^{i - t} r(s_i) | \pi, s_t = s \right ]$$
>
> And for time $t’ \neq t$:
> $$ V(s, t’) = \mathbb E \left [ \sum_{i= t’}^{T} \gamma^{i - t’} r(s_i) | \pi, s_{t’} = s \right ].$$
>
> Since the number of summands is not the same across two expectations it need not be that $V(s, t) = V(s, t’) $. Intuitively, if the agent has a different number of time steps remaining until termination it could collect different cumulative discounted rewards across those time steps. Therefore, for finite horizon problems, the value function depends on time. This is not the case in the infinite horizon case as is commonly stated.
>
> > Can you reiterate the novelty in your work compared to the work by Jia & Zhou …
>
> We have three novel contributions:
> 1. The exploratory dynamics are different in our case and truly exploratory even in a deterministic setting (see Section 3).
> 2. We present a non-parametric gradient update equivalent to parametric gradient based algorithms for deep RL in continuous time, whereas they provide a parametric gradient update model for continuous time RL.
> 3. We also show that linearized NNs learn successfully using the algorithm we have presented in LQR environments; this presents empirical backing that our theoretical model of neural networks applies to RL.
>
> We now re-assert point 1 in **Section 3**, where we introduce continuous time RL, and points 2 and 3 in the **Section 9**.
>
> > Does the presented framework allow for an extension to multi-layer finite-width networks? Can you elaborate on how this might look like?
>
> Deep learning theory employs a linearized model that extends to multi-layer infinite width NNs [1]. As for finite width, we show that the error from the proposed learning dynamics is $O(1/\sqrt{n})$ where $n$ is the width, meaning we show how far our theoretical model is from a finite width NN. But more broadly, there has been work which generalises works to finite widths in supervised learning [2] and we hope to incorporate this theoretical model of neural networks in the future into the study of deep RL.For now it remains beyond the scope of our work. **We now comment on this in the related works section (Section 8)**, thank you for pointing it out.

---

> > ### Author Response · Authors · 2025-11-23
> > **Response to Reviewer mQkp continued**
> >
> > > infinite-width neural networks is that the gradients become neglibly small, effectively renouncing the capability for feature learning
> >
> > Yes, this is indeed a problem here although learning is still feasible in this regime despite gradients being very small. This model of learning dynamics is often referred to as the “lazy regime” for neural networks [5]. Moreover, there is no feature learning in this setting. We have now added this discussion in the **related works section (Section 8)**. Please take a look. As an extension, we hope to incorporate feature learning into the theory of Deep RL in the future.
> >
> > > The figures are very small and hard to read.
> >
> > We have now fixed the font sizes and apologize for the difficulty.
> >
> > We hope we have resolved all your concerns and queries! Please let us know.
> >
> > -----------------------------------
> >
> > **References:**
> >
> > [1]  Wide neural networks of any depth evolve as linear models under gradient descent, Jaehoon Lee, Lechao Xiao, Samuel Schoenholz, Yasaman Bahri, Roman Novak, Jascha Sohl-Dickstein, and Jeffrey Pennington. Neruips 2019
> >
> > [2] Finite Depth and Width Corrections to the Neural Tangent Kernel, Boris Hanin and Mihai Nica, 2020
> >
> > [3] Numerical Methods for Stochastic Control Problems in Continuous Time, 2001, Harold J. Kushner and Paul Dupuis.
> >
> > [4] Geometry of neural reinforcement learning in continuous state and action spaces, Saket Tiwari, Omer Gottesman, and George Konidaris, ICLR 2025
> >
> > [5] Limitations of lazy training of two-layers neural network, Ghorbani, Behrooz and Mei, Song and Misiakiewicz, Theodor and Montanari, Andrea

---

> > > ### Comment · Reviewer_mQkp · 2025-11-26
> > >
> > > Thank you for the clarifications. Please understand that I am not able to adequately assess the novelty and rigour of your contribution. Therefore, I will keep my assessment unchanged.

---

### Author Response · Authors · 2025-11-25
**Overall Rebuttal**

We thank the reviewers for their careful reading and constructive feedback. We appreciate the time they invested in evaluating our submission. The revised draft incorporates all requested clarifications and additions, and we summarize the main updates below.
In addressing the comments we have made the following improvements:

 1. We expanded the explanations of the mathematical tools used in the paper, grounding each step in standard reinforcement-learning concepts and clarifying how the pieces fit into the overall argument.

2. We added experiments in higher-dimensional settings, now reporting results for $d_s = 2, 8, 32$.

3. We provided a clearer discussion of the limitations of our theoretical model in the context of the broader deep-learning theory literature. This now includes comments on finite-width corrections and feature-learning regimes, and outlines concrete future directions opened by our analysis.

4. We added a proof-sketch figure that illustrates how the key intermediate results connect to the main theorem.

**We have marked all the changes in red**. We have additionally fixed all the minor issues and incorporated the comments. We thank the reviewers again for recognizing the thoroughness and novelty of the work and the avenues it creates for further research.

---

### Author Response · Authors · 2025-12-03
**Final Rebuttal Summary**

Thank you once again to all reviewers for their time and acknowledgement of our rebuttal.

Dear AC,

We summarize all the changes and how we have addressed individual reviewer comments below:

**Empirical evaluation (mQkp, xDen)**
Added higher-dimensional LQR results ($d_s = 2, 8, 32$) and clarified why the simple online-SGD dynamics cannot scale to complex robotics tasks.

**Exposition and clarity (mQkp, Rcmf):**
Simplified Sections 2–4, fixed notation/typos, improved figures, added appendix table of contents and a proof-flow diagram.

**Higher-dimensional theory and scope (mQkp):**
Clarified which results generalize and stated explicitly that extending the main theorem to higher dimensions is non-trivial and future work.

**Finite-width networks and feature learning (mQkp, xDen, nz1G):**
Added discussion of finite-width error $O(1/\sqrt{n})$, the lazy regime, absence of feature learning, and how practical networks are related.

**Positioning and novelty (mQkp, xDen):**
Reasserted key contributions and clarified how the framework differs from prior continuous-time RL and linearized deep network work.

Hope this helps in the decision making process.

---

### Meta-Review · Area_Chair_EC5t · 2026-01-05

**Summary:**

This work establishes a theoretical framework for reinforcement learning by modeling the problem as a continuous-time stochastic process. Leveraging Stochastic Differential Equations (SDEs) and the Neural Tangent Kernel (NTK) regime, the authors derive the evolution of state distributions during gradient updates.

There is a significant expertise mismatch in the reviews. Three reviewers (mQkp, xDen, nz1G) reported low confidence (1-2) given the heavy reliance on stochastic analysis and SDEs. In contrast, Reviewer Rcmf strongly supported the paper (Score: 8), praising its rigor and novelty. The main discussions focused on:

(1) Theoretical Assumptions: The reliance on infinite-width, linearized networks and the lack of feature learning.

(2) Experimental Scope: The limitation of empirical validation to Linear Quadratic Regulator (LQR) tasks versus complex benchmarks.

(3) Accessibility: The complexity of the mathematical notation and SDE formulations.

**Reviewer Concerns:**

Addressed Concerns:

- High-Dimensional Verification (Reviewer mQkp, xDen): In response to requests for higher-dimensional evaluation, the authors provided LQR results with $d_s=32$, confirming that the linearized network policy converges as theoretically predicted.
- Clarity & Typos (Reviewer mQkp, Rcmf): The authors corrected notation inconsistencies and added a "proof flow diagram" (Figure 2) to help navigate the theoretical derivations, which addressed Reviewer Rcmf's suggestion.
- Finite-Width Limitations (Reviewer mQkp, nz1G): The authors acknowledged the lack of feature learning under the "lazy regime" assumption but justified it as a necessary trade-off for theoretical tractability, aligning with recent learning theory literature.

Outstanding Concerns:

- Experimental Complexity (Reviewer xDen, mQkp): Several reviewers were concerned that the experiments were limited to LQR.

**Reviewer Scores:**

- Reviewer Rcmf (Initial: 8): 8 (Maintain). This reviewer was the most engaged and positive, praising the work as a "convincing and tangible application" of stochastic analysis to RL.
- Reviewer mQkp (Initial: 6): 6 (Maintain). Explicitly stated "I am not able to adequately assess the novelty and rigour... I will keep my assessment unchanged" due to lack of expertise.
- Reviewer xDen (Initial: 6): 6 (Maintain). The reviewer's concerns were generic (requesting diverse environments), and it is unlikely the theoretical rebuttal significantly shifted their perspective.
- Reviewer nz1G (Initial: 6): 6 (Maintain). Deferred to other reviewers due to lack of domain expertise.

---

### Decision · Program_Chairs · 2026-01-26

Accept (Poster)